# A PRIMAL-DUAL PERSPECTIVE FOR DISTRIBUTED TD-LEARNING

## ABSTRACT

The goal of this paper is to investigate distributed temporal difference (TD) learning for a networked multi-agent Markov decision process. The proposed approach is based on distributed optimization algorithms, which can be interpreted as primal-dual Ordinary differential equation (ODE) dynamics subject to null-space constraints. Based on the exponential convergence behavior of the primal-dual ODE dynamics subject to null-space constraints, we examine the behavior of the final iterate in various distributed TD-learning scenarios, considering both constant and diminishing step-sizes and incorporating both i.i.d. and Markovian observation models. Unlike existing methods, the proposed algorithm does not require the assumption that the underlying communication network structure is characterized by a doubly stochastic matrix.

## 1 INTRODUCTION

Temporal-difference (TD) learning (Sutton, 1988) aims to solve the policy evaluation problem in Markov decision processes (MDPs), serving as the foundational pillar for many reinforcement learning (RL) algorithms (Mnih et al., 2015). Following the empirical success of RL in various fields (Kober et al., 2013; Li et al., 2019), theoretical exploration of TD learning has become an active area of research. For instance, Tsitsiklis and Van Roy (1996) studied the asymptotic convergence of TD learning, while non-asymptotic analysis has been examined in Bhandari et al. (2018); Srikant and Ying (2019); Lee and Kim (2022).

In contrast to the single-agent case, the theoretical understanding for TD-learning for networked multi-agent Markov decision processes (MAMDPs) has not been fully explored so far. In the networked MAMDPs, each agent follows its own policy and receives different local rewards while sharing their local learning parameters through communication networks. Under this scenario, several distributed TD-learning algorithms (Wang et al., 2020; Doan et al., 2019; 2021; Sun et al., 2020; Zeng et al., 2022) have been developed based on distributed optimization frameworks (Nedic and Ozdaglar, 2009; Pu and Nedić, 2021).

The main goal of this paper is to provide finite-time analysis of a distributed TD-learning algorithm for networked MAMDPs from the perspectives of the primal-dual algorithms (Wang and Elia, 2011; Mokhtari and Ribeiro, 2016; Yuan et al., 2018). The proposed algorithms are inspired by the control system model for distributed optimization problems Wang and Elia (2011); Lee (2023), and at the same time, it can also be interpreted as the primal-dual gradient dynamics in (Qu and Li, 2018). In this respect, we first study finite-time analysis of continuous-time primal-dual gradient dynamics in (Qu and Li, 2018) with special nullity structures on the system matrix. Based on the analysis of primal-dual gradient dynamics, we further provide a finite-time analysis of the proposed distributed TD-learning under both i.i.d. observation and Markov observation models. The main contributions are summarized as follows:

1. An improved or comparable to the state of art convergence rate for continuous-time primal-dual gradient dynamics (Qu and Li, 2018) with null-space constraints under specific conditions: the results can be applied to general classes of distributed optimization problems that can be reformulated as saddle-point problems (Wang and Elia, 2011; Mokhtari and Ribeiro, 2016; Yuan et al., 2018);

2. Development of new distributed TD-learning algorithms inspired by (Wang and Elia, 2011; Lee, 2023);

3. New mean-squared error bounds of the distributed TD-learning under our consideration for both i.i.d. and Markovian observation models and under various conditions of the step-sizes: the distributed TD-learning is based on the control system model in (Wang and Elia, 2011; Lee, 2023) which does not require doubly stochastic matrix corresponding to its associated network graph. Note that the doubly stochastic assumption is required in other distributed TD-learning algorithms based on the classical distributed optimization algorithms (Nedic and Ozdaglar, 2009; Pu and Nedić, 2021);

4. Empirical demonstrations of both the convergence and the rate of convergence of the algorithm studied in this paper.

**Related Works.** Nedic and Ozdaglar (2009) investigated a distributed optimization algorithm over a communication network whose structure graph is represented by a doubly stochastic matrix. In this approach, each agent exchanges information with its neighbors, with the exchange being weighted by the corresponding element in the doubly stochastic matrix. Wang and Elia (2011); Notarnicola et al. (2023) provided control system approach to study distributed optimization problem. Another line of research designs distributed algorithms based on primal-dual approach (Yuan et al., 2018; Mokhtari and Ribeiro, 2016).

The asymptotic convergence of distributed TD-learning has been studied in Mathkar and Borkar (2016); Stanković et al. (2023). Doan et al. (2019) provided finite-time analysis of distributed TD-learning based on the distributed optimization algorithm (Nedic and Ozdaglar, 2009) with i.i.d. observation model. Doan et al. (2021) extended the analysis of Doan et al. (2019) to the Markovian observation model. Sun et al. (2020) studied distributed TD-learning based on Nedic and Ozdaglar (2009) with the Markovian observation model using multi-step Lyapunov function (Wang et al., 2019). Wang et al. (2020) studied distributed TD-learning motivated by the gradient tracking method (Pu and Nedić, 2021). Zeng et al. (2022) studied finite-time behavior of distributed stochastic approximation algorithms (Robbins and Monro, 1951) with general mapping including TD-learning and Q-learning, using Lyapunov-Razumikhin function (Zhou and Luo, 2018). Hairi et al. (2021) studied distributed actor-critic (Konda and Tsitsiklis, 1999) where the critic update requires the size of mini-batch to be dependent on the number of agents.

In the context of policy evaluation, Macua et al. (2014); Lee et al. (2018); Wai et al. (2018); Ding et al. (2019); Cassano et al. (2020) studied distributed versions of gradient-TD (Sutton et al., 2009). The Gradient-TD method can be reformulated as saddle-point problem (Macua et al., 2014; Lee et al., 2022), and the aforementioned works can be understood as distributed optimization over a saddle-point problem (Boyd and Vandenberghe, 2004).

## 2 PRELIMINARIES

### 2.1 MARKOV DECISION PROCESS

Markov decision process (MDP) consists of five tuples $(\mathcal{S}, \mathcal{A}, \gamma, \mathcal{P}, r)$, where $\mathcal{S} := \{1, 2, \ldots, |\mathcal{S}|\}$ is the collection of states, $\mathcal{A}$ is the collection of actions, $\gamma \in (0, 1)$ is the discount factor, $\mathcal{P} : \mathcal{S} \times \mathcal{A} \times \mathcal{S} \to [0, 1]$ is the transition kernel, and $r : \mathcal{S} \times \mathcal{A} \times \mathcal{S} \to \mathbb{R}$ is the reward function. If action $a \in \mathcal{A}$ is chosen at state $s \in \mathcal{S}$, the transition to state $s' \in \mathcal{S}$ occurs with probability $\mathcal{P}(s, a, s')$, and incurs reward $r(s, a, s')$. Given a stochastic policy $\pi : \mathcal{S} \times \mathcal{A} \to [0, 1]$, the quantity $\pi(a \mid s)$ denotes the probability of taking action $a \in \mathcal{A}$ at state $s \in \mathcal{S}$. We will denote $\mathcal{P}^\pi(s, s') := \sum_{a \in \mathcal{A}} \mathcal{P}(s, a, s')\pi(a \mid s)$, and $\mathcal{R}^\pi(s) := \sum_{a \in \mathcal{A}} \sum_{s' \in \mathcal{S}} \mathcal{P}(s, a, s')\pi(a \mid s)r(s, a, s')$, which is the transition probability from state $s \in \mathcal{S}$ to $s' \in \mathcal{S}$ under policy $\pi$, and expected reward at state $s \in \mathcal{S}$, respectively. $d : \mathcal{S} \to [0, 1]$ denotes the stationary distribution of the state $s \in \mathcal{S}$ under policy $\pi$. The policy evaluation problem aims to estimate the expected sum of discounted rewards following policy $\pi$, the so-called the value function, $\boldsymbol{V}^\pi(s) = \mathbb{E}\left[\sum_{k=0}^\infty \gamma^k r(s_k, a_k, s_{k+1}) \big| s_0 = s, \pi\right], \ s \in \mathcal{S}$.

Given a feature function $\phi : \mathcal{S} \to \mathbb{R}^q$, our aim is to estimate the value function through learnable parameter $\boldsymbol{\theta}$, i.e., $\boldsymbol{V}^\pi(s) \approx \phi(s)^\top \boldsymbol{\theta}$, for $s \in \mathcal{S}$, which can be achieved through solving the optimization problem, $\min_{\boldsymbol{\theta} \in \mathbb{R}^q} \frac{1}{2} \|\boldsymbol{R}^\pi + \gamma \boldsymbol{P}^\pi \boldsymbol{\Phi} \boldsymbol{\theta} - \boldsymbol{\Phi} \boldsymbol{\theta}\|_{\boldsymbol{D}^\pi}^2$, where $\boldsymbol{D}^\pi := \mathrm{diag}(d(1), d(2), \ldots, d(|\mathcal{S}|)) \in \mathbb{R}^{|\mathcal{S}| \times |\mathcal{S}|}$, $\mathrm{diag}(\cdot)$ is diagonal matrix whose diagonal elements correspond to elements of the tu-

ple, $\boldsymbol{P}^\pi \in \mathbb{R}^{|\mathcal{S}| \times |\mathcal{S}|}$ whose elements are $[\boldsymbol{P}^\pi]_{ij} := \mathcal{P}^\pi(i,j)$ for $i,j \in \mathcal{S}$, $\boldsymbol{R}^\pi \in \mathbb{R}^{|\mathcal{S}|}$, $[\boldsymbol{R}^\pi]_i :=$ $\mathbb{E}\left[r(s,a,s')|s=i\right]$ for $i \in \mathcal{S}$, and $\boldsymbol{\Phi} := \begin{bmatrix} \phi(1) & \phi(2) & \cdots & \phi(|\mathcal{S}|) \end{bmatrix}^\top \in \mathbb{R}^{|\mathcal{S}| \times q}$. The solution of the optimization problem satisfies the so-called projected Bellman equation (Sutton et al., 2009): $\boldsymbol{\Phi}^\top \boldsymbol{D}^\pi \boldsymbol{\Phi} \boldsymbol{\theta} = \boldsymbol{\Phi}^\top \boldsymbol{D}^\pi \boldsymbol{R}^\pi + \gamma \boldsymbol{\Phi}^\top \boldsymbol{D}^\pi \boldsymbol{P}^\pi \boldsymbol{\Phi} \boldsymbol{\theta}$.

Throughout the paper, we adopt the common assumption on the feature matrix, which is widely used in the literature (Bhandari et al., 2018; Wang et al., 2020).

**Assumption 2.1.** $\|\phi(s)\|_2 \leq 1$ *for all* $s \in \mathcal{S}$ *and* $\boldsymbol{\Phi}$ *is full-column rank matrix.*

### 2.2 MULTI-AGENT MDP

Multi-agent Markov decision process (MAMDP) considers a set of agents cooperatively computing the value function for a shared environment. Considering $N$ agents, each agent can be denoted by $i \in \mathcal{V} := \{1, 2, \ldots, N\}$, and the agents communicate over networks that can be described by a connected and undirected simple graph $\mathcal{G} := (\mathcal{V}, \mathcal{E})$, where $\mathcal{E} \subset \mathcal{V} \times \mathcal{V}$ is the set of edges. $\mathcal{N}_i \subset \mathcal{V}$ denotes the neighbour of agent $i \in \mathcal{V}$, i.e., $j \in \mathcal{N}_i$ if and only if $(i,j) \in \mathcal{E}$ for $i,j \in \mathcal{V}$. Each agent $i \in \mathcal{V}$ has its local policy $\pi^i : \mathcal{S} \times \mathcal{A}_i \to [0,1]$, where $\mathcal{A}_i$ is the action space of agent $i$, and receives reward following its local reward function $r^i : \mathcal{S} \times \Pi_{i=1}^N \mathcal{A}_i \times \mathcal{S} \to \mathbb{R}$. As in the single-agent MDP, MAMDP consists of five tuples $(\mathcal{S}, \{\mathcal{A}_i\}_{i=1}^N, \mathcal{P}, \{r^i\}_{i=1}^N)$, where $\mathcal{P} : \mathcal{S} \times \{\mathcal{A}_i\}_{i=1}^N \times \mathcal{S} \to [0,1]$ is the Markov transition kernel. The agents share the same state $s \in \mathcal{S}$, and when action $\boldsymbol{a} := (a_1, a_2, \ldots, a_N) \in \Pi_{i=1}^N \mathcal{A}_i$ is taken, the state transits to $s' \in \mathcal{S}$ with probability $\mathcal{P}(s, \boldsymbol{a}, s')$, and for $i \in \mathcal{V}$, agent $i$ receives $r^i(s, \boldsymbol{a}, s')$. The aim of the policy evaluation under MAMDP is to estimate the expected sum of discounted rewards averaged over $N$ agents, i.e., $\boldsymbol{V}^\pi(s) = \mathbb{E}\left[\sum_{k=0}^\infty \gamma^k \frac{1}{N} \sum_{i=1}^N r^i(s_k, \boldsymbol{a}, s_{k+1})\right]$, for $s \in \mathcal{S}$. While learning, each agent $i \in \mathcal{V}$ can share its learning parameter over the communication network with its neighboring agents $j \in \mathcal{N}_i$.

Following the spirit of single-agent MDP, using the set of features $\boldsymbol{\Phi}$, the aim of each agent is now to compute the solution of the following equation:

$$\boldsymbol{\Phi}^\top \boldsymbol{D}^\pi \boldsymbol{\Phi} \boldsymbol{\theta} = \boldsymbol{\Phi}^\top \boldsymbol{D}^\pi \left(\frac{1}{N} \sum_{i=1}^N \boldsymbol{R}_i^\pi\right) + \gamma \boldsymbol{\Phi}^\top \boldsymbol{D}^\pi \boldsymbol{P}^\pi \boldsymbol{\Phi} \boldsymbol{\theta}, \tag{1}$$

where $\boldsymbol{R}_i^\pi \in \mathbb{R}^{|\mathcal{S}|}$ for $i \in \mathcal{V}$, whose elements are $[\boldsymbol{R}_i^\pi]_j = \mathbb{E}\left[r^i(s, \boldsymbol{a}, s') \mid s = j\right]$ for $j \in \mathcal{S}$. The equation (1) admits a unique solution $\boldsymbol{\theta}_c$, given by

$$\boldsymbol{\theta}_c = (\boldsymbol{\Phi}^\top \boldsymbol{D}^\pi \boldsymbol{\Phi} - \gamma \boldsymbol{\Phi}^\top \boldsymbol{D}^\pi \boldsymbol{P}^\pi \boldsymbol{\Phi})^{-1} \boldsymbol{\Phi}^\top \boldsymbol{D}^\pi \left(\frac{1}{N} \sum_{i=1}^N \boldsymbol{R}_i^\pi\right). \tag{2}$$

Note that the solution corresponds to the value function associated with the global reward $\sum_{k=0}^\infty \gamma^k \frac{1}{N} \sum_{i=1}^N r^i$. Moreover, for convenience of the notation, we will denote

$$\boldsymbol{A} := \gamma \boldsymbol{\Phi}^\top \boldsymbol{D}^\pi \boldsymbol{\Phi} - \boldsymbol{\Phi}^\top \boldsymbol{D}^\pi \boldsymbol{\Phi}, \quad \boldsymbol{b}_i = \boldsymbol{\Phi}^\top \boldsymbol{D}^\pi \boldsymbol{R}_i^\pi, \quad 1 \leq i \leq N, \tag{3}$$

and $w := \lambda_{\min}(\boldsymbol{\Phi}^\top \boldsymbol{D}^\pi \boldsymbol{\Phi})$. The bound on the reward will be denoted by some positive constant $R_{\max} \in \mathbb{R}$, i.e., $|r^i(s, \boldsymbol{a}, s')| \leq R_{\max}$, $1 \leq i \leq N, \forall s, \boldsymbol{a}, s' \in \mathcal{S} \times \Pi_{i=1}^N \mathcal{A}_i \times \mathcal{S}$.

## 3 ANALYSIS OF PRIMAL-DUAL GRADIENT DYNAMICS

The so-called primal-dual gradient dynamics (Arrow et al., 1958) will be the key tool to derive finite-time bounds of the proposed distributed TD-learning. The analysis provided in this section will serve as the foundation for the subsequent analysis in Section 4. This section establishes exponential convergent behavior of the primal-dual gradient dynamics in terms of the Lyapunov method. To this end, let us consider the following constrained optimization problem:

$$\min_{\boldsymbol{\theta} \in \mathbb{R}^n} \quad f(\boldsymbol{\theta}) \quad \text{such that} \quad \boldsymbol{M}\boldsymbol{\theta} = \mathbf{0}_n, \tag{4}$$

where $\boldsymbol{\theta} \in \mathbb{R}^n$, $\boldsymbol{M} \in \mathbb{R}^{n \times n}$ and $f : \mathbb{R}^n \to \mathbb{R}$ is a differentiable, smooth, and strongly convex function (Boyd and Vandenberghe, 2004). One of the popular approaches for solving (4)

is to formulate it into the saddle-point problem (Boyd and Vandenberghe, 2004), $L(\boldsymbol{\theta}, \boldsymbol{w}) = \min_{\boldsymbol{\theta} \in \mathbb{R}^n} \max_{\boldsymbol{w} \in \mathbb{R}^n} (f(\boldsymbol{\theta}) + \boldsymbol{w}^\top \boldsymbol{M} \boldsymbol{\theta})$, whose solution, $\boldsymbol{\theta}^*, \boldsymbol{w}^* \in \mathbb{R}^n$, exists and is unique when $\boldsymbol{M}$ has full-column rank (Qu and Li, 2018). When $\boldsymbol{M}$ is rank-deficient, i.e., it is not full-column rank, there exists multiple $\boldsymbol{w}^*$ solving the saddle-point problem (Ozaslan and Jovanović, 2023). Indeed, whether $\boldsymbol{M}$ is rank-deficient or not, $\boldsymbol{\theta}^*$ can be shown to be the optimal solution of (4) by Karush-Kuhn-Tucker condition (Boyd and Vandenberghe, 2004). It is known that its solution $\boldsymbol{\theta}^*, \boldsymbol{w}^*$ can be obtained by investigating the solution $\boldsymbol{\theta}_t, \boldsymbol{w}_t \in \mathbb{R}^n$ of the so-called primal-dual gradient dynamics (Qu and Li, 2018), with initial points $\boldsymbol{\theta}_0, \boldsymbol{w}_0 \in \mathbb{R}^n$,

$$\dot{\boldsymbol{\theta}}_t = -\nabla f(\boldsymbol{\theta}_t) - \boldsymbol{M}^\top \boldsymbol{w}_t, \quad \dot{\boldsymbol{w}}_t = \boldsymbol{M} \boldsymbol{\theta}_t.$$

Qu and Li (2018) studied exponential stability of the primal-dual gradient dynamics when $\boldsymbol{M}$ is full column-rank, using the classical Lyapunov approach (Sontag, 2013). As for the case when $\boldsymbol{M}$ is rank-deficient, Ozaslan and Jovanović (2023); Cisneros-Velarde et al. (2020); Gokhale et al. (2023) proved exponential convergence to a particular solution $\boldsymbol{\theta}^*, \boldsymbol{w}^*$ using the tools based on singular value decomposition (Horn and Johnson, 2012). In this paper, we will study the behavior of the system under the following particular scenarios:

1. $\nabla f(\boldsymbol{\theta}_t) = \boldsymbol{U} \boldsymbol{\theta}_t$, where $\boldsymbol{U} \in \mathbb{R}^{n \times n}$, which is not necessarily symmetric, is a positive definite matrix, i.e., $\boldsymbol{U} + \boldsymbol{U}^\top \succ 0$;

2. $\boldsymbol{M}$ is symmetric and rank-deficient. Distributed algorithms are typical examples satisfying such condition and will be elaborated in subsequent sections.

We note that previous works considered general matrix $\boldsymbol{M}$, not necessarily a symmetric matrix. Moreover, note that the primal-dual gradient dynamics under such scenarios will appear in the further sections as an O.D.E. model of the proposed distributed TD-learning. The corresponding system can be rewritten as

$$\frac{d}{dt} \begin{bmatrix} \boldsymbol{\theta}_t \\ \boldsymbol{w}_t \end{bmatrix} = \begin{bmatrix} -\boldsymbol{U} & -\boldsymbol{M}^\top \\ \boldsymbol{M} & \boldsymbol{0}_{n \times n} \end{bmatrix} \begin{bmatrix} \boldsymbol{\theta}_t \\ \boldsymbol{w}_t \end{bmatrix}, \quad \boldsymbol{\theta}_0, \boldsymbol{w}_0 \in \mathbb{R}^n. \tag{5}$$

To study its exponential stability, let us introduce the Lyapunov function candidate $V(\boldsymbol{\theta}, \boldsymbol{w}) = \begin{bmatrix} \boldsymbol{\theta} \\ \boldsymbol{M} \boldsymbol{M}^\dagger \boldsymbol{w} \end{bmatrix}^\top \boldsymbol{S} \begin{bmatrix} \boldsymbol{\theta} \\ \boldsymbol{M} \boldsymbol{M}^\dagger \boldsymbol{w} \end{bmatrix}$, where $\boldsymbol{S} \in \mathbb{R}^{2n \times 2n}$ is some symmetric positive definite matrix, and $\boldsymbol{\theta}, \boldsymbol{w} \in \mathbb{R}^n$. The candidate Lyapunov function considers projection of the iterate $\boldsymbol{w}_t$ to the range space of $\boldsymbol{M}$. As in Ozaslan and Jovanović (2023); Cisneros-Velarde et al. (2020), the difficulty coming from singularity of $\boldsymbol{M}$ can be avoided by considering the range space and null space conditions of $\boldsymbol{M}$. In particular, Ozaslan and Jovanović (2023) employed a Lyapunov function that involves the gradient of the Lagrangian function, and considered the projected iterate $\boldsymbol{M} \boldsymbol{M}^\dagger \boldsymbol{w}_t$, where $\boldsymbol{M} \boldsymbol{M}^\dagger$ is the projection matrix onto range space of $\boldsymbol{M}$. Moreover, Cisneros-Velarde et al. (2020) exploited a quadratic Lyapunov function in (Qu and Li, 2018) for the iterate $\boldsymbol{\theta}_t$ and $\boldsymbol{V} \boldsymbol{w}_t$, where $\boldsymbol{M} := \boldsymbol{T} \boldsymbol{\Sigma} \boldsymbol{V}^\top$, which is the singular value decomposition of $\boldsymbol{M}$. Gokhale et al. (2023) considered a positive semi-definite matrix $\boldsymbol{S}$ and used semi-contraction theory (De Pasquale et al., 2023) to prove exponential convergence of the primal-dual gradient dynamics.

In this paper, we will adopt the quadratic Lyapunov function in (Qu and Li, 2018) with the projected iterate $\boldsymbol{M} \boldsymbol{M}^\dagger \boldsymbol{w}_t$, and leverage the symmetric property of $\boldsymbol{M}$ to show improved or comparable to the state of art convergence rate under the particular conditions newly imposed in this paper. In particular, when $\boldsymbol{M}$ is symmetric, the fact that the projection onto the column space of $\boldsymbol{M}$ and row space of $\boldsymbol{M}$ being identical simplifies the overall bounds. We first present the following Lyapunov inequality.

**Lemma 3.1.** *Let* $\boldsymbol{S} := \begin{bmatrix} \beta \boldsymbol{I}_n & \boldsymbol{M} \\ \boldsymbol{M} & \beta \boldsymbol{I}_n \end{bmatrix}$ *where* $\beta := \max \left\{ \frac{2\lambda_{\max}(\boldsymbol{M})^2 + 2 + \|\boldsymbol{U}\|_2^2}{\lambda_{\min}(\boldsymbol{U} + \boldsymbol{U}^\top)}, 4\lambda_{\max}(\boldsymbol{M}) \right\}$. *Then,* $\frac{\beta}{2} \boldsymbol{I}_{2n} \prec \boldsymbol{S} \prec 2\beta \boldsymbol{I}_{2n}$, *and we have, for any* $\boldsymbol{\theta}, \boldsymbol{w} \in \mathbb{R}^n$,

$$\begin{bmatrix} \boldsymbol{\theta} \\ \boldsymbol{M} \boldsymbol{M}^\dagger \boldsymbol{w} \end{bmatrix}^\top \boldsymbol{S} \begin{bmatrix} -\boldsymbol{U} & -\boldsymbol{M} \\ \boldsymbol{M} & \boldsymbol{0}_{n \times n} \end{bmatrix} \begin{bmatrix} \boldsymbol{\theta} \\ \boldsymbol{M} \boldsymbol{M}^\dagger \boldsymbol{w} \end{bmatrix} \leq -\min\{1, \lambda_{\min}^+(\boldsymbol{M})^2\} \left\| \begin{bmatrix} \boldsymbol{\theta} \\ \boldsymbol{M} \boldsymbol{M}^\dagger \boldsymbol{w} \end{bmatrix} \right\|_2^2.$$

The proof is given in Appendix Section A.4. Using the above Lemma 3.1, we can now prove the exponential stability of the O.D.E. dynamics in (5).

**Theorem 3.2.** *Let* $V(\boldsymbol{\theta}, \boldsymbol{w}) = \begin{bmatrix} \boldsymbol{\theta} \\ \boldsymbol{M}\boldsymbol{M}^{\dagger}\boldsymbol{w} \end{bmatrix}^{\top} \boldsymbol{S} \begin{bmatrix} \boldsymbol{\theta} \\ \boldsymbol{M}\boldsymbol{M}^{\dagger}\boldsymbol{w} \end{bmatrix}$. *For* $\boldsymbol{\theta}_0, \boldsymbol{w}_0 \in \mathbb{R}^n$ *and* $t \in \mathbb{R}^+$, *we have*

$$V(\boldsymbol{\theta}_t, \boldsymbol{w}_t) \leq \exp\left(-\frac{\min\{1, \lambda_{\min}^+(\boldsymbol{M})^2\}}{\max\left\{\frac{2\lambda_{\max}(\boldsymbol{M})^2 + 2 + \|\boldsymbol{U}\|_2^2}{\lambda_{\min}(\boldsymbol{U} + \boldsymbol{U}^{\top})}, 4\lambda_{\max}(\boldsymbol{M})\right\}} t\right) V(\boldsymbol{\theta}_0, \boldsymbol{w}_0).$$

The proof is given in Appendix Section A.5. We show that the above bound enjoys sharper or comparable to the state of the art convergence rate under particular conditions. With slight modifications, the Lyapunov function becomes identical to that of Gokhale et al. (2023). However, we directly rely on classical Lyapunov theory (Khalil, 2015) rather than the result from semi-contraction theory (De Pasquale et al., 2023) used in Gokhale et al. (2023).[1] The classical Lyapunov approach simplifies the proof steps compared to that of semi-contraction theory.The detailed comparative analysis is in Appendix Section A.6. The fact that $\boldsymbol{M}$ is symmetric and considering the projected iterate $\boldsymbol{M}\boldsymbol{M}^{\dagger}\boldsymbol{w}_t$, provides improved and comparable bound.

## 4 DISTRIBUTED TD-LEARNING WITH LINEAR FUNCTION APPROXIMATION

In this section, we propose a new distributed TD-learning algorithm to solve (1) based on the result in Wang and Elia (2011); Lee (2023). In this scenario, each agent keeps its own parameter estimate $\boldsymbol{\theta}^i \in \mathbb{R}^q$, $1 \leq i \leq N$, and the goal of each agent is to estimate the value function $\boldsymbol{V}^{\pi}(s) \approx \boldsymbol{\phi}(s)^{\top}\boldsymbol{\theta}_c$ satisfying (1) (i.e., the value function associated with the global reward $\sum_{k=0}^{\infty} \gamma^k \frac{1}{N} \sum_{i=1}^{N} r^i$) under the assumption that each agent has access only to its local reward $r^i$. The parameter of each agent can be shared over the communication network whose structure is represented by the graph $\mathcal{G}$, i.e., agents can share their parameters only with their neighbors over the network to solve the global problem. The connections among the agents can be represented by graph Laplacian matrix (Anderson Jr and Morley, 1985), $\boldsymbol{L} \in \mathbb{R}^{|\mathcal{S}| \times |\mathcal{S}|}$, which characterizes the graph $\mathcal{G}$, i.e., $[\boldsymbol{L}]_{ij} = -1$ if $(i, j) \in \mathcal{E}$ and $[\boldsymbol{L}]_{ij} = 0$ if $(i, j) \notin \mathcal{E}$, and $[\boldsymbol{L}]_{ii} = |\mathcal{N}_i|$ for $i \in \mathcal{V}$. Note that $\boldsymbol{L}$ is symmetric positive semi-definite matrix, i.e., $\boldsymbol{x}^{\top}\boldsymbol{L}\boldsymbol{x} \geq 0$ for $\boldsymbol{x} \in \mathbb{R}^{|\mathcal{S}|}$, and $\boldsymbol{L}\mathbf{1}_{|\mathcal{S}|} = 0$. To proceed, let us first introduce a set of matrix notations: $\bar{\boldsymbol{L}} := \boldsymbol{L} \otimes \boldsymbol{I}_q \in \mathbb{R}^{Nq \times Nq}$, $\bar{\boldsymbol{D}}^{\pi} := \boldsymbol{I}_N \otimes \boldsymbol{D}^{\pi} \in \mathbb{R}^{Nq \times Nq}$, $\bar{\boldsymbol{P}}^{\pi} := \boldsymbol{I}_N \otimes \boldsymbol{P}^{\pi} \in \mathbb{R}^{Nq \times Nq}$, $\bar{\boldsymbol{\Phi}} := \boldsymbol{I}_N \otimes \boldsymbol{\Phi} \in \mathbb{R}^{Nq \times Nq}$, $\bar{\boldsymbol{\theta}} = \begin{bmatrix} \boldsymbol{\theta}^{1\top} & \boldsymbol{\theta}^{2\top} & \cdots & \boldsymbol{\theta}^{N\top} \end{bmatrix}^{\top} \in \mathbb{R}^{Nq}$, $\bar{\boldsymbol{R}}^{\pi} = \begin{bmatrix} (\boldsymbol{R}_1^{\pi})^{\top} & (\boldsymbol{R}_2^{\pi})^{\top} & \cdots & (\boldsymbol{R}_N^{\pi})^{\top} \end{bmatrix}^{\top} \in \mathbb{R}^{Nq}$, $\bar{\boldsymbol{A}} = \boldsymbol{I}_N \otimes \boldsymbol{A} \in \mathbb{R}^{Nq \times Nq}$, $\bar{\boldsymbol{b}} = \begin{bmatrix} \boldsymbol{b}_1^{\top} & \boldsymbol{b}_2^{\top} & \cdots & \boldsymbol{b}_N^{\top} \end{bmatrix}^{\top} \in \mathbb{R}^{Nq}$, $\bar{\boldsymbol{w}} = \begin{bmatrix} \boldsymbol{w}^{1\top} & \boldsymbol{w}^{2\top} & \cdots & \boldsymbol{w}^{N\top} \end{bmatrix}^{\top} \in \mathbb{R}^{Nq}$, where $\otimes$ denotes Kronecker product, and $\bar{\boldsymbol{w}}$ is another collection of learnable parameters $\{\boldsymbol{w}^i \in \mathbb{R}^q\}_{i=1}^N$, where $\boldsymbol{w}^i$ assigned to each agent $i$ for $1 \leq i \leq N$.

Meanwhile, Wang and Elia (2011) studied distributed optimization algorithms (Tsitsiklis, 1984) from the control system perspectives in continuous-time domain, which can be represented as an Lagrangian problem (Hestenes, 1969). Compared to other distributed optimization algorithms (Nedic and Ozdaglar, 2009; Pu and Nedić, 2021), the method in Wang and Elia (2011) does not require any specific initialization, diminishing step-sizes, and doubly stochastic matrix that corresponds to the underlying communication graph. Due to these advantages, this framework has been further studied in Droge and Egerstedt (2014); Shi et al. (2015); Hatanaka et al. (2018); Bin et al. (2022). Inspired by Wang and Elia (2011), Lee (2023) developed a distributed TD-learning algorithm and provided an asymptotic convergence analysis based on the O.D.E. method. The analysis relies on Barbalat's lemma (Khalil, 2015), which makes extension to the non-asymptotic finite-time analysis difficult. Moreover, they mostly focus on the deterministic continuous-time algorithms. The corresponding distributed TD-learning is summarized in Algorithm 1, where each agent updates its local parameter using the local TD-error in (6). The updates in (7) and (8) in Algorithm 1 can be obtained by discretizing the continuous-time O.D.E. introduced in (Wang and Elia, 2011) and implementing it using stochastic samples.

---

[1] Gokhale et al. (2023) appeared on arxiv nearby the submission of this manuscript.

---

**Algorithm 1** Distributed TD-learning

Initialize $\alpha_0 \in (0,1)$, $\{\boldsymbol{\theta}_0^i, \boldsymbol{w}_0^i \in \mathbb{R}^q\}_{i=1}^N$, $\eta \in (0,\infty)$.
**for** $k = 1, 2, \ldots$ **do**
    **for** $i = 1, 2, \ldots, N$ **do**
        Agent $i$ observes $o_k^i := (s_k, s_k', r_k^i)$.
        Exchange $\{(\boldsymbol{\theta}_k^j, \boldsymbol{w}_k^j)\}$ with $j \in \mathcal{N}_i$ and update as follows:

$$\delta(o_k^i; \boldsymbol{\theta}_k^i) = r_k^i + \gamma \boldsymbol{\phi}^\top(s_k') \boldsymbol{\theta}_k^i - \boldsymbol{\phi}^\top(s_k) \boldsymbol{\theta}_k^i \tag{6}$$

$$\boldsymbol{\theta}_{k+1}^i = \boldsymbol{\theta}_k^i + \alpha_k(\delta(o_k^i; \boldsymbol{\theta}_k^i) \boldsymbol{\phi}(s_k) - \eta(|\mathcal{N}_i|\boldsymbol{\theta}_k^i - \textstyle\sum_{j \in \mathcal{N}_i} \boldsymbol{\theta}_k^j) - \eta(|\mathcal{N}_i|\boldsymbol{w}_k^i - \textstyle\sum_{j \in \mathcal{N}_i} \boldsymbol{w}_k^j)) \tag{7}$$

$$\boldsymbol{w}_{k+1}^i = \boldsymbol{w}_k^i + \alpha_k \eta(|\mathcal{N}_i|\boldsymbol{\theta}_k^i - \textstyle\sum_{j \in \mathcal{N}_i} \boldsymbol{\theta}_k^j) \tag{8}$$

    **end for**
**end for**

---

Using the stacked vector representation, $\begin{bmatrix} \bar{\boldsymbol{\theta}}_k \\ \bar{\boldsymbol{w}}_k \end{bmatrix} \in \mathbb{R}^{2Nq}$, $k \in \mathbb{N}_0$, the updates in (7) and (8) in Algorithm 1 can be rewritten in compact form, for $k \in \mathbb{N}_0$:

$$\begin{bmatrix} \bar{\boldsymbol{\theta}}_{k+1} \\ \bar{\boldsymbol{w}}_{k+1} \end{bmatrix} = \begin{bmatrix} \bar{\boldsymbol{\theta}}_k \\ \bar{\boldsymbol{w}}_k \end{bmatrix} + \alpha_k \begin{bmatrix} \bar{\boldsymbol{A}} - \eta \bar{\boldsymbol{L}} & -\eta \bar{\boldsymbol{L}} \\ \eta \bar{\boldsymbol{L}} & \boldsymbol{0}_{Nq \times Nq} \end{bmatrix} \begin{bmatrix} \bar{\boldsymbol{\theta}}_k \\ \bar{\boldsymbol{w}}_k \end{bmatrix} + \alpha_k \begin{bmatrix} \bar{\boldsymbol{b}} \\ \boldsymbol{0}_{Nq} \end{bmatrix} + \alpha_k \bar{\boldsymbol{\epsilon}}(o_k; \bar{\boldsymbol{\theta}}_k), \tag{9}$$

where, $o_k := \{o_k^i\}_{i=1}^N$, and for $1 \le i \le N$, $\boldsymbol{\epsilon}^i(o_k^i; \boldsymbol{\theta}_k^i) := \delta(o_k^i; \boldsymbol{\theta}_k^i) \boldsymbol{\phi}(s_k) - \boldsymbol{A} \boldsymbol{\theta}_k^i - \boldsymbol{b}^i \in \mathbb{R}^q$, and

$$\bar{\boldsymbol{\epsilon}}(o_k; \bar{\boldsymbol{\theta}}_k) := \begin{bmatrix} \boldsymbol{\epsilon}^1(o_k^1; \boldsymbol{\theta}_k^1)^\top & \boldsymbol{\epsilon}^2(o_k^2; \boldsymbol{\theta}_k^2)^\top & \cdots & \boldsymbol{\epsilon}^N(o_k^N; \boldsymbol{\theta}_k^N)^\top & \boldsymbol{0}_{Nq}^\top \end{bmatrix}^\top \in \mathbb{R}^{2Nq}. \tag{10}$$

Note that the superscript of $\bar{\boldsymbol{\epsilon}}^i$ corresponds to the superscript of $\boldsymbol{b}^i$. Compared to the algorithm in Lee (2023), we introduce an additional positive variable $\eta > 0$ multiplied with the graph Laplacian matrix, which results in the factor $\eta$ multiplied with the mixing part in Algorithm 1 in order to control the variance of the update. We note that when the the number of neighbors of an agent $i \in \mathcal{V}$ is large, then so is the variance of the corresponding updates of the agent. In this case, the variance can be controlled by adjusting $\eta$ to be small.

The behavior of stochastic algorithm is known to be closely related to its continuous-time O.D.E. counterpart (Borkar and Meyn, 2000; Srikant and Ying, 2019). In this respect, the corresponding O.D.E. model of (9) is given by

$$\frac{d}{dt} \begin{bmatrix} \bar{\boldsymbol{\theta}}_t \\ \bar{\boldsymbol{w}}_t \end{bmatrix} = \begin{bmatrix} \bar{\boldsymbol{A}} - \eta \bar{\boldsymbol{L}} & -\eta \bar{\boldsymbol{L}} \\ \eta \bar{\boldsymbol{L}} & \boldsymbol{0}_{Nq \times Nq} \end{bmatrix} \begin{bmatrix} \bar{\boldsymbol{\theta}}_t \\ \bar{\boldsymbol{w}}_t \end{bmatrix} + \begin{bmatrix} \bar{\boldsymbol{b}} \\ \boldsymbol{0}_{Nq} \end{bmatrix}, \quad \bar{\boldsymbol{\theta}}_0, \bar{\boldsymbol{w}}_0 \in \mathbb{R}^{Nq}, \tag{11}$$

for $t \in \mathbb{R}^+$. The above linear system is closely related to the primal-dual gradient dynamics in (5). Compared to (5), the difference lies in the fact that the above system corresponds to the the dynamics of the distributed TD-learning represented by matrix $\bar{\boldsymbol{A}}$ instead of the gradient of a particular objective function. It is straightforward to check that the equilibrium point of the above system is $\boldsymbol{1}_N \otimes \boldsymbol{\theta}_c$ and $\frac{1}{\eta} \bar{\boldsymbol{w}}_\infty$ such that $\bar{\boldsymbol{L}} \bar{\boldsymbol{w}}_\infty = \bar{\boldsymbol{A}}(\boldsymbol{1}_N \otimes \boldsymbol{\theta}_c) + \bar{\boldsymbol{b}}$.

In what follows, we will analyze finite-time behavior of (9) based on the Lyapunov equation in Lemma 4.1. For the analysis, we will follow the spirit of Srikant and Ying (2019), which provided a finite-time bound of the standard single-agent TD-learning based on the Lyapunov method (Sontag, 2013). To proceed further, let us consider the coordinate change of $\tilde{\boldsymbol{\theta}}_k := \bar{\boldsymbol{\theta}}_k - \boldsymbol{1}_N \otimes \boldsymbol{\theta}_c$ and $\tilde{\boldsymbol{w}}_k := \bar{\boldsymbol{w}}_k - \frac{1}{\eta} \bar{\boldsymbol{w}}_\infty$, with which we can rewrite (9) by

$$\begin{bmatrix} \tilde{\boldsymbol{\theta}}_{k+1} \\ \tilde{\boldsymbol{w}}_{k+1} \end{bmatrix} = \begin{bmatrix} \tilde{\boldsymbol{\theta}}_k \\ \tilde{\boldsymbol{w}}_k \end{bmatrix} + \alpha_k \begin{bmatrix} \bar{\boldsymbol{A}} - \eta \bar{\boldsymbol{L}} & -\eta \bar{\boldsymbol{L}} \\ \eta \bar{\boldsymbol{L}} & \boldsymbol{0}_{Nq \times Nq} \end{bmatrix} \begin{bmatrix} \tilde{\boldsymbol{\theta}}_k \\ \tilde{\boldsymbol{w}}_k \end{bmatrix} + \alpha_k \bar{\boldsymbol{\epsilon}}(o_k; \bar{\boldsymbol{\theta}}_k). \tag{12}$$

We will now derive a Lyapunov inequality (Sontag, 2013) for the above system based on the results in Lemma 4.1 for (11), To this end, we will rely on the analysis in Qu and Li (2018), which proved exponential convergence of the continuous-time primal-dual gradient dynamics (Arrow et al., 1958)

based on the Lyapunov method. However, the newly introduced singularity of $\bar{L}$ imposes difficulty in directly applying the results from Qu and Li (2018) which does not allow the singularity. To overcome this difficulty, we will multiply $\bar{L}\bar{L}^\dagger$ to the dual update $\tilde{w}_{k+1}$ in (12), which is the projection to the range space of $\bar{L}$. Moreover, the symmetric assumption of $\bar{L}$ helps construct an explicit solution of the Lyapunov inequality in Lemma 4.1. In particular, multiplying $\begin{bmatrix} I_N & 0_{Nq \times Nq} \\ 0_{Nq \times Nq} & \bar{L}\bar{L}^\dagger \end{bmatrix}$ to (12) leads to

$$
\begin{bmatrix} \tilde{\boldsymbol{\theta}}_{k+1} \\ \bar{L}\bar{L}^\dagger \tilde{\boldsymbol{w}}_{k+1} \end{bmatrix} = \begin{bmatrix} \tilde{\boldsymbol{\theta}}_k \\ \bar{L}\bar{L}^\dagger \tilde{\boldsymbol{w}}_k \end{bmatrix} + \alpha_k \begin{bmatrix} \bar{A} - \eta\bar{L} & -\eta\bar{L} \\ \eta\bar{L} & 0_{Nq \times Nq} \end{bmatrix} \begin{bmatrix} \tilde{\boldsymbol{\theta}}_k \\ \bar{L}\bar{L}^\dagger \tilde{\boldsymbol{w}}_k \end{bmatrix} + \alpha_k \bar{\boldsymbol{\epsilon}}_k(o_k; \bar{\boldsymbol{\theta}}_k), \qquad (13)
$$

which can be proved using Lemma A.2 in the Appendix Section A.3. For this modified system, we now derive the following Lyapunov inequality.

**Lemma 4.1.** *There exists positive symmetric definite matrix* $\boldsymbol{G} \in \mathbb{R}^{2Nq \times 2Nq}$ *such that* $\frac{8+\eta+4\eta^2\lambda_{\max}(\bar{L})^2}{2\eta(1-\gamma)w} I_{2Nq} \prec \boldsymbol{G} \prec 2\frac{8+\eta+4\eta^2\lambda_{\max}(\bar{L})^2}{\eta(1-\gamma)w} I_{2Nq}$, *and for* $\tilde{\boldsymbol{\theta}}, \tilde{\boldsymbol{w}} \in \mathbb{R}^{Nq}$,

$$
2 \begin{bmatrix} \tilde{\boldsymbol{\theta}} \\ \bar{L}\bar{L}^\dagger \tilde{\boldsymbol{w}} \end{bmatrix}^\top \boldsymbol{G} \begin{bmatrix} \bar{A} - \eta\bar{L} & -\eta\bar{L} \\ \eta\bar{L} & 0_{Nq \times Nq,} \end{bmatrix} \begin{bmatrix} \tilde{\boldsymbol{\theta}} \\ \bar{L}\bar{L}^\dagger \tilde{\boldsymbol{w}} \end{bmatrix} \le -\min\{1, \eta\lambda_{\min}^+(\bar{L})^2\} \left\| \begin{bmatrix} \tilde{\boldsymbol{\theta}} \\ \bar{L}\bar{L}^\dagger \tilde{\boldsymbol{w}} \end{bmatrix} \right\|_2^2.
$$

The proof is given in Appendix Section A.7. The proof can be done by noting that $\bar{A} - \eta\bar{L}$ is negative semi-definite and $\bar{L}$ is rank-deficient, and applying Lemma 3.1.

### 4.1 I.I.D. OBSERVATION CASE

We are now in position to provide the first main result, a finite-time analysis of Algorithm 1 under the i.i.d. observation model. We note that the i.i.d. observation model is common in the literature, and provides simple and clean theoretical insights.

**Theorem 4.2.** *1. Suppose we use constant step-size* $\alpha_0 = \alpha_1 = \cdots = \alpha_k$ *for* $k \in \mathbb{N}_0$, *and* $\alpha_0 \le \bar{\alpha}$ *such that* $\bar{\alpha} = \mathcal{O}\left( \frac{\min\{1, \eta\lambda_{\min}^+(\bar{L})^2\}}{\lambda_{\max}(\bar{L})^2\left(\frac{8}{\eta} + 4\eta\lambda_{\max}(\bar{L})^2\right)}(1-\gamma)w \right)$. *Then, we have*

$$
\frac{1}{N}\mathbb{E}\left[ \left\| \begin{bmatrix} \tilde{\boldsymbol{\theta}}_{k+1} \\ \bar{L}\bar{L}^\dagger \tilde{\boldsymbol{w}}_{k+1} \end{bmatrix} \right\|_2^2 \right] = \mathcal{O}\left( \exp(-\alpha_0 k) + \alpha_0 \frac{R_{\max}^2}{w^3(1-\gamma)^3} \frac{2 + \eta^2\lambda_{\max}(\bar{L})^2}{\eta\min\{1, \eta\lambda_{\min}(\bar{L})^2\}} \right).
$$

*2. Suppose we have* $\alpha_k = \frac{h_1}{k+h_2}$ *for* $k \in \mathbb{N}_0$. *There exist* $\bar{h}_1$ *and* $\bar{h}_2$ *such that letting* $h_1 = \Theta(\bar{h}_1)$ *and* $h_2 = \Theta(\bar{h}_2)$ *yields*

$$
\frac{1}{N}\mathbb{E}\left[ \left\| \begin{bmatrix} \tilde{\boldsymbol{\theta}}_{k+1} \\ \bar{L}\bar{L}^\dagger \tilde{\boldsymbol{w}}_{k+1} \end{bmatrix} \right\|_2^2 \right] = \mathcal{O}\left( \frac{1}{k} \frac{(2 + \eta^2\lambda_{\max}(\bar{L})^2)^2}{\eta^2\min\{1, \eta\lambda_{\min}^+(\bar{L})^2\}^2} \frac{R_{\max}^2}{w^4(1-\gamma)^4} \right).
$$

The proof and the exact constants can be found in Appendix Section A.9. Using constant step-size, we can guarantee exponential convergence rate with small bias term $\mathcal{O}\left( \alpha_0 \frac{R_{\max}^2 \lambda_{\max}(\bar{L})}{w^3(1-\gamma)^3} \right)$ when $\eta \approx \frac{1}{\lambda_{\max}(\bar{L})}$ and $\lambda_{\min}^+(\bar{L})^2 \ge \lambda_{\max}(\bar{L})$. Appropriate choice of $\eta$ allows wider range of step-size, and this will be clear in the experimental results in Section 5. Furthermore, the algorithm's performance is closely tied to the properties of the graph structure. $\lambda_{\min}^+(\bar{L})$, the smallest non-zero eigenvalue of graph Laplacian, characterizes the connectivity of the graph Chung (1997), and a graph with lower connectivity will yield larger bias. $\lambda_{\max}(\bar{L})$ is the largest eigenvalue of the graph Laplacian, and it can be upper bounded by twice the maximum degree of the graph (Anderson Jr and Morley, 1985). That is, a graph with higher maximum degree could incur larger bias. As for diminishing step-size, we achieve $\mathcal{O}\left(\frac{1}{k}\right)$ convergence rate from the second item in Theorem 4.2, and similar observations hold as in the constant step-size, i.e., the convergence rate depends on the smallest non-zero and maximum eigenavalue of graph Laplacian. Lastly, as in Wang et al. (2020), our bound does not explicitly depend on the number of agents, $N$, compared to the bound in Doan et al. (2019) and Sun et al. (2020), where the bias term and convergence rate scale at the order of $N$.

### 4.2 MARKOVIAN OBSERVATION CASE

In this section, we consider the Markovian observation model, where the sequence of observations $\{(s_k, s_k', r_k)\}_{k=1}^T$ follows a Markov chain. Compared to the i.i.d. observation model, the correlation between the observation and the updated iterates imposes difficulty in the analysis. To overcome this issue, an assumption on the Markov chain that ensures a geometric mixing property is helpful. In particular, the so-called ergodic Markov chain can be characterized by the metric called total variation distance (Levin and Peres, 2017), $d_{\mathrm{TV}}(P, Q) = \frac{1}{2}\sum_{x \in \mathcal{S}} |P(x) - Q(x)|$, where $P$ and $Q$ is probability measure on $\mathcal{S}$. A Markov chain is said to be ergodic if it is irreducible and aperiodic (Levin and Peres, 2017). An ergodic Markov chain is known to converge to its unique stationary exponentially fast, i.e., for $k \in \mathbb{N}_0$, $\sup_{1 \le i \le |\mathcal{S}|} d_{\mathrm{TV}}(e_i^\top (P^\pi)^k, \mu_\infty) \le m\rho^k$, where $e_i \in \mathbb{R}^{|\mathcal{S}|}$ for $1 \le i \le N$ is the $|\mathcal{S}|$-dimensional vector whose $i$-th element is one and others are zero, $\mu_\infty \in \mathbb{R}^{|\mathcal{S}|}$ is the stationary distribution of the Markov chain induced by transition matrix $P^\pi$, $m \in \mathbb{R}$ is a positive constant, and $\rho \in (0, 1)$. The assumption on the geometric mixing property of the Markov chain is common in the literature (Srikant and Ying, 2019; Bhandari et al., 2018; Wang et al., 2020). The mixing time of Markov chain is an important quantity of a Markov chain, defined as

$$\tau(\delta) := \min\{k \in \mathbb{N} \mid m\rho^k \le \delta\}. \tag{14}$$

For simplicity, we will use $\tau := \tau(\alpha_T)$, where $T \in \mathbb{N}_0$ denotes the total number of iterations, and $\alpha_k, k \in \mathbb{N}_0$, is the step-size at $k$-th iteration. If we use the step-size $\alpha_k = \frac{1}{1+k}$, $k \in \mathbb{N}$, the mixing time $\tau$ only contributes to the logarithmic factor, $\log T$ in the finite-time bound (Bhandari et al., 2018). As in the proof of i.i.d. case, using the Lypaunov argument in Lemma 4.1, we can prove the finite-time bound on the mean-squared error, following the spirit of Srikant and Ying (2019). To simplify the proof, we will investigate the case $\eta = 1$.

**Theorem 4.3.** *1. Suppose we use constant step-size $\alpha_0 = \alpha_1 = \cdots = \alpha_T$ such that $\alpha_0 \le \bar{\alpha}$ where*

$$\bar{\alpha} = \mathcal{O}\left(\frac{\min\{1, \lambda_{\min}^+(\bar{L})^2\}(1-\gamma)w}{\tau \max\left\{\frac{\sqrt{Nq}R_{\max}}{w(1-\gamma)}, q\right\}\lambda_{\max}(L)^2}\right). \text{ Then, we have, for } \tau \le k \le T,$$

$$\frac{1}{N}\mathbb{E}\left[\left\|\begin{bmatrix}\tilde{\theta}_{k+1} \\ \bar{L}\bar{L}^\dagger\tilde{w}_{k+1}\end{bmatrix}\right\|_2^2\right] = \mathcal{O}\left(\exp\left(-\alpha_0(k-\tau)\right) + \alpha_0\tau\frac{R_{\max}^2}{w^3(1-\gamma)^3}\frac{\lambda_{\max}(L)^2}{\min\{1, \lambda_{\min}^+(L)^2\}}\right).$$

*2. Considering diminishing step-size, with $\alpha_t = \frac{h_1}{t+h_2}$ for $t \in \mathbb{N}_0$, there exits $\bar{h}_1$ and $\bar{h}_2$ such that for $h_1 = \Theta(\bar{h}_1)$ and $h_2 = \Theta(\bar{h}_2)$, we have for $\tau \le k \le T$,*

$$\frac{1}{N}\mathbb{E}\left[\left\|\begin{bmatrix}\tilde{\theta}_{k+1} \\ \bar{L}\bar{L}^\dagger\tilde{w}_{k+1}\end{bmatrix}\right\|_2^2\right] = \mathcal{O}\left(\frac{\tau}{k}\frac{qR_{\max}^2}{w^4(1-\gamma)^4}\frac{\lambda_{\max}(L)^5}{\min\{1, \lambda_{\min}^+(L)^2\}^2}\right).$$

The proof and the exact values can be found in Appendix Section A.11. For the constant step-size, we can see that the bias term scales at the order of $\mathcal{O}\left(\tau\alpha_0\lambda_{\max}(L)^2\right)$, and the bounds have additional mixing time factors compared to the i.i.d. case. Considering diminishing step-size, the convergence rate of $\mathcal{O}\left(\frac{\tau}{k}\right)$ can be verified, incorporating a multiplication by the mixing time $\tau$. As summarized in Table 1, the proposed distributed TD-learning does not require doubly stochastic matrix or any specific initializations. The algorithms requiring the doubly stochastic matrix, whose definition is given in Appendix A.2, face challenges when extending to directed graph and time-varying graph scenarios. However, our algorithm does not require major modifications. Moreover, the performance of the algorithm is sensitive to the choice of doubly stochastic matrix as can be seen in Appendix A.13.

| | Method | Observation model | Step-size | Requirement | Doubly stochastic matrix |
|---|---|---|---|---|---|
| Doan et al. (2019) | Nedic and Ozdaglar (2009) | i.i.d. | Constant/$\frac{1}{\sqrt{k+1}}$ | Projection | ✓ |
| Doan et al. (2021) | Nedic and Ozdaglar (2009) | Markovian | Constant/$\frac{h_1}{k+1}$ | ✗ | ✓ |
| Sun et al. (2020) | Nedic and Ozdaglar (2009) | i.i.d./Markovian | Constant | ✗ | ✓ |
| Zeng et al. (2022) | Nedic and Ozdaglar (2009) | i.i.d./Markovian | Constant | ✗ | ✓ |
| Wang et al. (2020) | Pu and Nedić (2021) | i.i.d./Markovian | Constant | Specific initialization | ✓ |
| Ours | Wang and Elia (2011) | i.i.d./Markovian | Constant/$\frac{h_1}{k+h_2}$ | ✗ | ✗ |

Table 1: Comparison with existing works.

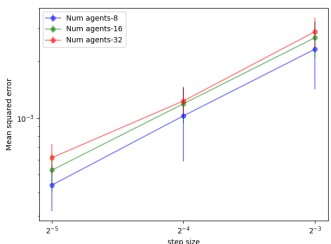 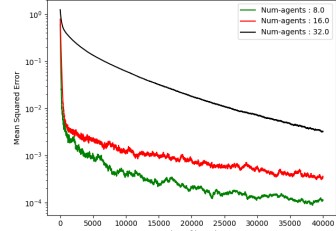 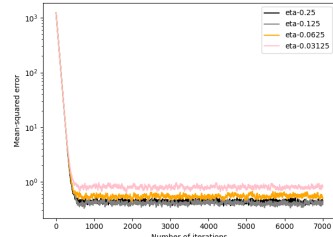

(a) The result shows mean-squared error of final iterate with $\eta = 1$ after 40,000 iterations. The experiment was averaged over 50 runs.

(b) The result shows mean-squared error for the step-size, $\alpha_k = \frac{N^2}{N^3+k}$, $k \in \mathbb{N}$, where $N$ stands for number of agents. The experiment was averaged over 50 runs.

(c) The result shows mean-squared error of the iterates of Algorithm 1. The experiments were averaged over 50 runs.

Figure 1: Experiment results of Algorithm 1.

## 5 EXPERIMENTS

This section provides the experimental results of Algorithm 1. To begin, we give an explanation of the MAMDP setup, where the number of states is three and the dimension of the feature is two. An agent can transit to every state with uniform probability. The feature matrix is set as $\mathbf{\Phi}^\top = \begin{bmatrix} 1 & 0 & 1 \\ 0 & 1 & 0 \end{bmatrix}$, which is a full-column rank matrix. The rewards are generated uniformly random between the interval $(0, 1)$. The discount factor is set as $0.8$.

For each experiment with $N \in \{2^3, 2^4, 2^5\}$ number of agents, we construct a cycle, i.e., a graph $\mathcal{G}$ consisting of $\mathcal{V} := \{1, 2, \ldots, N\}$ and $\mathcal{E} := \{(i, i+1)\}_{i=1}^{N-1} \cup \{(N, 1)\}$ (West, 2020). The smallest non-zero eigenvalue of graph Laplacian corresponding to a cycle with even number of vertices decreases as the number of vertices increases, while maximum eigenvalue remains same. The smallest non-zero eigenvalue is $2 - 2\cos\left(\frac{2\pi}{N}\right)$, and the largest eigenvalue is four (Mohar, 1997). As $N$ gets larger, the smallest non-zero eigenvalue gets smaller, which becomes $0.59, 0.15, 0.04$ for $N = 2^3, 2^4, 2^5$, respectively. Therefore, as number of agents increases, the bias in the final error term will be larger as expected in Theorem 4.2, and this can be verified in the plot in Figure (1a). The plot shows the result for constant step-size $\alpha_0 \in \{2^{-3}, 2^{-4}, 2^{-5}\}$. To investigate the effect of $\lambda_{\max}(\bar{\boldsymbol{L}})$, we construct a star graph, where one vertex has degree $N-1$ and the others have degree one. The maximum eigenvalue of star graph is $N$ and the smallest non-zero eigenvalue is one (Nica, 2016). As $N$ gets larger, we expect the bias term to be larger from the bound in Theorem 4.2. The result is further discussed in Appendix Section A.12.

To verify the effect of $\eta$, we construct a random graph model (Erdős et al., 1960), where among possible $N(N-1)/2$ edges, $(N-3)(N-4)/2$ edges are randomly selected. The plot in Figure (1c) shows the evolution of the mean squared error for $N = 32$, and step-size $0.1$ with different $\eta$ values. When $\eta = 0.5$ or $\eta = 1$, the algorithm diverges. Moreover, the bias gets smaller around $\frac{\sqrt{2}}{\lambda_{\max}(\boldsymbol{L})} \approx 0.046$. This implies that appropriate choice of $\eta$ can control the variance when the number of neighbors is large but if $\eta$ is too small or large, Algorithm 1 may cause divergence or large bias. This matches the result of the bound in Theorem 4.2. Further results can be found in Appendix Section A.12.

## 6 CONCLUSION

In this study, we have studied primal-dual gradient dynamics subject to some null-space constraints and its application to a distributed TD-learning. We have derived finite-time error bounds for both the gradient dynamics and the distributed TD-learning. The results have been experimentally demonstrated. Potential future studies include extending the study to finite-time bounds of distributed TD-learning with nonlinear function approximation.

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

# A APPENDIX

## A.1 NOTATIONS

$\mathbb{R}$: set of real numbers; $\mathbb{R}^+$: set of positive real numbers ; $\mathbb{N}$: set of natural numbers; $\mathbb{N}_0$: union of set of natural numbers and element zero; $\mathrm{diag}(\boldsymbol{A}_1, \boldsymbol{A}_2, \ldots, \boldsymbol{A}_n) \in \mathbb{R}^{m \times m}$ : block diagonal matrix constructed from $\boldsymbol{A}_1 \in \mathbb{R}^{d_1 \times d_1}, \boldsymbol{A}_2 \in \mathbb{R}^{d_2 \times d_2}, \ldots, \boldsymbol{A}_n \in \mathbb{R}^{d_n \times d_n}$ where $m = \sum_{i=1}^n d_i$; $\boldsymbol{1}_p \in \mathbb{R}^p$ : $p$-dimensional vector whose elements are all one; $\boldsymbol{0}_N \in \mathbb{R}^N$ : $N$-dimensional vector whose elements are all zero; $\boldsymbol{0}_{m \times n} \in \mathbb{R}^{m \times n}$ : $m \times n$-dimensional matrix whose elements are all zero; $\boldsymbol{I}_n \in \mathbb{R}^{n \times n}$: $n \times n$-dimensional identity matrix; $\boldsymbol{A}^\dagger \in \mathbb{R}^{n \times n}$: Moore-Penrose inverse of $\boldsymbol{A} \in \mathbb{R}^{n \times n}$; $\boldsymbol{A} \succeq \boldsymbol{B}$ for $\boldsymbol{A}, \boldsymbol{B} \in \mathbb{R}^{n \times n}$: $\boldsymbol{A} - \boldsymbol{B}$ is positive semi-definite matrix; $\|\boldsymbol{x}\|_{\boldsymbol{Q}}^2$ for positive-semi definite matrix $\boldsymbol{Q} \in \mathbb{R}^{n \times n}$ and $\boldsymbol{x} \in \mathbb{R}^n$: $\boldsymbol{x}^\top \boldsymbol{Q} \boldsymbol{x}$ ;$[\boldsymbol{v}]_i$, $1 \leq i \leq n$ for $\boldsymbol{v} \in \mathbb{R}^n$: $i$-th element of $\boldsymbol{v}$; $[\boldsymbol{A}]_{ij}$, $1 \leq i, j \leq n$ for $\boldsymbol{A} \in \mathbb{R}^{n \times n}$: $i$-th row and $j$-th column element of $\boldsymbol{A}$; $\lambda_{\max}(\boldsymbol{A})$ for $\boldsymbol{A} \in \mathbb{R}^{n \times n}$: maximum eigenvalue of $\boldsymbol{A}$; $\lambda_{\min}(\boldsymbol{A})$ for $\boldsymbol{A} \in \mathbb{R}^{n \times n}$: minimum eigenvalue of $\boldsymbol{A}$; $\lambda_{\min}^+(\boldsymbol{A})$ for $\boldsymbol{A} \in \mathbb{R}^{n \times n}$: minimum non-zero eigenvalue of $\boldsymbol{A}$; $\sigma(\mathcal{C})$: sigma algebra generated by a family of sets $\mathcal{C}$.

## A.2 DOUBLY STOCHASTIC MATRIX

**Definition A.1** (Doubly stochastic matrix (Doan et al., 2019))**.** *A doubly stochastic matrix $\boldsymbol{W} \in \mathbb{R}^{N \times N}$ is a stochastic matrix of which the row sum and column sum equal one, i.e., $\sum_{i=1}^N [\boldsymbol{W}]_{ji} = 1$ and $\sum_{i=1}^N [\boldsymbol{W}]_{ij} = 1$ for $1 \leq j \leq N$. A doubly stochastic corresponding to a graph $\mathcal{G} := (\mathcal{V}, \mathcal{E})$ requires additional assumption that $[\boldsymbol{W}]_{ii} > 0$ for $i \in \mathcal{V}$, and $[\boldsymbol{W}]_{uv} = 0$ for $(u, v) \notin \mathcal{E}$.*

One of the key advantage of our algorithm over other distributed TD algorithms is that we do not require doubly stochastic matrix corresponding to the graph network. We have outlined several reasons highlighting the importance of removing the requirement on doubly stochastic matrix:

To begin, constructing a doubly stochastic matrix in directed graph scenario is known to be more challenging than the undirected case, or may not be possible (Xin and Khan, 2018). However, our algorithm can be extended to the directed graph setting without major modifications.

Moreover, when dealing with a time-varying graph, whenever the graph changes, the doubly stochastic matrix needs to be constructed again. However, our analysis can be easily extended to the time-varying graph setting without any modifications.

Lastly, as from our experiment, the performance of distributed TD algorithms using doubly stochastic matrix is quite sensitive to the choice of doubly stochastic matrix, and the results can be found in Appendix A.13 in the revised version.

## A.3 TECHNICAL LEMMAS

**Lemma A.2** ( Pavlíková and Ševčovič (2023), p. 2)**.** *For real symmetric matrix $\boldsymbol{A} \in \mathbb{R}^n$, and its Moore-Penrose pseudo inverse $\boldsymbol{A}^\dagger$, the following holds:*

$$\boldsymbol{A}\boldsymbol{A}^\dagger = \boldsymbol{A}^\dagger \boldsymbol{A}, \quad \boldsymbol{A}\boldsymbol{A}^\dagger \boldsymbol{A} = \boldsymbol{A}.$$

**Lemma A.3** (Schur complement and symmetric positive definite matrices, Theorem 1.12 in Horn and Zhang (2005))**.** *Let $\boldsymbol{H} \in \mathbb{R}^{(n+m) \times (n+m)}$ be a symmetric matrix partitioned as*

$$\boldsymbol{H} := \begin{bmatrix} \boldsymbol{H}_{11} & \boldsymbol{H}_{12} \\ \boldsymbol{H}_{12}^\top & \boldsymbol{H}_{22} \end{bmatrix},$$

*where $\boldsymbol{H}_{11} \in \mathbb{R}^{n \times n}, \boldsymbol{H}_{12} \in \mathbb{R}^{n \times m}, \boldsymbol{H}_{22} \in \mathbb{R}^{m \times n}$. Then, the following holds:*

$$\boldsymbol{H} \succ 0 \iff \boldsymbol{H}_{11} \succ 0, \text{ and } \boldsymbol{H}_{22} - \boldsymbol{H}_{12}^\top \boldsymbol{H}_{11}^{-1} \boldsymbol{H}_{12} \succ 0.$$

**Lemma A.4** (Proposition 4.5 in Levin and Peres (2017))**.** *Let $\mu$ and $\nu$ be two probability distributions on $\mathcal{X}$. For $f : \mathcal{X} \to \mathbb{R}$, the total variation distance can be represented as*

$$d_{\mathrm{TV}}(\mu, \nu) := \frac{1}{2} \sup_{f: \sup_{x \in \mathcal{X}} |f(x)| \leq 1} \left| \sum_{x \in \mathcal{X}} f(x)\mu(x) - f(x)\nu(x) \right|.$$

**Lemma A.5.** *Consider the Markov chain in Section 4.2. Let $Y := (s_{k+\tau}, s_{k+\tau+1})$ for $k, \tau \in \mathbb{N}_0$, and $(s_{k+\tau}, s_{k+\tau+1}) \in \mathcal{S} \times \mathcal{S}$. For bounded function $f : \mathcal{S} \times \mathcal{S} \to \mathbb{R}$, i.e., $\sup_{x \in \mathcal{S} \times \mathcal{S}} |f(x)| < \infty$, we have*

$$|\mathbb{E}[f(Y) \mid s_k] - \mathbb{E}[f(Y)]| \leq 2 \sup_{x \in \mathcal{S} \times \mathcal{S}} |f(x)| m \rho^\tau.$$

*Moreover, for $\boldsymbol{v} : \mathcal{S} \times \mathcal{S} \to \mathbb{R}^{Nq}$, whose elements are bounded, we have*

$$\|\mathbb{E}[\boldsymbol{v}(Y) \mid s_k] - \mathbb{E}[\boldsymbol{v}(Y)]\|_2 \leq 2\sqrt{Nq} \sup_{x \in \mathcal{S} \times \mathcal{S}} \|\boldsymbol{v}(x)\|_\infty m \rho^\tau.$$

*For $\boldsymbol{M} : \mathcal{S} \times \mathcal{S} \to \mathbb{R}^{Nq \times Nq}$, whose elements are bounded, we have*

$$\|\mathbb{E}[\boldsymbol{M}(Y) \mid s_k] - \mathbb{E}[\boldsymbol{M}(Y)]\|_2 \leq 2Nq \sup_{x \in \mathcal{S} \times \mathcal{S}} \max_{1 \leq i,j \leq Nq} |[\boldsymbol{M}(x)]_{ij}| m \rho^\tau.$$

*Proof.* Let the probability measure $P(Y \in \cdot) = \mathbb{P}[Y \in \cdot \mid s_k]$ and $Q(Y \in \cdot) = \mathbb{P}[Y \in \cdot]$. For simplicity of the proof, let $f_\infty := 2 \sup_{x \in \mathcal{S} \times \mathcal{S}} |f(x)|$. Then, we have

$$|\mathbb{E}[f(Y) \mid s_k] - \mathbb{E}[f(Y)]|$$

$$= \left| \int f(Y) dP - \int f(Y) dQ \right|$$

$$= 2f_\infty \left| \int \frac{f}{2f_\infty} dP - \int \frac{f}{2f_\infty} dQ \right|$$

$$\leq 2f_\infty d_{\mathrm{TV}}(\mathbb{P}[Y \in \cdot \mid s_k], \mathbb{P}[Y \in \cdot])$$

$$= f_\infty \sum_{s,s' \in \mathcal{S} \times \mathcal{S}} |\mathbb{P}[s_{k+\tau} = s, s_{k+\tau+1} = s' \mid s_k] - \mathbb{P}[s_{k+\tau} = s, s_{k+\tau+1} = s']|$$

$$= f_\infty \sum_{s,s' \in \mathcal{S} \times \mathcal{S}} |\mathbb{P}[s_{k+\tau+1} = s' \mid s_k, s_{k+\tau} = s]\mathbb{P}[s_{k+\tau} = s \mid s_k] - \mathbb{P}[s_{k+\tau+1} = s' \mid s_{k+\tau} = s]\mathbb{P}[s_{k+\tau} = s]|$$

$$= f_\infty \sum_{s' \in \mathcal{S}} \sum_{s \in \mathcal{S}} |\mathbb{P}[s_{k+\tau+1} = s' \mid s_{k+\tau} = s]\mathbb{P}[s_{k+\tau} = s \mid s_k] - \mathbb{P}[s_{k+\tau+1} = s' \mid s_{k+\tau} = s]\mathbb{P}[s_{k+\tau} = s]|$$

$$\leq f_\infty \sum_{s' \in \mathcal{S}} \sum_{s \in \mathcal{S}} |\mathbb{P}[s_{k+\tau+1} = s' \mid s_{k+\tau} = s]||\mathbb{P}[s_{k+\tau} = s \mid s_k] - \mathbb{P}[s_{k+\tau} = s]|$$

$$= f_\infty \sum_{s \in \mathcal{S}} |\mathbb{P}[s_{k+\tau} = s \mid s_k] - \mathbb{P}[s_{k+\tau} = s]| \sum_{s' \in \mathcal{S}} |\mathbb{P}[s_{k+\tau+1} = s' \mid s_{k+\tau} = s]|$$

$$= 2f_\infty d_{\mathrm{TV}}(\mathbb{P}[s_{k+\tau} = s \mid s_k], \mathbb{P}[s_{k+\tau} = s]).$$

The first inequality follows from the definition of total variation distance in Lemma A.4. The last equality follows from the fact that $\sum_{s' \in \mathcal{S}} |\mathbb{P}[s_{k+\tau+1} = s' \mid s_{k+\tau} = s]| = 1$. We obtain the desired result from the ergodicity of the Markov chain.

For the second item, we have

$$\|\mathbb{E}[\boldsymbol{v}(Y) \mid s_k] - \mathbb{E}[\boldsymbol{v}(Y)]\|_2 = \sqrt{\sum_{i=1}^{Nq} (\mathbb{E}[\boldsymbol{v}_i(Y) \mid s_k] - \mathbb{E}[\boldsymbol{v}_i(Y)])^2},$$

where $\boldsymbol{v}_i$ denotes the $i$-th element of $\boldsymbol{v}$. The rest of the proof follows as in the proof of first item.

For the third item, we have

$$\|\mathbb{E}[\boldsymbol{M}(Y) \mid s_k] - \mathbb{E}[\boldsymbol{M}(Y)]\|_2 \leq \|\mathbb{E}[\boldsymbol{M}(Y) \mid s_k] - \mathbb{E}[\boldsymbol{M}(Y)]\|_F$$

$$= \sqrt{\sum_{i=1}^{Nq} \sum_{j=1}^{Nq} (\mathbb{E}[\boldsymbol{M}(Y)]_{ij} \mid s_k] - \mathbb{E}[\boldsymbol{M}(Y)]_{ij})^2},$$

where $\|\boldsymbol{B}\|_F = \sqrt{\sum_{i=1}^n \sum_{j=1}^n [\boldsymbol{B}]_{ij}^2}$ for $\boldsymbol{B} \in \mathbb{R}^{n \times n}$. The rest of the proof follows as in the proof of first item. $\square$

The following lemma provides similar bound as in Lemma 7 in Bhandari et al. (2018):

**Lemma A.6.** *Consider $\boldsymbol{\theta}_c$ in (2). We have*

$$\|\boldsymbol{\theta}_c\|_2 \leq \frac{R_{\max}}{(1-\gamma)w},$$

*where $w = \lambda_{\min}(\boldsymbol{\Phi}^\top \boldsymbol{D}^\pi \boldsymbol{\Phi})$.*

*Proof.* From (2), $\boldsymbol{\theta}_c$ satisfies

$$\boldsymbol{\Phi}^\top \boldsymbol{D}^\pi \boldsymbol{\Phi} \boldsymbol{\theta}_c - \gamma \boldsymbol{\Phi}^\top \boldsymbol{D}^\pi \boldsymbol{P}^\pi \boldsymbol{\Phi} \boldsymbol{\theta}_c = \frac{1}{N} \sum_{i=1}^N \boldsymbol{b}_i,$$

where $\boldsymbol{A}$ and $\boldsymbol{b}_i$ are defined in (3). Multiplying $\boldsymbol{\theta}_c$ on both sides of the equations, we have

$$\boldsymbol{\theta}_c^\top (\boldsymbol{\Phi}^\top \boldsymbol{D}^\pi \boldsymbol{\Phi} - \gamma \boldsymbol{\Phi}^\top \boldsymbol{D}^\pi \boldsymbol{P}^\pi \boldsymbol{\Phi}) \boldsymbol{\theta}_c = \boldsymbol{\theta}_c^\top \left( \frac{1}{N} \sum_{i=1}^N \boldsymbol{b}_i \right)$$

$$\leq \|\boldsymbol{\theta}_c\|_2 R_{\max},$$

where the inequality follows from Cauchy-Schwartz inequality. From Lemma A.7 in the Appendix Section A.3, we have

$$(-\boldsymbol{A} - \boldsymbol{A}^\top) \succeq 2(1-\gamma)\boldsymbol{\Phi}^\top \boldsymbol{D}^\pi \boldsymbol{\Phi},$$

which leads to

$$(1-\gamma)w \|\boldsymbol{\theta}_c\|_2^2 \leq \|\boldsymbol{\theta}_c\|_2 R_{\max}.$$

Therefore, we have

$$\|\boldsymbol{\theta}_c\|_2 \leq \frac{R_{\max}}{(1-\gamma)w}.$$

$\square$

The negative definiteness of $\boldsymbol{A}$ and upper bound on norm of $\boldsymbol{A}$ are established in the following lemma, which resembles that of Lemma 3 and 4 in Bhandari et al. (2018):

**Lemma A.7.** *We have*

$$\boldsymbol{A}^\top + \boldsymbol{A} \preceq 2(1-\gamma)\boldsymbol{\Phi}^\top \boldsymbol{D}^\pi \boldsymbol{\Phi}, \quad \|\boldsymbol{A}\|_2 \leq 2.$$

*Proof.* We will first prove the negative definiteness of $\boldsymbol{A}$. For any $\boldsymbol{v} \in \mathbb{R}^{|\mathcal{S}|}$, we have

$$\|\boldsymbol{P}^\pi \boldsymbol{v}\|_{\boldsymbol{D}^\pi} = \sqrt{\sum_{i=1}^{|\mathcal{S}|} d(i) \left( \sum_{j=1}^{|\mathcal{S}|} \mathcal{P}^\pi(i,j)[\boldsymbol{v}]_j \right)^2}$$

$$\leq \sqrt{\sum_{i=1}^{|\mathcal{S}|} d(i) \sum_{j=1}^{|\mathcal{S}|} \mathcal{P}^\pi(i,j)[\boldsymbol{v}]_j^2}$$

$$= \sqrt{\sum_{j=1}^{|\mathcal{S}|} [\boldsymbol{v}]_j^2 \sum_{i=1}^{|\mathcal{S}|} d(i)\mathcal{P}^\pi(i,j)}$$

$$= \sqrt{\sum_{j=1}^{|\mathcal{S}|} [\boldsymbol{v}]_j^2 d(j)}$$

$$= \|\boldsymbol{v}\|_{\boldsymbol{D}^\pi},$$

where the first inequality follow from Jensen's inequality and the second last equality follows from the fact that $d(s), s \in \mathcal{S}$ is the stationary distribution of Markov chain induced by $\mathcal{P}^\pi$. Therefore, we get

$$
\begin{aligned}
\boldsymbol{v}^\top \boldsymbol{A} \boldsymbol{v} =& \gamma \boldsymbol{v}^\top \boldsymbol{\Phi}^\top \boldsymbol{D}^\pi \boldsymbol{P}^\pi \boldsymbol{\Phi} \boldsymbol{v} - \boldsymbol{v}^\top \boldsymbol{\Phi}^\top \boldsymbol{D}^\pi \boldsymbol{\Phi} \boldsymbol{v} \\
\leq& \gamma \|\boldsymbol{\Phi} \boldsymbol{v}\|_{\boldsymbol{D}^\pi} \|\boldsymbol{P}^\pi \boldsymbol{\Phi} \boldsymbol{v}\|_{\boldsymbol{D}^\pi} - \boldsymbol{v}^\top \boldsymbol{\Phi}^\top \boldsymbol{D}^\pi \boldsymbol{\Phi} \boldsymbol{v} \\
\leq& \gamma \|\boldsymbol{\Phi} \boldsymbol{v}\|_{\boldsymbol{D}^\pi}^2 - \|\boldsymbol{\Phi} \boldsymbol{v}\|_{\boldsymbol{D}^\pi}^2 \\
=& (\gamma - 1) \boldsymbol{v}^\top \boldsymbol{\Phi}^\top \boldsymbol{D}^\pi \boldsymbol{\Phi} \boldsymbol{v}
\end{aligned}
$$

Now, we will prove the upper bound on $\|\boldsymbol{A}\|_2$. First, note that the following holds:

$$
\begin{aligned}
\left\|\boldsymbol{\Phi}^\top \boldsymbol{D}^\pi \boldsymbol{\Phi}\right\|_2 =& \left\| \sum_{i=1}^{|\mathcal{S}|} d(i) \boldsymbol{\phi}(i) \boldsymbol{\phi}(i)^\top \right\|_2 \\
\leq& \sum_{i=1}^{|\mathcal{S}|} d(i) \|\boldsymbol{\phi}(i)\|_2^2 \\
\leq& \sum_{i=1}^{|\mathcal{S}|} d(i) \\
=& 1,
\end{aligned}
$$

where the first inequality follows from triangle inequality, and the second inequality follows from the assumption that $\|\boldsymbol{\phi}(s)\|_2 \leq 1$ for $s \in \mathcal{S}$. Now, we have

$$
\begin{aligned}
\|\boldsymbol{A}\|_2 =& \left\| \sum_{s \in \mathcal{S}} d(s) \boldsymbol{\phi}(s) \left( -\boldsymbol{\phi}(s)^\top + \gamma \sum_{s' \in \mathcal{S}} \mathcal{P}^\pi(s, s') \boldsymbol{\phi}(s')^\top \right) \right\|_2 \\
\leq& \left\| \sum_{s \in \mathcal{S}} d(s) \boldsymbol{\phi}(s) \boldsymbol{\phi}(s)^\top \right\|_2 + \gamma \left\| \sum_{s \in \mathcal{S}} d(s) \sum_{s' \in \mathcal{S}} \mathcal{P}^\pi(s, s') \boldsymbol{\phi}(s) \boldsymbol{\phi}(s')^\top \right\|_2 \\
\leq& \sum_{s \in \mathcal{S}} d(s) + \gamma \sum_{s \in \mathcal{S}} d(s) \sum_{s' \mathcal{S}} \mathcal{P}^\pi(s, s') \\
\leq& 2.
\end{aligned}
$$

The first inequality follows from triangle inequality. Then second inequality follows from the assumption that $\|\boldsymbol{\phi}(s)\|_2 \leq 1$ for $s \in \mathcal{S}$. $\qquad\square$

**Lemma A.8.** *For $1 \leq i \leq N$, consider $\boldsymbol{b}_i$ in (3). We have*

$$
\|\boldsymbol{b}_i\|_2 \leq R_{\max}.
$$

*Proof.* For $1 \leq i \leq N$, we have

$$
\begin{aligned}
\|\boldsymbol{b}_i\|_2 =& \left\| \sum_{s \in \mathcal{S}} \boldsymbol{\phi}(s) d^\pi(s) [\boldsymbol{R}_i^\pi]_s \right\|_2 \\
\leq& \sum_{s \in \mathcal{S}} d^\pi(s) R_{\max} \\
=& R_{\max},
\end{aligned}
$$

where the first inequality follows from $\|\boldsymbol{\phi}(s)\|_2 \leq 1$ for $s \in \mathcal{S}$ and boundedness on the reward. $\qquad\square$

**Lemma A.9.** *We have*

$$
\left\| \begin{bmatrix} \bar{\boldsymbol{A}} - \bar{\boldsymbol{L}} & -\bar{\boldsymbol{L}} \\ \bar{\boldsymbol{L}} & \boldsymbol{0}_{Nq \times Nq} \end{bmatrix} \right\|_2 \leq 2 + 2\lambda_{\max}(\bar{\boldsymbol{L}}).
$$

*Proof.* Applying triangle inequality, we have

$$\left\| \begin{bmatrix} \bar{\boldsymbol{A}} - \bar{\boldsymbol{L}} & -\bar{\boldsymbol{L}} \\ \bar{\boldsymbol{L}} & \boldsymbol{0}_{Nq \times Nq} \end{bmatrix} \right\|_2 = \left\| \begin{bmatrix} \bar{\boldsymbol{A}} - \bar{\boldsymbol{L}} & \boldsymbol{0}_{Nq \times Nq} \\ \boldsymbol{0}_{Nq \times Nq} & \boldsymbol{0}_{Nq \times Nq} \end{bmatrix} + \begin{bmatrix} \boldsymbol{0}_{Nq \times Nq} & -\bar{\boldsymbol{L}} \\ \bar{\boldsymbol{L}} & \boldsymbol{0}_{Nq \times Nq} \end{bmatrix} \right\|_2$$

$$\leq \left\| \bar{\boldsymbol{A}} - \bar{\boldsymbol{L}} \right\|_2 + \left\| \bar{\boldsymbol{L}} \right\|_2$$

$$\leq 2 + 2\lambda_{\max}(\bar{\boldsymbol{L}}).$$

The last inequality follows again from triangle inequality and Lemma A.7.

$\square$

**Lemma A.10.** *For $k \in \mathbb{N}_0$, consider a sequence of observations $\{o_i\}_{i=1}^k$. Then, we have*

$$\left\| \bar{\boldsymbol{\epsilon}}(o_k; \bar{\boldsymbol{\theta}}_k) \right\|_2 \leq 6 \left\| \tilde{\boldsymbol{\theta}}_k \right\|_2 + \frac{9\sqrt{N} R_{\max}}{(1-\gamma)w}.$$

*In particular, if $\{o_i\}_{i=1}^k$ is sampled from i.i.d. distribution, we have*

$$\mathbb{E}\left[ \left\| \bar{\boldsymbol{\epsilon}}(o_k; \bar{\boldsymbol{\theta}}_k) \right\|_2^2 \right] \leq 16 \left\| \tilde{\boldsymbol{\theta}}_k \right\|^2 + \frac{32 N R_{\max}^2}{w^2(1-\gamma)^2}. \tag{15}$$

*Proof.* First, consider that for $1 \leq i \leq N$, we have

$$\left\| \boldsymbol{\epsilon}^i(o_k^i; \boldsymbol{\theta}_k^i) \right\|_2^2$$

$$= \left\| (r_k^i + \gamma \boldsymbol{\phi}^\top(s_k') \boldsymbol{\theta}_k^i - \boldsymbol{\phi}^\top(s_k) \boldsymbol{\theta}_k^i) \boldsymbol{\phi}(s_k) - \boldsymbol{A} \boldsymbol{\theta}_k^i - \boldsymbol{b}_i \right\|_2^2$$

$$\leq 2 \left\| (r_k^i + \gamma \boldsymbol{\phi}^\top(s_k') \boldsymbol{\theta}_k^i - \boldsymbol{\phi}^\top(s_k) \boldsymbol{\theta}_k^i) \boldsymbol{\phi}(s_k) \right\|_2^2 + 2 \left\| \boldsymbol{A} \boldsymbol{\theta}_k^i + \boldsymbol{b}_i \right\|_2^2$$

$$\leq 4 \left\| r_k^i \boldsymbol{\phi}(s_k) \right\|_2^2 + 4 \left\| (\gamma \boldsymbol{\phi}^\top(s_k') \boldsymbol{\theta}_k^i - \boldsymbol{\phi}^\top(s_k) \boldsymbol{\theta}_k^i) \boldsymbol{\phi}(s_k) \right\|_2^2 + 4\sigma_{\max}(\boldsymbol{A})^2 \left\| \boldsymbol{\theta}_k^i \right\|_2^2 + 4R_{\max}^2$$

$$\leq \left( 4\sigma_{\max}(\boldsymbol{A})^2 + 16 \right) \left\| \boldsymbol{\theta}_k^i \right\|_2^2 + 8R_{\max}^2, \tag{16}$$

where $\boldsymbol{\epsilon}^i(o_k^i; \boldsymbol{\theta}_k^i)$ is defined in (10). The second inequality follows from Lemma A.8. The last inequality follows from the assumption that $\|\boldsymbol{\phi}(s)\|_2 \leq 1$ for $s \in \mathcal{S}$ in Assumption 2.1, and $\|\boldsymbol{a} + \boldsymbol{b}\|_2^2 \leq 2\|\boldsymbol{a}\|_2^2 + 2\|\boldsymbol{b}\|_2^2$ for $\boldsymbol{a}, \boldsymbol{b} \in \mathbb{R}^{Nq}$.

Now, we have

$$\left\| \bar{\boldsymbol{\epsilon}}(o_k; \bar{\boldsymbol{\theta}}_k) \right\|_2 = \left\| \begin{bmatrix} \boldsymbol{\epsilon}^1(o_k^1; \boldsymbol{\theta}_k^1) \\ \boldsymbol{\epsilon}^2(o_k^2; \boldsymbol{\theta}_k^2) \\ \vdots \\ \boldsymbol{\epsilon}^N(o_k^N; \boldsymbol{\theta}_k^N) \\ \boldsymbol{0}_{Nq} \end{bmatrix} \right\|_2$$

$$= \sqrt{\sum_{i=1}^N \left\| \boldsymbol{\epsilon}^i(o_k^i; \boldsymbol{\theta}_k^i) \right\|_2^2}$$

$$\leq \sqrt{\sum_{i=1}^N \left( 4\sigma_{\max}(\boldsymbol{A})^2 + 16 \right) \left\| \boldsymbol{\theta}_k^i \right\|_2^2 + 8R_{\max}^2}$$

$$\leq \sqrt{\left( 4\sigma_{\max}(\boldsymbol{A})^2 + 16 \right)} \sqrt{\sum_{i=1}^N \left\| \boldsymbol{\theta}_k^i \right\|_2^2} + \sqrt{8N R_{\max}^2}$$

$$\leq 6 \left\| \bar{\boldsymbol{\theta}}_k \right\|_2 + 3\sqrt{N} R_{\max} \tag{17}$$

$$\leq 6 \left\| \tilde{\boldsymbol{\theta}}_k \right\|_2 + 6 \left\| \boldsymbol{1}_N \otimes \boldsymbol{\theta}_c \right\|_2 + 3\sqrt{N} R_{\max}$$

$$\leq 6 \left\| \tilde{\boldsymbol{\theta}}_k \right\|_2 + 6\sqrt{N} \frac{R_{\max}}{(1-\gamma)w} + 3\sqrt{N} R_{\max}$$

$$\leq 6 \left\| \tilde{\boldsymbol{\theta}}_k \right\|_2 + \frac{9\sqrt{N} R_{\max}}{(1-\gamma)w}.$$

The second equality follows from the definition of Euclidean norm. The first inequality follows from (16). The third inequality follows from bound on $\sigma_{\max}(\boldsymbol{A})$ in Lemma A.7. The fourth inequality follows from triangle inequality. The second last inequality follows from Lemma A.6.

We will now prove the inequality (15). For simplicity of the proof, let

$$
\bar{\boldsymbol{\delta}}(o_k; \bar{\boldsymbol{\theta}}_k) := \begin{bmatrix} \delta(o_k^1; \boldsymbol{\theta}_k^1)\boldsymbol{\phi}(s_k) \\ \delta(o_k^2; \boldsymbol{\theta}_k^2)\boldsymbol{\phi}(s_k) \\ \vdots \\ \delta(o_k^N; \boldsymbol{\theta}_k^N)\boldsymbol{\phi}(s_k) \end{bmatrix} \in \mathbb{R}^{Nq},
$$

where $\delta(o_k^i; \boldsymbol{\theta}_k^i)$, $1 \le i \le N$ is defined in (6). Since $\mathbb{E}\left[\bar{\boldsymbol{\delta}}(o_k; \bar{\boldsymbol{\theta}}_k) \middle| \mathcal{F}_{k-1}\right] = \bar{\boldsymbol{A}}\bar{\boldsymbol{\theta}}_k + \bar{\boldsymbol{b}}$, we have

$$
\begin{aligned}
&\mathbb{E}\left[\left\|\bar{\boldsymbol{\epsilon}}(o_k; \bar{\boldsymbol{\theta}}_k)\right\|_2^2 \middle| \mathcal{F}_{k-1}\right] \\
=&\mathbb{E}\left[\left\|\begin{bmatrix} \bar{\boldsymbol{\delta}}(o_k; \bar{\boldsymbol{\theta}}_k) \\ \mathbf{0}_{Nq} \end{bmatrix} - \begin{bmatrix} \bar{\boldsymbol{A}}\bar{\boldsymbol{\theta}}_k + \bar{\boldsymbol{b}} \\ \mathbf{0}_{Nq} \end{bmatrix}\right\|_2^2 \middle| \mathcal{F}_{k-1}\right] \\
=&\mathbb{E}\left[\left\|\bar{\boldsymbol{\delta}}(o_k; \bar{\boldsymbol{\theta}}_k)\right\|_2^2 \middle| \mathcal{F}_{k-1}\right] - 2\mathbb{E}\left[\begin{bmatrix} \bar{\boldsymbol{\delta}}(o_k; \bar{\boldsymbol{\theta}}_k) \\ \mathbf{0}_{Nq} \end{bmatrix}^\top \middle| \mathcal{F}_{k-1}\right]\begin{bmatrix} \bar{\boldsymbol{A}}\bar{\boldsymbol{\theta}}_k + \bar{\boldsymbol{b}} \\ \mathbf{0}_{Nq} \end{bmatrix} + \mathbb{E}\left[\left\|\begin{bmatrix} \bar{\boldsymbol{A}}\bar{\boldsymbol{\theta}}_k + \bar{\boldsymbol{b}} \\ \mathbf{0}_{Nq} \end{bmatrix}\right\|_2^2\right] \\
=&\mathbb{E}\left[\left\|\bar{\boldsymbol{\delta}}(o_k; \bar{\boldsymbol{\theta}}_k)\right\|_2^2 \middle| \mathcal{F}_{k-1}\right] - \mathbb{E}\left[\left\|\begin{bmatrix} \bar{\boldsymbol{A}}\bar{\boldsymbol{\theta}}_k + \bar{\boldsymbol{b}} \\ \mathbf{0}_{Nq} \end{bmatrix}\right\|_2^2\right] \\
\le&\mathbb{E}\left[\left\|\bar{\boldsymbol{\delta}}(o_k; \bar{\boldsymbol{\theta}}_k)\right\|_2^2 \middle| \mathcal{F}_{k-1}\right].
\end{aligned}
$$

Taking total expectation, we get

$$
\begin{aligned}
\mathbb{E}\left[\left\|\bar{\boldsymbol{\epsilon}}(o_k; \bar{\boldsymbol{\theta}}_k)\right\|_2^2\right] \le&\mathbb{E}\left[\left\|\bar{\boldsymbol{\delta}}(o_k; \bar{\boldsymbol{\theta}}_k)\right\|_2^2\right] \\
=&\mathbb{E}\left[\sum_{i=1}^N \left\|\delta(o_k; \boldsymbol{\theta}_k^i)\boldsymbol{\phi}(s_k)\right\|_2^2\right] \\
=&\mathbb{E}\left[\sum_{i=1}^N \left\|(r_k^i + \gamma\boldsymbol{\phi}^\top(s_k')\boldsymbol{\theta}_k^i - \boldsymbol{\phi}^\top(s_k)\boldsymbol{\theta}_k^i)\boldsymbol{\phi}(s_k)\right\|_2^2\right] \\
\le&\mathbb{E}\left[\sum_{i=1}^N \left(2\left\|r_k^i\boldsymbol{\phi}(s_k)\right\|_2^2 + 2\left\|\gamma\boldsymbol{\phi}(s_k)\boldsymbol{\phi}^\top(s_k) - \boldsymbol{\phi}(s_k)\boldsymbol{\phi}(s_k)^\top\right\|_2^2 \left\|\boldsymbol{\theta}_k^i\right\|_2^2\right)\right] \\
\le&\mathbb{E}\left[2\sum_{i=1}^N \left(R_{\max}^2 + 4\left\|\boldsymbol{\theta}_k^i\right\|_2^2\right)\right] \\
=&2NR_{\max}^2 + 8\left\|\bar{\boldsymbol{\theta}}_k\right\|_2^2.
\end{aligned}
$$

The second last inequality follows from the fact that $\|\boldsymbol{a} + \boldsymbol{b}\|_2^2 \le 2\|\boldsymbol{a}\|_2^2 + 2\|\boldsymbol{b}\|_2^2$ for $\boldsymbol{a}, \boldsymbol{b} \in \mathbb{R}^{Nq}$. The last inequality follows from the assumption that $\|\boldsymbol{\phi}(s)\|_2 \le 1$ for $s \in \mathcal{S}$ in Assumption 2.1. Using triangle inequality, we get

$$\mathbb{E}\left[\left\|\bar{\boldsymbol{\epsilon}}(o_k; \bar{\boldsymbol{\theta}}_k)\right\|_2^2\right] \leq 2NR_{\max}^2 + 8\left\|\bar{\boldsymbol{\theta}}_k - \mathbf{1}_N \otimes \boldsymbol{\theta}_c + \mathbf{1}_N \otimes \boldsymbol{\theta}_c\right\|_2^2$$

$$\leq 2NR_{\max}^2 + 16\left\|\mathbf{1}_N \otimes \boldsymbol{\theta}_c\right\|_2^2 + 16\left\|\tilde{\boldsymbol{\theta}}_k\right\|^2$$

$$\leq 2NR_{\max}^2 + 16N\left\|\boldsymbol{\theta}_c\right\|_2^2 + 16\left\|\tilde{\boldsymbol{\theta}}_k\right\|^2$$

$$\leq 2NR_{\max}^2 + 16N\left(\frac{R_{\max}}{w(1-\gamma)}\right)^2 + 16\left\|\tilde{\boldsymbol{\theta}}_k\right\|^2$$

$$= \frac{32NR_{\max}^2}{w^2(1-\gamma)^2} + 16\left\|\tilde{\boldsymbol{\theta}}_k\right\|^2.$$

The second inequality follows from the fact that $\|\boldsymbol{a} + \boldsymbol{b}\|_2^2 \leq 2\|\boldsymbol{a}\|_2^2 + 2\|\boldsymbol{b}\|_2^2$ for $\boldsymbol{a}, \boldsymbol{b} \in \mathbb{R}^{Nq}$. The last inequality follows from Lemma A.6.

□

### A.4 PROOF OF LEMMA 3.1

We will consider the following positive definite matrix:

$$\boldsymbol{S} = \begin{bmatrix} \beta\boldsymbol{I}_n & \boldsymbol{M} \\ \boldsymbol{M} & \beta\boldsymbol{I}_n \end{bmatrix} \in \mathbb{R}^{2n \times 2n}, \tag{18}$$

where the choice of positive constant $\beta \in \mathbb{R}$ in the statement of Lemma 3.1 will be deferred. Using the Schur complement in Lemma A.3 in the Appendix Section A.3, we can see that if $\beta > 2\lambda_{\max}(\boldsymbol{M})$, the following holds:

$$\begin{bmatrix} \frac{\beta}{2}\boldsymbol{I}_n & \boldsymbol{0}_{n \times n} \\ \boldsymbol{0}_{n \times n} & \frac{\beta}{2}\boldsymbol{I}_n \end{bmatrix} \prec \boldsymbol{S} \prec \begin{bmatrix} 2\beta\boldsymbol{I}_n & \boldsymbol{0}_{n \times n} \\ \boldsymbol{0}_{n \times n} & 2\beta\boldsymbol{I}_n \end{bmatrix}.$$

Now, we have the following relation:

$$2\begin{bmatrix} \boldsymbol{\theta} \\ \boldsymbol{M}\boldsymbol{M}^\dagger\boldsymbol{w} \end{bmatrix}^\top \boldsymbol{S} \begin{bmatrix} -\boldsymbol{U} & -\boldsymbol{M} \\ \boldsymbol{M} & \boldsymbol{0}_{n \times n} \end{bmatrix} \begin{bmatrix} \boldsymbol{\theta} \\ \boldsymbol{M}\boldsymbol{M}^\dagger\boldsymbol{w} \end{bmatrix}$$

$$= \begin{bmatrix} \boldsymbol{\theta} \\ \boldsymbol{M}\boldsymbol{M}^\dagger\boldsymbol{w} \end{bmatrix}^\top \begin{bmatrix} \beta\boldsymbol{I}_n & \boldsymbol{M} \\ \boldsymbol{M} & \beta\boldsymbol{I}_n \end{bmatrix} \begin{bmatrix} -\boldsymbol{U} & -\boldsymbol{M} \\ \boldsymbol{M} & \boldsymbol{0}_{n \times n} \end{bmatrix} \begin{bmatrix} \boldsymbol{\theta} \\ \boldsymbol{M}\boldsymbol{M}^\dagger\boldsymbol{w} \end{bmatrix} + \begin{bmatrix} \boldsymbol{\theta} \\ \boldsymbol{M}\boldsymbol{M}^\dagger\boldsymbol{w} \end{bmatrix}^\top \begin{bmatrix} -\boldsymbol{U}^\top & \boldsymbol{M} \\ -\boldsymbol{M} & \boldsymbol{0}_{n \times n} \end{bmatrix} \begin{bmatrix} \beta\boldsymbol{I}_n & \boldsymbol{M} \\ \boldsymbol{M} & \beta\boldsymbol{I}_n \end{bmatrix} \begin{bmatrix} \boldsymbol{\theta} \\ \boldsymbol{M}\boldsymbol{M}^\dagger\boldsymbol{w} \end{bmatrix}$$

$$= \begin{bmatrix} \boldsymbol{\theta} \\ \boldsymbol{M}\boldsymbol{M}^\dagger\boldsymbol{w} \end{bmatrix}^\top \begin{bmatrix} -\beta\boldsymbol{U} + \boldsymbol{M}^2 & -\beta\boldsymbol{M} \\ -\boldsymbol{M}\boldsymbol{U} + \beta\boldsymbol{M} & -\boldsymbol{M}^2 \end{bmatrix} \begin{bmatrix} \boldsymbol{\theta} \\ \boldsymbol{M}\boldsymbol{M}^\dagger\boldsymbol{w} \end{bmatrix} + \begin{bmatrix} \boldsymbol{\theta}_t \\ \boldsymbol{M}\boldsymbol{M}^\dagger\boldsymbol{w} \end{bmatrix}^\top \begin{bmatrix} -\beta\boldsymbol{U}^\top + \boldsymbol{M}^2 & -\boldsymbol{U}^\top\boldsymbol{M} + \beta\boldsymbol{M} \\ -\beta\boldsymbol{M} & -\boldsymbol{M}^2 \end{bmatrix} \begin{bmatrix} \boldsymbol{\theta} \\ \boldsymbol{M}\boldsymbol{M}^\dagger\boldsymbol{w} \end{bmatrix}$$

$$= \begin{bmatrix} \boldsymbol{\theta} \\ \boldsymbol{M}\boldsymbol{M}^\dagger\boldsymbol{w} \end{bmatrix}^\top \begin{bmatrix} -\beta(\boldsymbol{U} + \boldsymbol{U}^\top) + 2\boldsymbol{M}^2 & -\boldsymbol{U}^\top\boldsymbol{M} \\ -\boldsymbol{M}\boldsymbol{U} & -2\boldsymbol{M}^2 \end{bmatrix} \begin{bmatrix} \boldsymbol{\theta} \\ \boldsymbol{M}\boldsymbol{M}^\dagger\boldsymbol{w} \end{bmatrix},$$

where the first equality follows from plugging in $\boldsymbol{S}$ in (18). Expanding the terms, we get

$$2\begin{bmatrix} \boldsymbol{\theta} \\ \boldsymbol{M}\boldsymbol{M}^\dagger\boldsymbol{w} \end{bmatrix}^\top \boldsymbol{S} \begin{bmatrix} -\boldsymbol{U} & -\boldsymbol{M} \\ \boldsymbol{M} & \boldsymbol{0}_{n \times n} \end{bmatrix} \begin{bmatrix} \boldsymbol{\theta} \\ \boldsymbol{M}\boldsymbol{M}^\dagger\boldsymbol{w} \end{bmatrix}$$

$$= \begin{bmatrix} \boldsymbol{\theta} \\ \boldsymbol{M}\boldsymbol{M}^\dagger\boldsymbol{w} \end{bmatrix}^\top \begin{bmatrix} -\beta(\boldsymbol{U} + \boldsymbol{U}^\top) + 2\boldsymbol{M}^2 & -\boldsymbol{U}^\top\boldsymbol{M} \\ -\boldsymbol{M}\boldsymbol{U} & -2\boldsymbol{M}^2 \end{bmatrix} \begin{bmatrix} \boldsymbol{\theta} \\ \boldsymbol{M}\boldsymbol{M}^\dagger\boldsymbol{w} \end{bmatrix}$$

$$= \boldsymbol{\theta}^\top(-\beta(\boldsymbol{U} + \boldsymbol{U}^\top) + 2\boldsymbol{M}^2)\boldsymbol{\theta} - \boldsymbol{w}^\top\boldsymbol{M}\boldsymbol{U}\boldsymbol{\theta} - \boldsymbol{\theta}^\top\boldsymbol{U}^\top\boldsymbol{M}\boldsymbol{w} - 2\boldsymbol{w}^\top\boldsymbol{M}^2\boldsymbol{w}$$

$$= \begin{bmatrix} \boldsymbol{\theta} \\ \boldsymbol{M}\boldsymbol{w} \end{bmatrix}^\top \begin{bmatrix} -\beta(\boldsymbol{U} + \boldsymbol{U}^\top) + 2\boldsymbol{M}^2 & -\boldsymbol{U}^\top \\ -\boldsymbol{U} & -2\boldsymbol{I}_n \end{bmatrix} \begin{bmatrix} \boldsymbol{\theta} \\ \boldsymbol{M}\boldsymbol{w} \end{bmatrix},$$

where the second last equality follows from the axiom of Moore-Penrose pseudo inverse of symmetric matrices in Lemma A.2 in the Appendix Section A.3, i.e., $\boldsymbol{MM^\dagger M} = \boldsymbol{MMM^\dagger} = \boldsymbol{M^\dagger MM} = \boldsymbol{M}$.

Now, it is enough to choose $\beta > 0$ that satisfies following relation:

$$\begin{bmatrix} -\beta(\boldsymbol{U} + \boldsymbol{U}^\top) + 2\boldsymbol{M}^2 & -\boldsymbol{U}^\top \\ -\boldsymbol{U} & -2\boldsymbol{I}_n \end{bmatrix} \prec - \begin{bmatrix} \boldsymbol{I}_n & \boldsymbol{0}_{n \times n} \\ \boldsymbol{0}_{n \times n} & \boldsymbol{I}_n \end{bmatrix}$$

$$\iff \begin{bmatrix} -\beta(\boldsymbol{U} + \boldsymbol{U}^\top) + 2\boldsymbol{M}^2 + \boldsymbol{I}_n & -\boldsymbol{U}^\top \\ -\boldsymbol{U} & -\boldsymbol{I}_n \end{bmatrix} \prec \boldsymbol{0}_{2n \times 2n}.$$

The above relation can be shown using Schur's complement Lemma A.2 in the Appendix Section A.3,

$$-\beta(\boldsymbol{U} + \boldsymbol{U}^\top) + 2\boldsymbol{M}^2 + \boldsymbol{I}_n + \boldsymbol{UU}^\top \prec 0,$$

which holds when $\beta$ satisfies

$$\beta\lambda_{\min}(\boldsymbol{U} + \boldsymbol{U}^\top) > 2\lambda_{\max}(\boldsymbol{M})^2 + 1 + \|\boldsymbol{U}\|_2^2$$

$$\iff \beta > \frac{2\lambda_{\max}(\boldsymbol{M})^2 + 1 + \|\boldsymbol{U}\|_2^2}{\lambda_{\min}(\boldsymbol{U} + \boldsymbol{U}^\top)}.$$

Therefore, we get

$$\begin{bmatrix} \boldsymbol{\theta} \\ \boldsymbol{Mw} \end{bmatrix}^\top \begin{bmatrix} -\beta(\boldsymbol{U} + \boldsymbol{U}^\top) + 2\boldsymbol{M}^2 & \boldsymbol{U}^\top \\ \boldsymbol{U} & -2\boldsymbol{I} \end{bmatrix} \begin{bmatrix} \boldsymbol{\theta} \\ \boldsymbol{Mw} \end{bmatrix} \leq - \begin{bmatrix} \boldsymbol{\theta} \\ \boldsymbol{Mw} \end{bmatrix}^\top \begin{bmatrix} \boldsymbol{\theta} \\ \boldsymbol{Mw} \end{bmatrix}$$

$$\leq - \|\boldsymbol{\theta}\|_2^2 - \|\boldsymbol{Mw}\|_2^2$$

$$\leq - \min\{1, \lambda_{\min}^+(\boldsymbol{M})^2\} \left\| \begin{bmatrix} \boldsymbol{\theta} \\ \boldsymbol{MM^\dagger w} \end{bmatrix} \right\|_2^2,$$

where the last inequality follows from the inequality that $\|\boldsymbol{MM^\dagger w}\|_2 = \|\boldsymbol{M^\dagger Mw}\|_2 \leq \|\boldsymbol{M^\dagger}\|_2 \|\boldsymbol{Mw}\|_2 \leq \frac{1}{\lambda_{\min}^+(\boldsymbol{M})} \|\boldsymbol{Mw}\|_2$. Hence, it is sufficient to choose $\beta = \max\left\{ \frac{2\lambda_{\max}(\boldsymbol{M})^2 + 2 + \|\boldsymbol{U}\|_2^2}{\lambda_{\min}(\boldsymbol{U} + \boldsymbol{U}^\top)}, 4\lambda_{\max}(\boldsymbol{M}) \right\}$.

## A.5 PROOF OF THEOREM 3.2

*Proof.* Let us consider the quadratic Lyapunov function candidate $V(\boldsymbol{\theta}, \boldsymbol{w}) = \begin{bmatrix} \boldsymbol{\theta} \\ \boldsymbol{MM^\dagger w} \end{bmatrix}^\top \boldsymbol{S} \begin{bmatrix} \boldsymbol{\theta} \\ \boldsymbol{MM^\dagger w} \end{bmatrix}$ where $\boldsymbol{S} \in \mathbb{R}^{2n \times 2n}$ is symmetric positive definite matrix in Lemma 3.1. The time derivative of $V(\boldsymbol{\theta}_t, \boldsymbol{w}_t)$ along the solution of (5) becomes

$$\frac{d}{dt}V(\boldsymbol{\theta}_t, \boldsymbol{w}_t) = 2\left( \frac{d}{dt} \begin{bmatrix} \boldsymbol{\theta}_t \\ \boldsymbol{MM^\dagger w}_t \end{bmatrix} \right)^\top \boldsymbol{S} \begin{bmatrix} \boldsymbol{\theta} \\ \boldsymbol{MM^\dagger w} \end{bmatrix}$$

$$= 2 \begin{bmatrix} -\boldsymbol{U\theta}_t - \boldsymbol{Mw}_t \\ \boldsymbol{MM^\dagger M\theta}_t \end{bmatrix}^\top \boldsymbol{S} \begin{bmatrix} \boldsymbol{\theta}_t \\ \boldsymbol{MM^\dagger w}_t \end{bmatrix}$$

$$= 2 \begin{bmatrix} -\boldsymbol{U\theta}_t - \boldsymbol{MMM^\dagger w}_t \\ \boldsymbol{M\theta}_t \end{bmatrix}^\top \boldsymbol{S} \begin{bmatrix} \boldsymbol{\theta}_t \\ \boldsymbol{MM^\dagger w}_t \end{bmatrix}$$

$$= 2 \begin{bmatrix} \boldsymbol{\theta}_t \\ \boldsymbol{MM^\dagger w}_t \end{bmatrix}^\top \begin{bmatrix} -\boldsymbol{U} & -\boldsymbol{M} \\ \boldsymbol{M} & \boldsymbol{0}_{n \times n} \end{bmatrix}^\top \boldsymbol{S} \begin{bmatrix} \boldsymbol{\theta}_t \\ \boldsymbol{MM^\dagger w}_t \end{bmatrix}$$

$$\leq - 2\min\{1, \lambda_{\min}^+(\boldsymbol{M})^2\} \left\| \begin{bmatrix} \boldsymbol{\theta}_t \\ \boldsymbol{MM^\dagger w}_t \end{bmatrix} \right\|_2^2$$

$$\leq - 2\min\{1, \lambda_{\min}^+(\boldsymbol{M})^2\} \frac{1}{\lambda_{\max}(\boldsymbol{S})} V(\boldsymbol{\theta}_t, \boldsymbol{w}_t),$$

where the second last inequality comes from Lemma 3.1. The last inequality follows from the fact that $V(\boldsymbol{\theta}_t, \boldsymbol{w}_t) \leq \lambda_{\max}(\boldsymbol{S}) \left\| \begin{bmatrix} \boldsymbol{\theta}_t \\ \boldsymbol{M}\boldsymbol{M}^{\dagger}\boldsymbol{w}_t \end{bmatrix} \right\|_2^2$. From the Lyapunov method, this inequality results in

$$V(\boldsymbol{\theta}_t, \boldsymbol{w}_t) \leq \exp\left( -\frac{\min\{1, \lambda_{\min}^+(\boldsymbol{M})^2\}}{\max\left\{ \frac{2\lambda_{\max}(\boldsymbol{M})^2 + 2 + \|\boldsymbol{U}\|_2^2}{\lambda_{\min}(\boldsymbol{U} + \boldsymbol{U}^{\top})}, 4\lambda_{\max}(\boldsymbol{M}) \right\}} t \right) V(\boldsymbol{\theta}_0, \boldsymbol{w}_0).$$

This completes the proof. □

## A.6 Comparison with the result of Ozaslan and Jovanović (2023); Cisneros-Velarde et al. (2020); Gokhale et al. (2023)

We will consider $f(\boldsymbol{x}) = \frac{1}{2}\|\boldsymbol{x}\|_{\boldsymbol{B}}^2$ where $\boldsymbol{x} \in \mathbb{R}^n$ and $\boldsymbol{B} \in \mathbb{R}^{n \times n}$ is symmetric positive definite matrix. Then, $\nabla^2 f(\boldsymbol{x}) = \boldsymbol{B}$, and $f(\boldsymbol{x})$ is $\lambda_{\min}(\boldsymbol{B})$-strongly convex and $\lambda_{\max}(\boldsymbol{B})$-smooth. Theorem 8 in Gokhale et al. (2023) states exponential convergence rate of $\mathcal{O}\left( \exp\left( -\min\left\{ \frac{\lambda_{\min}^+(\boldsymbol{M})^2}{\lambda_{\max}(\boldsymbol{U})}, \frac{\lambda_{\min}^+(\boldsymbol{M})^2}{\lambda_{\max}(\boldsymbol{M})^2}\lambda_{\min}(\boldsymbol{U}) \right\} t \right) \right)$. When $\frac{\lambda_{\min}^+(\boldsymbol{M})^2}{\lambda_{\max}(\boldsymbol{M})^2}$ is the dominant term, the bound yields the convergence rate $\mathcal{O}\left( \exp\left( -\frac{\lambda_{\min}^+(\boldsymbol{M})^2}{\lambda_{\max}(\boldsymbol{M})^2} t \right) \right)$. Our bound in Theorem 3.2 also results to the convergence rate of $\mathcal{O}\left( \exp\left( -\frac{\lambda_{\min}^+(\boldsymbol{M})^2}{\lambda_{\max}(\boldsymbol{M})^2} t \right) \right)$ when $\lambda_{\min}^+(\boldsymbol{M})$ is small.

Letting $V(\boldsymbol{\theta}_t, \boldsymbol{w}_t) = \|\boldsymbol{\theta}_t\|_2^2 + \|\boldsymbol{M}\boldsymbol{M}^{\dagger}\boldsymbol{w}_t - \boldsymbol{w}^*\|_2^2$, the result of Theorem 2 in Ozaslan and Jovanović (2023) leads to

$$V(\boldsymbol{\theta}_t, \boldsymbol{w}_t)$$
$$\leq 2\exp\left( -\frac{2\lambda_{\min}(\boldsymbol{B})\min\{\lambda_{\min}(\boldsymbol{B})^2, \lambda_{\min}^+(\boldsymbol{M})^2\}}{(\lambda_{\max}(\boldsymbol{B})^2 + \lambda_{\max}(\boldsymbol{M})^2 + 1)(1 + 2\lambda_{\min}(\boldsymbol{B})\lambda_{\max}(\boldsymbol{B}))} t \right) \left( \|\nabla L(\boldsymbol{\theta}_0, \boldsymbol{w}_0)\|_2^2 + V(\boldsymbol{\theta}_0, \boldsymbol{w}_0) \right).$$

When $\lambda_{\min}(\boldsymbol{B}) \to 0$, the above convergence rate becomes $\mathcal{O}(\exp(-\lambda_{\min}(\boldsymbol{B})^3 t))$. Whereas, from Theorem 3.2, our result states $\mathcal{O}\left( \exp(-\lambda_{\min}(\boldsymbol{B})t) \right)$ convergence rate under the same condition, which implies tighter convergence rate.

Cisneros-Velarde et al. (2020) proved exponential convergence rate for $\begin{bmatrix} \boldsymbol{\theta}_t \\ \boldsymbol{R}\boldsymbol{w}_t \end{bmatrix}$, where $\boldsymbol{M} := \boldsymbol{R}\boldsymbol{\Sigma}\boldsymbol{R}^{\top}$ is the singular value decomposition of $\boldsymbol{M}$. Theorem 4 in Cisneros-Velarde et al. (2020) leads to the following convergence rate:

$$\mathcal{O}\left( \exp\left( -\frac{\lambda_{\min}(\boldsymbol{B})}{\lambda_{\max}(\boldsymbol{M})^2 + \frac{3}{4}\lambda_{\max}(\boldsymbol{M})\lambda_{\min}^+(\boldsymbol{M})^2 + \lambda_{\max}(\boldsymbol{B})^2} \frac{\lambda_{\max}(\boldsymbol{M})\lambda_{\min}^+(\boldsymbol{M})^2}{\lambda_{\max}(\boldsymbol{M}) + 1} t \right) \right).$$

When $\lambda_{\max}(\boldsymbol{M}) \approx \lambda_{\min}^+(\boldsymbol{M}) \to 0$, the bound implies
$$\mathcal{O}\left( \exp\left( -\lambda_{\max}(\boldsymbol{M})\lambda_{\min}^+(\boldsymbol{M})^2 t \right) \right),$$
where as our bound in Theorem 3.2 implies tighter convergence rate of
$$\mathcal{O}\left( \exp\left( -\lambda_{\min}^+(\boldsymbol{M})^2 t \right) \right).$$

The overall comparison with Ozaslan and Jovanović (2023); Cisneros-Velarde et al. (2020) is summarized in the Table 1.

## A.7 Proof of Lemma 4.1

We will consider the following positive definite matrix:

$$\boldsymbol{G} := \begin{bmatrix} \beta \boldsymbol{I}_{Nq} & \bar{\boldsymbol{L}} \\ \bar{\boldsymbol{L}} & \beta \boldsymbol{I}_{Nq} \end{bmatrix} \in \mathbb{R}^{2Nq \times 2Nq}, \tag{19}$$

| | Convergence rate | Condition |
|---|---|---|
| Ozaslan and Jovanović (2023) | $\mathcal{O}\left(\exp\left(-\lambda_{\min}(\boldsymbol{U})^3 t\right)\right)$ | $\lambda_{\min}(\boldsymbol{U}) \to 0$ |
| Ours | $\mathcal{O}\left(\exp\left(-\lambda_{\min}(\boldsymbol{U}) t\right)\right)$ | |
| Cisneros-Velarde et al. (2020) | $\mathcal{O}\left(\exp\left(-\lambda_{\min}^+(\boldsymbol{M})^3 t\right)\right)$ | $\lambda_{\max}(\boldsymbol{M}) \approx \lambda_{\min}^+(\boldsymbol{M}) \to 0$ |
| Ours | $\mathcal{O}\left(\exp\left(-\lambda_{\min}^+(\boldsymbol{M})^2 t\right)\right)$ | |

Table 2: $t \geq 0$ stands for time.

where the choice of positive constant $\beta \in \mathbb{R}$ will be deferred. Using the Schur complement in Lemma A.3 in the Appendix Section A.3, we can see that if $\beta > 2\lambda_{\max}(\bar{\boldsymbol{L}})$, the following holds:

$$\begin{bmatrix} \frac{\beta}{2}\boldsymbol{I} & \boldsymbol{0}_{Nq \times Nq} \\ \boldsymbol{0}_{Nq \times Nq} & \frac{\beta}{2}\boldsymbol{I} \end{bmatrix} \prec \boldsymbol{G} \prec \begin{bmatrix} 2\beta\boldsymbol{I} & \boldsymbol{0}_{Nq \times Nq} \\ \boldsymbol{0}_{Nq \times Nq} & 2\beta\boldsymbol{I} \end{bmatrix}.$$

Now, we have the following relation:

$$2\begin{bmatrix} \tilde{\boldsymbol{\theta}} \\ \bar{\boldsymbol{L}}\bar{\boldsymbol{L}}^\dagger \tilde{\boldsymbol{w}} \end{bmatrix}^\top \boldsymbol{G} \begin{bmatrix} \bar{\boldsymbol{A}} - \eta\bar{\boldsymbol{L}} & -\eta\bar{\boldsymbol{L}} \\ \eta\bar{\boldsymbol{L}} & \boldsymbol{0}_{Nq \times Nq} \end{bmatrix} \begin{bmatrix} \tilde{\boldsymbol{\theta}} \\ \bar{\boldsymbol{L}}\bar{\boldsymbol{L}}^\dagger \tilde{\boldsymbol{w}} \end{bmatrix}$$

$$= \begin{bmatrix} \tilde{\boldsymbol{\theta}} \\ \bar{\boldsymbol{L}}\bar{\boldsymbol{L}}^\dagger \tilde{\boldsymbol{w}} \end{bmatrix}^\top \boldsymbol{G} \begin{bmatrix} \bar{\boldsymbol{A}} - \eta\bar{\boldsymbol{L}} & -\eta\bar{\boldsymbol{L}} \\ \eta\bar{\boldsymbol{L}} & \boldsymbol{0}_{Nq \times Nq}, \end{bmatrix} \begin{bmatrix} \tilde{\boldsymbol{\theta}} \\ \bar{\boldsymbol{L}}\bar{\boldsymbol{L}}^\dagger \tilde{\boldsymbol{w}} \end{bmatrix} + \begin{bmatrix} \tilde{\boldsymbol{\theta}} \\ \bar{\boldsymbol{L}}\bar{\boldsymbol{L}}^\dagger \tilde{\boldsymbol{w}} \end{bmatrix}^\top \begin{bmatrix} \bar{\boldsymbol{A}}^\top - \eta\bar{\boldsymbol{L}} & \eta\bar{\boldsymbol{L}} \\ -\eta\bar{\boldsymbol{L}} & \boldsymbol{0}_{Nq \times Nq}, \end{bmatrix} \boldsymbol{G} \begin{bmatrix} \tilde{\boldsymbol{\theta}} \\ \bar{\boldsymbol{L}}\bar{\boldsymbol{L}}^\dagger \tilde{\boldsymbol{w}} \end{bmatrix}$$

$$= \begin{bmatrix} \tilde{\boldsymbol{\theta}} \\ \bar{\boldsymbol{L}}\bar{\boldsymbol{L}}^\dagger \tilde{\boldsymbol{w}} \end{bmatrix}^\top \begin{bmatrix} \beta(\bar{\boldsymbol{A}} + \bar{\boldsymbol{A}}^\top - 2\eta\bar{\boldsymbol{L}}) + 2\eta\bar{\boldsymbol{L}}^2 & (\bar{\boldsymbol{A}}^\top - \eta\bar{\boldsymbol{L}})\bar{\boldsymbol{L}} \\ \bar{\boldsymbol{L}}(\bar{\boldsymbol{A}} - \eta\bar{\boldsymbol{L}}) & -2\eta\bar{\boldsymbol{L}}^2 \end{bmatrix} \begin{bmatrix} \tilde{\boldsymbol{\theta}} \\ \bar{\boldsymbol{L}}\bar{\boldsymbol{L}}^\dagger \tilde{\boldsymbol{w}} \end{bmatrix},$$

where the last equality follows from plugging the choice of $\boldsymbol{G}$ in (19). Expanding the terms, we get

$$\begin{bmatrix} \tilde{\boldsymbol{\theta}} \\ \bar{\boldsymbol{L}}\bar{\boldsymbol{L}}^\dagger \tilde{\boldsymbol{w}} \end{bmatrix}^\top \begin{bmatrix} \beta(\bar{\boldsymbol{A}} + \bar{\boldsymbol{A}}^\top - 2\eta\bar{\boldsymbol{L}}) + 2\eta\bar{\boldsymbol{L}}^2 & (\bar{\boldsymbol{A}}^\top - \eta\bar{\boldsymbol{L}})\bar{\boldsymbol{L}} \\ \bar{\boldsymbol{L}}(\bar{\boldsymbol{A}} - \eta\bar{\boldsymbol{L}}) & -2\eta\bar{\boldsymbol{L}}^2 \end{bmatrix} \begin{bmatrix} \tilde{\boldsymbol{\theta}} \\ \bar{\boldsymbol{L}}\bar{\boldsymbol{L}}^\dagger \tilde{\boldsymbol{w}} \end{bmatrix}$$

$$= \tilde{\boldsymbol{\theta}}^\top (\beta(\bar{\boldsymbol{A}} + \bar{\boldsymbol{A}}^\top - 2\eta\bar{\boldsymbol{L}}) + 2\eta\bar{\boldsymbol{L}}^2)\tilde{\boldsymbol{\theta}} + \tilde{\boldsymbol{\theta}}^\top (\bar{\boldsymbol{A}}^\top - \eta\bar{\boldsymbol{L}})\bar{\boldsymbol{L}}\bar{\boldsymbol{L}}\bar{\boldsymbol{L}}^\dagger \tilde{\boldsymbol{w}}$$

$$\quad + \tilde{\boldsymbol{w}}^\top \bar{\boldsymbol{L}}^\dagger \bar{\boldsymbol{L}}\bar{\boldsymbol{L}}(\bar{\boldsymbol{A}} - \eta\bar{\boldsymbol{L}})\bar{\boldsymbol{\theta}} - 2\eta\tilde{\boldsymbol{w}}^\top \bar{\boldsymbol{L}}^\dagger \bar{\boldsymbol{L}}\bar{\boldsymbol{L}}^2 \bar{\boldsymbol{L}}\bar{\boldsymbol{L}}^\dagger \tilde{\boldsymbol{w}}$$

$$= \tilde{\boldsymbol{\theta}}^\top (\beta(\bar{\boldsymbol{A}} + \bar{\boldsymbol{A}}^\top - 2\eta\bar{\boldsymbol{L}}) + 2\eta\bar{\boldsymbol{L}}^2)\tilde{\boldsymbol{\theta}} + \tilde{\boldsymbol{\theta}}(\bar{\boldsymbol{A}}^\top - \eta\bar{\boldsymbol{L}})\bar{\boldsymbol{L}}\tilde{\boldsymbol{w}}$$

$$\quad + \tilde{\boldsymbol{w}}^\top \bar{\boldsymbol{L}}(\bar{\boldsymbol{A}} - \eta\bar{\boldsymbol{L}})\bar{\boldsymbol{\theta}} - 2\eta \left\| \bar{\boldsymbol{L}}\tilde{\boldsymbol{w}} \right\|_2^2$$

$$= \begin{bmatrix} \tilde{\boldsymbol{\theta}} \\ \bar{\boldsymbol{L}}\tilde{\boldsymbol{w}} \end{bmatrix}^\top \begin{bmatrix} \beta(\bar{\boldsymbol{A}} + \bar{\boldsymbol{A}}^\top - 2\eta\bar{\boldsymbol{L}}) + 2\eta\bar{\boldsymbol{L}}^2 & \bar{\boldsymbol{A}}^\top - \eta\bar{\boldsymbol{L}} \\ \bar{\boldsymbol{A}} - \eta\bar{\boldsymbol{L}} & -2\eta\boldsymbol{I} \end{bmatrix} \begin{bmatrix} \tilde{\boldsymbol{\theta}} \\ \bar{\boldsymbol{L}}\tilde{\boldsymbol{w}} \end{bmatrix}, \quad (20)$$

where the second last equality follows from the axiom of Moore-Penrose axiom of symmetric matrices in Lemma A.2 in the Appendix Section A.3, i.e., $\bar{\boldsymbol{L}}\bar{\boldsymbol{L}}^\dagger \bar{\boldsymbol{L}} = \bar{\boldsymbol{L}}\bar{\boldsymbol{L}}\bar{\boldsymbol{L}}^\dagger = \bar{\boldsymbol{L}}^\dagger \bar{\boldsymbol{L}}\bar{\boldsymbol{L}} = \bar{\boldsymbol{L}}$.

Now, it is enough to choose $c$ that satisfies following relation:

$$\begin{bmatrix} \beta(\bar{\boldsymbol{A}} + \bar{\boldsymbol{A}}^\top - 2\eta\bar{\boldsymbol{L}}) + 2\eta\bar{\boldsymbol{L}}^2 & \bar{\boldsymbol{A}}^\top - \eta\bar{\boldsymbol{L}} \\ \bar{\boldsymbol{A}} - \eta\bar{\boldsymbol{L}} & -2\eta\boldsymbol{I} \end{bmatrix} \preceq - \begin{bmatrix} \boldsymbol{I}_{Nq} & \boldsymbol{0}_{Nq \times Nq} \\ \boldsymbol{0}_{Nq \times Nq} & \eta\boldsymbol{I}_{Nq} \end{bmatrix} \quad (21)$$

$$\iff \begin{bmatrix} \beta(\bar{\boldsymbol{A}} + \bar{\boldsymbol{A}}^\top - 2\eta\bar{\boldsymbol{L}}) + 2\eta\bar{\boldsymbol{L}}^2 & \bar{\boldsymbol{A}}^\top - \eta\bar{\boldsymbol{L}} \\ \bar{\boldsymbol{A}} - \eta\bar{\boldsymbol{L}} & -2\eta\boldsymbol{I} \end{bmatrix} + \begin{bmatrix} \boldsymbol{I}_{Nq} & \boldsymbol{0}_{Nq \times Nq} \\ \boldsymbol{0}_{Nq \times Nq} & \eta\boldsymbol{I}_{Nq} \end{bmatrix} \preceq \boldsymbol{0}_{2Nq \times 2Nq}.$$

Using the result $\bar{\boldsymbol{A}} + \bar{\boldsymbol{A}}^\top \preceq 2(\gamma - 1)w$ from Lemma A.7, we have

$$\begin{bmatrix} \beta(\bar{\boldsymbol{A}} + \bar{\boldsymbol{A}}^\top - 2\eta\bar{\boldsymbol{L}}) + 2\eta\bar{\boldsymbol{L}}^2 & \bar{\boldsymbol{A}}^\top - \eta\bar{\boldsymbol{L}} \\ \bar{\boldsymbol{A}} - \eta\bar{\boldsymbol{L}} & -2\eta\boldsymbol{I} \end{bmatrix} + \begin{bmatrix} \boldsymbol{I}_{Nq} & \boldsymbol{0}_{Nq \times Nq} \\ \boldsymbol{0}_{Nq \times Nq} & \eta\boldsymbol{I}_{Nq} \end{bmatrix}$$

$$\preceq \begin{bmatrix} (2\beta(\gamma - 1)w + 1 + 2\eta\lambda_{\max}(\bar{\boldsymbol{L}})^2)\boldsymbol{I}_{Nq} & \bar{\boldsymbol{A}}^\top - \eta\bar{\boldsymbol{L}} \\ \bar{\boldsymbol{A}} - \eta\bar{\boldsymbol{L}} & -\eta\boldsymbol{I} \end{bmatrix}.$$

The inequality follows from the fact that $\bar{\boldsymbol{L}}^2$ is positive semi-definite matrix. To make the above matrix negative definite, according to the Schur complement argument in Lemma A.3, we need

$$(2\beta(\gamma - 1)w + 1 + 2\eta\lambda_{\max}(\bar{\boldsymbol{L}})^2)\boldsymbol{I}_{Nq} + \frac{1}{\eta}(\bar{\boldsymbol{A}} - \eta\bar{\boldsymbol{L}})(\bar{\boldsymbol{A}}^\top - \eta\bar{\boldsymbol{L}}) \prec 0, \tag{22}$$

which can be satisfied if the following holds for $c$:

$$(2\beta(\gamma - 1)w + 1 + 2\eta\lambda_{\max}(\bar{\boldsymbol{L}})^2) + \frac{1}{\eta}\left\|\bar{\boldsymbol{A}} - \eta\bar{\boldsymbol{L}}\right\|_2^2 < 0$$

$$\iff \frac{\frac{1}{\eta}\left\|\bar{\boldsymbol{A}} - \eta\bar{\boldsymbol{L}}\right\|_2^2 + 1 + 2\eta\lambda_{\max}(\bar{\boldsymbol{L}})^2}{2(1 - \gamma)w} < \beta.$$

Since $\left\|\bar{\boldsymbol{A}}\right\|_2^2 \leq 4$ from Lemma A.7, and $a^2 + b^2 \geq 2ab$ for $a, b \in \mathbb{R}$, it suffices to satisfy

$$\beta > \frac{8 + \eta + 4\eta^2\lambda_{\max}(\bar{\boldsymbol{L}})^2}{2\eta(1 - \gamma)w}.$$

Therefore, choosing

$$\beta = \frac{8 + \eta + 4\eta^2\lambda_{\max}(\bar{\boldsymbol{L}})^2}{\eta(1 - \gamma)w}$$

suffices to satisfy (22). Note that $\beta \geq \frac{1}{(1-\gamma)w} + \frac{8}{\eta(1-\gamma)w} + \frac{4\eta\lambda_{\max}(\bar{\boldsymbol{L}})^2}{(1-\gamma)w} > 4\lambda_{\max}(\bar{\boldsymbol{L}})\frac{1}{(1-\gamma)w} \geq 4\lambda_{\max}(\bar{\boldsymbol{L}})$. Applying the relation (21) to (20) yields the following result:

$$\begin{bmatrix} \tilde{\boldsymbol{\theta}} \\ \bar{\boldsymbol{L}}\tilde{\boldsymbol{w}} \end{bmatrix}^\top \begin{bmatrix} \beta(\bar{\boldsymbol{A}} + \bar{\boldsymbol{A}}^\top - 2\eta\bar{\boldsymbol{L}}) + 2\eta\bar{\boldsymbol{L}}^2 & \bar{\boldsymbol{A}}^\top - \eta\bar{\boldsymbol{L}} \\ \bar{\boldsymbol{A}} - \eta\bar{\boldsymbol{L}} & -2\eta\boldsymbol{I} \end{bmatrix} \begin{bmatrix} \tilde{\boldsymbol{\theta}} \\ \bar{\boldsymbol{L}}\tilde{\boldsymbol{w}} \end{bmatrix} \leq - \begin{bmatrix} \tilde{\boldsymbol{\theta}} \\ \bar{\boldsymbol{L}}\tilde{\boldsymbol{w}} \end{bmatrix}^\top \begin{bmatrix} \boldsymbol{I}_{Nq} & \boldsymbol{0}_{Nq \times Nq} \\ \boldsymbol{0}_{Nq \times Nq} & \eta\boldsymbol{I}_{Nq} \end{bmatrix} \begin{bmatrix} \tilde{\boldsymbol{\theta}} \\ \bar{\boldsymbol{L}}\tilde{\boldsymbol{w}} \end{bmatrix}$$

$$= -\left\|\tilde{\boldsymbol{\theta}}\right\|_2^2 - \eta\left\|\bar{\boldsymbol{L}}\tilde{\boldsymbol{w}}\right\|_2^2$$

$$\leq -\left\|\tilde{\boldsymbol{\theta}}\right\|_2^2 - \eta\lambda_{\min}^+(\bar{\boldsymbol{L}})^2\left\|\bar{\boldsymbol{L}}^\dagger\bar{\boldsymbol{L}}\tilde{\boldsymbol{w}}\right\|_2^2$$

$$= -\min\left\{1, \eta\lambda_{\min}^+(\bar{\boldsymbol{L}})^2\right\}\left\|\begin{bmatrix} \tilde{\boldsymbol{\theta}} \\ \bar{\boldsymbol{L}}^\dagger\bar{\boldsymbol{L}}\tilde{\boldsymbol{w}} \end{bmatrix}\right\|_2^2,$$

where the last inequality follows from the following relation:

$$\left\|\bar{\boldsymbol{L}}^\dagger\bar{\boldsymbol{L}}\tilde{\boldsymbol{w}}\right\|_2 \leq \left\|\bar{\boldsymbol{L}}^\dagger\right\|_2\left\|\bar{\boldsymbol{L}}\tilde{\boldsymbol{w}}\right\|_2 = \frac{1}{\lambda_{\min}^+(\bar{\boldsymbol{L}})}\left\|\bar{\boldsymbol{L}}\tilde{\boldsymbol{w}}\right\|_2.$$

## A.8 STOCHASTIC RECURSIVE UPDATE : I.I.D. OBSERVATION MODEL

In this section, we will consider the i.i.d. observation model of the sequence $\{o_k\}_{k \in \mathbb{N}_0}$ and $o_k \in \mathcal{S} \times \mathcal{S} \times \Pi_{i=1}^N I$ where $I$ is the closed interval $[-R_{\max}, R_{\max}]$ in $\mathbb{R}$. We consider the following general stochastic recursive update (Robbins and Monro, 1951), for $k \in \mathbb{N}_0$ and $\boldsymbol{z}_0 \in \mathbb{R}^{2Nq}$:

$$\boldsymbol{z}_{k+1} = \boldsymbol{z}_k + \alpha_k(\boldsymbol{E}\boldsymbol{z}_k + \boldsymbol{\xi}(o_k; \boldsymbol{z}_k)), \tag{23}$$

where $\boldsymbol{E} \in \mathbb{R}^{2Nq \times 2Nq}$, $\boldsymbol{\xi}(\cdot; \boldsymbol{z}) : \mathcal{S} \times \mathcal{S} \times \Pi_{i=1}^N I \to \mathbb{R}^{2Nq}$ is a function parameterized by $\boldsymbol{z} \in \mathbb{R}^{2Nq}$, and $\alpha_k \in (0, 1)$.

**Assumption A.11.** *1. For $k \in \mathbb{N}_0$, $\boldsymbol{\xi}(o_k; \boldsymbol{z}_k)$ has the following bound:*

$$\mathbb{E}\left[\left\|\boldsymbol{\xi}(o_k; \boldsymbol{z}_k)\right\|_2^2\right] \leq C_1\mathbb{E}\left[\left\|\boldsymbol{z}_k\right\|_2^2\right] + C_2$$

*2. For $k \in \mathbb{N}_0$, $\{o_i\}_{i=1}^k$ is sampled from i.i.d. distribution, and*

$$\mathbb{E}\left[\boldsymbol{\xi}(o_k; \boldsymbol{z}_k)|\mathcal{F}_{k-1}\right] = 0,$$

*where $\mathcal{F}_k := \sigma(o_1, o_2, \ldots, o_k)$ for $k \in \mathbb{N}$.*

3. *There exists a positive symmetric definite matrix $\boldsymbol{Q} \in \mathbb{R}^{2Nq \times 2Nq}$ and positive real constant $\kappa$ such that, for $k \in \mathbb{N}_0$,*

$$\boldsymbol{z}_k^\top \boldsymbol{E} \boldsymbol{Q} \boldsymbol{z}_k \leq -\kappa \left\| \boldsymbol{z}_k \right\|_2.$$

We will introduce one lemma:.

**Lemma A.12.** *Under the Assumption A.11, for $k \in \mathbb{N}_0$, we have*

$$\mathbb{E}\left[(\boldsymbol{z}_{k+1} - \boldsymbol{z}_k)^\top \boldsymbol{Q}(\boldsymbol{z}_{k+1} - \boldsymbol{z}_k)\right] \leq 2\alpha_k^2 \left\| \boldsymbol{Q} \right\|_2 \left(\left(\left\| \boldsymbol{E} \right\|_2^2 + C_1\right) \mathbb{E}\left[\left\| \boldsymbol{z}_k \right\|_2^2\right] + C_2\right).$$

*Proof.* We will first consider the conditional expectation:

$$\mathbb{E}\left[(\boldsymbol{z}_{k+1} - \boldsymbol{z}_k)^\top \boldsymbol{Q}(\boldsymbol{z}_{k+1} - \boldsymbol{z}_k)\right]$$
$$\leq \left\| \boldsymbol{Q} \right\|_2 \mathbb{E}\left[\left\| \boldsymbol{z}_{k+1} - \boldsymbol{z}_k \right\|_2^2\right]$$
$$= \left\| \boldsymbol{Q} \right\|_2 \mathbb{E}\left[\left\| \alpha_k \boldsymbol{E} \boldsymbol{z}_k + \alpha_k \boldsymbol{\xi}(o_k; \boldsymbol{z}_k) \right\|_2^2\right]$$
$$\leq 2\alpha_k^2 \left\| \boldsymbol{Q} \right\|_2 \left(\mathbb{E}\left[\left\| \boldsymbol{E} \right\|_2^2 \left\| \boldsymbol{z}_k \right\|_2^2\right] + \mathbb{E}\left[\left\| \boldsymbol{\xi}(o_k; \boldsymbol{z}_k) \right\|_2^2\right]\right)$$
$$\leq 2\alpha_k^2 \left\| \boldsymbol{Q} \right\|_2 \left(\left(\left\| \boldsymbol{E} \right\|_2^2 + C_1\right) \mathbb{E}\left[\left\| \boldsymbol{z}_k \right\|_2^2\right] + C_2\right).$$

The first inequality follows from positive definiteness of $\boldsymbol{Q}$. The first equality follows from the update in (23). The second inequality follows from the relation $\left\| \boldsymbol{a} + \boldsymbol{b} \right\|_2^2 \leq 2\left\| \boldsymbol{a} \right\|_2^2 + 2\left\| \boldsymbol{b} \right\|_2^2$ for $\boldsymbol{a}, \boldsymbol{b} \in \mathbb{R}^{2Nq}$. The last inequality follows from the first item in Assumption A.11. $\qquad\square$

**Theorem A.13.** *Suppose Assumption A.11 holds, and let $V(\boldsymbol{z}) := \boldsymbol{z}^\top \boldsymbol{Q} \boldsymbol{z}$ for $\boldsymbol{z} \in \mathbb{R}^{2Nq}$.*

1. *Suppose we use constant step-size, i.e., $\alpha_0 = \alpha_1 = \cdots = \alpha_k$, and $\alpha_0 \leq \frac{\kappa \lambda_{\min}(\boldsymbol{Q})}{2\lambda_{\max}(\boldsymbol{Q}) \left\| \boldsymbol{Q} \right\|_2 (E^2 + C_1)}$, where $\left\| \boldsymbol{E} \right\|_2 \leq E$. For $k \in \mathbb{N}_0$, we have*

$$\mathbb{E}\left[V(\boldsymbol{z}_{k+1})\right] \leq \exp\left(-\frac{\kappa}{\lambda_{\max}(\boldsymbol{Q})} k\alpha_0\right) V(\boldsymbol{x}_0) + 2\alpha_0 C_2 \left\| \boldsymbol{Q} \right\|_2 \frac{\lambda_{\max}(\boldsymbol{Q})}{\kappa} + 2\alpha_0^2 \left\| \boldsymbol{Q} \right\|_2 C_2.$$

2. *Suppose we have $\alpha_t = \frac{h_1}{t + h_2}$ for $t \in \mathbb{N}_0$ and $h_1 \geq \max\{2, \frac{2\lambda_{\max}(\boldsymbol{Q})}{\kappa}\}$ and $\max\left\{2, h_1, h_1 \frac{2\lambda_{\max}(\boldsymbol{Q}) \left\| \boldsymbol{Q} \right\|_2 (E^2 + C_1)}{\kappa \lambda_{\min}(\boldsymbol{Q})}\right\} \leq h_2$. Then, we have*

$$\mathbb{E}\left[V(\boldsymbol{z}_{k+1})\right] \leq \left(\frac{h_2}{k + h_2}\right)^{\frac{h_1 \kappa}{\lambda_{\max}(\boldsymbol{Q})}} V(\boldsymbol{x}_0) + \frac{2\left\| \boldsymbol{Q} \right\|_2 C_2 h_1^2}{(k - 1 + h_2)} \frac{2^{\frac{2h_1 \kappa}{\lambda_{\max}(\boldsymbol{Q})}}}{\frac{h_1 \kappa}{\lambda_{\max}(\boldsymbol{Q})} - 1} + 2\alpha_k^2 \left\| \boldsymbol{Q} \right\|_2 C_2.$$

*Proof.* From simple algebraic manipulation in Srikant and Ying (2019), we have the following decomposition:

$$\mathbb{E}\left[V(\boldsymbol{z}_{k+1}) - V(\boldsymbol{z}_k)\right]$$
$$= \mathbb{E}\left[(\boldsymbol{z}_{k+1} - \boldsymbol{z}_k)^\top \boldsymbol{Q}(\boldsymbol{z}_{k+1} - \boldsymbol{z}_k)\right] + \mathbb{E}\left[2\boldsymbol{z}_k^\top \boldsymbol{Q} \boldsymbol{z}_{k+1}\right] - 2\mathbb{E}\left[V(\boldsymbol{z}_k)\right]$$
$$= \mathbb{E}\left[(\boldsymbol{z}_{k+1} - \boldsymbol{z}_k)^\top \boldsymbol{Q}(\boldsymbol{z}_{k+1} - \boldsymbol{z}_k)\right] + \mathbb{E}\left[2\boldsymbol{z}_k^\top \boldsymbol{Q}(\boldsymbol{z}_{k+1} - \boldsymbol{z}_k)\right]$$
$$= \underbrace{\mathbb{E}\left[(\boldsymbol{z}_{k+1} - \boldsymbol{z}_k)^\top \boldsymbol{Q}(\boldsymbol{z}_{k+1} - \boldsymbol{z}_k)\right]}_{I_1} + \underbrace{\mathbb{E}\left[2\boldsymbol{z}_k^\top \boldsymbol{Q}(\boldsymbol{z}_{k+1} - \boldsymbol{z}_k - \alpha_k \boldsymbol{E} \boldsymbol{z}_k)\right]}_{I_2} + \underbrace{2\alpha_k \mathbb{E}\left[\boldsymbol{z}_k^\top \boldsymbol{Q} \boldsymbol{E} \boldsymbol{z}_k\right]}_{I_3}.$$

$$(24)$$

To bound $I_1$, the result in Lemma A.12 yields

$$\mathbb{E}\left[(\boldsymbol{z}_{k+1} - \boldsymbol{z}_k)^\top \boldsymbol{Q}(\boldsymbol{z}_{k+1} - \boldsymbol{z}_k)\right] \leq 2\alpha_k^2 \left\| \boldsymbol{Q} \right\|_2 \left(\left(\left\| \boldsymbol{E} \right\|_2^2 + C_1\right) \mathbb{E}\left[\left\| \boldsymbol{z}_k \right\|_2^2\right] + C_2\right).$$

The term $I_2$ becomes zero due to the second item in Assumption A.11, which leads to $\mathbb{E}\left[2z_k^\top \boldsymbol{Q}(z_{k+1} - z_k - \alpha_k \boldsymbol{E} z_k)\right] = \alpha_k \mathbb{E}\left[2z_k^\top \boldsymbol{Q}\mathbb{E}\left[\boldsymbol{\xi}(o_k; z_k)|\mathcal{F}_{k-1}\right]\right] = 0$. Finally we can apply the third item in Assumption A.11 to bound $I_3$. Collecting the terms to bound (24), we get

$$
\begin{aligned}
\mathbb{E}[V(z_{k+1}) - V(z_k)] \leq & 2\alpha_k^2 \|\boldsymbol{Q}\|_2 \left(\left(\|\boldsymbol{E}\|_2^2 + C_1\right) \mathbb{E}\left[\|z_k\|_2^2\right] + C_2\right) - 2\kappa\alpha_k \|z_k\|_2^2 \\
\leq & 2\alpha_k^2 \|\boldsymbol{Q}\|_2 \left(\frac{\|\boldsymbol{E}\|_2^2 + C_1}{\lambda_{\min}(\boldsymbol{Q})} \mathbb{E}\left[V(z_k)\right] + C_2\right) - 2\frac{\kappa}{\lambda_{\max}(\boldsymbol{Q})}\alpha_k \mathbb{E}\left[V(z_k)\right] \\
= & \left(2\alpha_k^2 \|\boldsymbol{Q}\|_2 \frac{\|\boldsymbol{E}\|_2^2 + C_1}{\lambda_{\min}(\boldsymbol{Q})} - 2\frac{\kappa}{\lambda_{\max}(\boldsymbol{Q})}\alpha_k\right) \mathbb{E}\left[V(z_k)\right] + 2\alpha_k^2 \|\boldsymbol{Q}\|_2 C_2.
\end{aligned}
\tag{25}
$$

The second inequality follows from $\lambda_{\min}(\boldsymbol{Q}) \|z\|_2^2 \leq \|z\|_{\boldsymbol{Q}}^2 \leq \lambda_{\max}(\boldsymbol{Q}) \|z\|_2^2$. Moreover, the step-size conditions for both constant step-size and diminishing step-size leads to

$$
2\|\boldsymbol{Q}\|_2 \frac{\|\boldsymbol{E}\|_2^2 + C_1}{\lambda_{\min}(\boldsymbol{Q})}\alpha_k^2 - 2\frac{\kappa}{\lambda_{\max}(\boldsymbol{Q})}\alpha_k \leq 2\|\boldsymbol{Q}\|_2 \frac{E^2 + C_1}{\lambda_{\min}(\boldsymbol{Q})}\alpha_k^2 - 2\frac{\kappa}{\lambda_{\max}(\boldsymbol{Q})}\alpha_k \leq -\frac{\kappa}{\lambda_{\max}(\boldsymbol{Q})}\alpha_k.
$$

Applying the above result to (25), we get

$$
\begin{aligned}
& \mathbb{E}[V(z_{k+1})] \\
\leq & \left(1 - \frac{\kappa}{\lambda_{\max}(\boldsymbol{Q})}\alpha_k\right) \mathbb{E}\left[V(z_k)\right] + 2\alpha_k^2 \|\boldsymbol{Q}\|_2 C_2 \\
\leq & \Pi_{i=0}^k \left(1 - \frac{\kappa}{\lambda_{\max}(\boldsymbol{Q})}\alpha_i\right) \mathbb{E}\left[V(z_0)\right] + 2\sum_{i=0}^{k-1}\alpha_i^2 \|\boldsymbol{Q}\|_2 C_2 \Pi_{j=i+1}^k \left(1 - \frac{\kappa}{\lambda_{\max}(\boldsymbol{Q})}\alpha_j\right) + 2\alpha_k^2 \|\boldsymbol{Q}\|_2 C_2 \\
\leq & \exp\left(-\frac{\kappa}{\lambda_{\max}(\boldsymbol{Q})}\sum_{i=0}^k \alpha_i\right) \mathbb{E}\left[V(z_0)\right] + 2\sum_{i=0}^{k-1}\alpha_i^2 \|\boldsymbol{Q}\|_2 C_2 \exp\left(-\frac{\kappa}{\lambda_{\max}(\boldsymbol{Q})}\sum_{j=i+1}^k \alpha_j\right) + 2\alpha_k^2 \|\boldsymbol{Q}\|_2 C_2,
\end{aligned}
\tag{26}
$$

where the last inequality follows from the relation $1 - x \leq \exp(-x)$ for $x \in \mathbb{R}$.

1. First, we will consider the case for the constant step-size. Using the fact that the step-size is constant, we can rewrite in (26) into

$$
\begin{aligned}
& \mathbb{E}\left[V(z_{k+1})\right] \\
\leq & \exp\left(-\frac{\kappa}{\lambda_{\max}(\boldsymbol{Q})}k\alpha_0\right) \mathbb{E}\left[V(z_0)\right] \\
& + 2\sum_{i=0}^{k-1}\alpha_0^2 \|\boldsymbol{Q}\|_2 C_2 \exp\left(-\frac{\kappa}{\lambda_{\max}(\boldsymbol{Q})}\alpha_0(k - i)\right) + 2\alpha_0^2 \|\boldsymbol{Q}\|_2 C_2 \\
\leq & \exp\left(-\frac{\kappa}{\lambda_{\max}(\boldsymbol{Q})}k\alpha_0\right) \mathbb{E}\left[V(z_0)\right] + 2\alpha_0^2 \|\boldsymbol{Q}\|_2 C_2 \frac{\exp\left(-\frac{\kappa}{\lambda_{\max}(\boldsymbol{Q})}\alpha_0\right)}{1 - \exp\left(-\frac{\kappa}{\lambda_{\max}(\boldsymbol{Q})}\alpha_0\right)} + 2\alpha_0^2 \|\boldsymbol{Q}\|_2 C_2.
\end{aligned}
$$

The second inequality follows from summation of geometric series. Since $\exp(x) - 1 \geq x$ for $x > 0$, we have $\frac{1}{\exp(x)-1} \leq \frac{1}{x}$, and this leads to

$$
\begin{aligned}
\mathbb{E}\left[V(z_{k+1})\right] \leq & \exp\left(-\frac{\kappa}{\lambda_{\max}(\boldsymbol{Q})}k\alpha_0\right) \mathbb{E}\left[V(z_0)\right] + 2\alpha_0^2 C_2 \|\boldsymbol{Q}\|_2 \frac{1}{\frac{\kappa}{\lambda_{\max}(\boldsymbol{Q})}\alpha_0} + 2\alpha_0^2 \|\boldsymbol{Q}\|_2 C_2 \\
= & \exp\left(-\frac{\kappa}{\lambda_{\max}(\boldsymbol{Q})}k\alpha_0\right) \mathbb{E}\left[V(z_0)\right] + 2\alpha_0 C_2 \|\boldsymbol{Q}\|_2 \frac{\lambda_{\max}(\boldsymbol{Q})}{\kappa} + 2\alpha_0^2 \|\boldsymbol{Q}\|_2 C_2.
\end{aligned}
$$

2. The result for diminishing step-size becomes

$$
\mathbb{E}\left[V(\boldsymbol{z}_{k+1})\right]
$$

$$
\leq \exp\left(-\frac{\kappa}{\lambda_{\max}(\boldsymbol{Q})} \sum_{i=0}^{k} \alpha_i\right) V(\boldsymbol{z}_0)
$$

$$
+ 2\sum_{i=0}^{k-1} \alpha_i^2 \left\|\boldsymbol{Q}\right\|_2 C_2 \exp\left(-\frac{\kappa}{\lambda_{\max}(\boldsymbol{Q})} \sum_{j=i+1}^{k-1} \alpha_j\right) + 2\alpha_k^2 \left\|\boldsymbol{Q}\right\|_2 C_2
$$

$$
\leq \exp\left(-\frac{h_1\kappa}{\lambda_{\max}(\boldsymbol{Q})} \log\left(\frac{k+h_2}{h_2}\right)\right) \mathbb{E}[V(\boldsymbol{z}_0)]
$$

$$
+ 2\sum_{i=0}^{k-1} \frac{h_1^2}{(i+h_2)^2} \left\|\boldsymbol{Q}\right\|_2 C_2 \exp\left(-\frac{h_1\kappa}{\lambda_{\max}(\boldsymbol{Q})} \log\left(\frac{k-1+h_2}{i+1+h_2}\right)\right) + 2\alpha_k^2 \left\|\boldsymbol{Q}\right\|_2 C_2
$$

$$
\leq \left(\frac{h_2}{k+h_2}\right)^{\frac{h_1\kappa}{\lambda_{\max}(\boldsymbol{Q})}} V(\boldsymbol{z}_0) + 2\sum_{i=0}^{k-1} \frac{h_1^2}{(i+h_2)^2} \left\|\boldsymbol{Q}\right\|_2 C_2 \left(\frac{i+1+h_2}{k-1+h_2}\right)^{\frac{h_1\kappa}{\lambda_{\max}(\boldsymbol{Q})}} + 2\alpha_k^2 \left\|\boldsymbol{Q}\right\|_2 C_2,
$$

The second inequality follows from $\int_{t=0}^{k} \frac{h_1}{t+h_2} dt \leq \sum_{i=0}^{k} \alpha_i$. From the choice of step-size, we have $\frac{h_1\kappa}{\lambda_{\max}(\boldsymbol{Q})} \geq 2$, which leads to

$$
\mathbb{E}\left[V(\boldsymbol{z}_{k+1})\right] \leq \left(\frac{h_2}{k+h_2}\right)^{\frac{h_1\kappa}{\lambda_{\max}(\boldsymbol{Q})}} V(\boldsymbol{z}_0)
$$

$$
+ 2\left\|\boldsymbol{Q}\right\|_2 C_2 \sum_{i=0}^{k-1} \frac{h_1^2}{(i+h_2)^2} \left(\frac{i+1+h_2}{k-1+h_2}\right)^{\frac{h_1\kappa}{\lambda_{\max}(\boldsymbol{Q})}} + 2\alpha_k^2 \left\|\boldsymbol{Q}\right\|_2 C_2
$$

$$
\leq \left(\frac{h_2}{k+h_2}\right)^{\frac{h_1\kappa}{\lambda_{\max}(\boldsymbol{Q})}} V(\boldsymbol{z}_0)
$$

$$
+ \frac{2\left\|\boldsymbol{Q}\right\|_2 C_2 h_1^2}{(k-1+h_2)^{\frac{h_1\kappa}{\lambda_{\max}(\boldsymbol{Q})}}} 2^{\frac{h_1\kappa}{\lambda_{\max}(\boldsymbol{Q})}} \sum_{i=0}^{k-1} (i+h_2)^{\frac{h_1\kappa}{\lambda_{\max}(\boldsymbol{Q})}-2} + 2\alpha_k^2 \left\|\boldsymbol{Q}\right\|_2 C_2
$$

$$
\leq \left(\frac{h_2}{k+h_2}\right)^{\frac{h_1\kappa}{\lambda_{\max}(\boldsymbol{Q})}} V(\boldsymbol{z}_0)
$$

$$
+ \frac{2\left\|\boldsymbol{Q}\right\|_2 C_2 h_1^2}{(k-1+h_2)^{\frac{h_1\kappa}{\lambda_{\max}(\boldsymbol{Q})}}} 2^{\frac{h_1\kappa}{\lambda_{\max}(\boldsymbol{Q})}} \int_0^k (s+h_1)^{\frac{h_1\kappa}{\lambda_{\max}(\boldsymbol{Q})}-2} ds + 2\alpha_k^2 \left\|\boldsymbol{Q}\right\|_2 C_2
$$

$$
\leq \left(\frac{h_2}{k+h_2}\right)^{\frac{h_1\kappa}{\lambda_{\max}(\boldsymbol{Q})}} V(\boldsymbol{z}_0)
$$

$$
+ \frac{2\left\|\boldsymbol{Q}\right\|_2 C_2 h_1^2}{(k-1+h_2)^{\frac{h_1\kappa}{\lambda_{\max}(\boldsymbol{Q})}}} \frac{2^{\frac{h_1\kappa}{\lambda_{\max}(\boldsymbol{Q})}}}{\frac{h_1\kappa}{\lambda_{\max}(\boldsymbol{Q})}-1} (k+h_1)^{\frac{h_1\kappa}{\lambda_{\max}(\boldsymbol{Q})}-1} + 2\alpha_k^2 \left\|\boldsymbol{Q}\right\|_2 C_2
$$

$$
\leq \left(\frac{h_2}{k+h_2}\right)^{\frac{h_1\kappa}{\lambda_{\max}(\boldsymbol{Q})}} V(\boldsymbol{z}_0)
$$

$$
+ \frac{2\left\|\boldsymbol{Q}\right\|_2 C_2 h_1^2}{(k-1+h_2)} \frac{2^{2\frac{h_1\kappa}{\lambda_{\max}(\boldsymbol{Q})}}}{\frac{h_1\kappa}{\lambda_{\max}(\boldsymbol{Q})}-1} + 2\alpha_k^2 \left\|\boldsymbol{Q}\right\|_2 C_2
$$

The second inequality follows from the fact that $i+h_2+1 \leq 2i+2h_2$ for $i \in \mathbb{N}$. The last inequality follows from the fact that $k+h_1 \leq 2k-2+2h_2$.

$\square$

## A.9 Proof of Theorem 4.2

Let us prove the first item in Theorem 4.2, which is the constant step-size case. To this end, we will apply Theorem A.13 in the Appendix Section A.8, and it is enough to check the conditions in Assumption A.11 in the Appendix Section A.8. Let $\boldsymbol{z}_k := \begin{bmatrix} \tilde{\boldsymbol{\theta}}_k \\ \bar{\boldsymbol{L}}\bar{\boldsymbol{L}}^\dagger \tilde{\boldsymbol{w}}_k \end{bmatrix}$. The first item in Assumption A.11 follows from Lemma A.10 in the Appendix Section A.8. That is, the constants in the first item in Assumption A.11 becomes

$$C_1 = 16, \quad C_2 = \frac{32NR_{\max}^2}{w^2(1-\gamma)^2}, \quad E = 2 + 2\lambda_{\max}(\bar{\boldsymbol{L}}).$$

The second item in Assumption A.11 is straightforward from the fact that $(s_k, s_k', r_k)$ is sampled from i.i.d. distribution.

The third item in Assumption A.11 is satisfied by letting $\kappa = \min\left\{1, \eta\lambda_{\min}^+(\bar{\boldsymbol{L}})^2\right\}/2$, which follows from Lemma 4.1. Therefore, from the first item in Theorem A.13, letting the constant step-size to satisfy

$$\alpha_0 \leq \frac{\min\left\{1, \eta\lambda_{\min}^+(\bar{\boldsymbol{L}})^2\right\}}{4(20 + 8\lambda_{\max}(\bar{\boldsymbol{L}}) + 4\lambda_{\max}(\bar{\boldsymbol{L}})^2)} \frac{\lambda_{\min}(\boldsymbol{G})}{\lambda_{\max}(\boldsymbol{G})^2},$$

Hence, there exists $\bar{\alpha}$ such that

$$\begin{aligned}
\bar{\alpha} =& \mathcal{O}\left(\frac{\min\left\{1, \eta\lambda_{\min}^+(\bar{\boldsymbol{L}})^2\right\}}{\left(20 + 8\lambda_{\max}(\bar{\boldsymbol{L}}) + 4\lambda_{\max}(\bar{\boldsymbol{L}})^2\right)\left(\frac{8+\eta+4\eta^2\lambda_{\max}(\bar{\boldsymbol{L}})^2}{\eta(1-\gamma)w}\right)}\right) \\
=& \mathcal{O}\left(\frac{\min\left\{1, \eta\lambda_{\min}^+(\bar{\boldsymbol{L}})^2\right\}}{\lambda_{\max}(\bar{\boldsymbol{L}})^2\left(\frac{8}{\eta} + 4\eta\lambda_{\max}(\bar{\boldsymbol{L}})^2\right)}(1-\gamma)w\right),
\end{aligned}$$

since $\|\boldsymbol{G}\|_2 = \Theta\left(\frac{\lambda_{\max}(\boldsymbol{L})^2}{(1-\gamma)w}\right)$. Letting $\boldsymbol{x}_0 := \begin{bmatrix} \bar{\boldsymbol{\theta}}_0 - \boldsymbol{1}_N \otimes \boldsymbol{\theta}_c \\ \bar{\boldsymbol{L}}\bar{\boldsymbol{L}}^\dagger\left(\bar{\boldsymbol{w}}_0 - \frac{1}{\eta}\bar{\boldsymbol{w}}_\infty\right) \end{bmatrix}$, This leads to the following result for the convergence rate:

$$\begin{aligned}
& \mathbb{E}\left[\left\|\tilde{\boldsymbol{\theta}}_{k+1}\right\|_2^2 + \left\|\bar{\boldsymbol{L}}\bar{\boldsymbol{L}}^\dagger\tilde{\boldsymbol{w}}_{k+1}\right\|_2^2\right] \\
\leq& \frac{\lambda_{\max}(\boldsymbol{G})}{\lambda_{\min}(\boldsymbol{G})}\exp\left(-\frac{\kappa}{\lambda_{\max}(\boldsymbol{G})}k\alpha_0\right)\|\boldsymbol{x}_0\|_2^2 \\
& + \frac{\|\boldsymbol{G}\|_2}{\lambda_{\min}(\boldsymbol{G})}2\alpha_0 C_2 \frac{\lambda_{\max}(\boldsymbol{G})}{\kappa} + \frac{1}{\lambda_{\min}(\boldsymbol{G})}2\alpha_0^2\|\boldsymbol{G}\|_2 C_2 \\
\leq& 4\exp\left(-\frac{\min\left\{1, \eta\lambda_{\min}^+(\bar{\boldsymbol{L}})^2\right\}}{2\left(\frac{8+\eta+4\eta^2\lambda_{\max}(\bar{\boldsymbol{L}})^2}{\eta(1-\gamma)w}\right)}k\alpha_0\right)\|\boldsymbol{x}_0\|_2^2 \\
& + 16\alpha_0\left(\frac{32NR_{\max}^2}{w^2(1-\gamma)^2}\right)\frac{8+\eta+4\eta^2\lambda_{\max}(\bar{\boldsymbol{L}})^2}{\eta(1-\gamma)w}\frac{1}{\min\{1, \eta\lambda_{\min}(\bar{\boldsymbol{L}})^2\}} \\
& + 8\alpha_0^2\left(\frac{32NR_{\max}^2}{w^2(1-\gamma)^2}\right) \\
=& \mathcal{O}\left(\exp\left(-(1-\gamma)w\frac{\min\{1, \eta\lambda_{\min}^+(\bar{\boldsymbol{L}})^2\}}{\frac{8}{\eta} + 4\eta\lambda_{\max}(\bar{\boldsymbol{L}})^2}k\alpha_0\right)\|\boldsymbol{x}_0\|_2^2 + \alpha_0\frac{NR_{\max}^2}{w^3(1-\gamma)^3}\frac{2+\eta^2\lambda_{\max}(\bar{\boldsymbol{L}})^2}{\eta\min\{1, \eta\lambda_{\min}(\bar{\boldsymbol{L}})^2\}}\right),
\end{aligned}$$

where the second inequality follows from Lemma 4.1. Dividing by the number of agents, $N$, leads to the desired result.

Similarly we can derive the second item in Theorem 4.2, which corresponds to the diminishing step-size case. From the second item in Theorem A.13, the step-size parameters have the following constraints:

$$h_1 \geq \max\left\{\frac{2\lambda_{\max}(\boldsymbol{G})}{\kappa}, 2\right\} \geq \max\left\{\frac{8 + \eta + 4\eta^2\lambda_{\max}(\bar{\boldsymbol{L}})^2}{\eta(1-\gamma)w}\frac{2}{\min\{1, \eta\lambda_{\min}^+(\bar{\boldsymbol{L}})^2\}}, 2\right\},$$

$$h_2 \geq \max\left\{2, h_1, h_1\frac{2\frac{8+\eta+4\eta^2\lambda_{\max}(\bar{\boldsymbol{L}})^2}{\eta(1-\gamma)w}\left((2 + 2\lambda_{\max}(\bar{\boldsymbol{L}}))^2 + 16\right)}{\min\{1, \eta\lambda_{\min}^+(\bar{\boldsymbol{L}})^2\}}\right\}.$$

It suffices to choose $h_1$ and $h_2$ to have the following order:

$$h_1 = \Theta\left(\frac{2 + \eta^2\lambda_{\max}(\bar{\boldsymbol{L}})^2}{\eta(1-\gamma)w\min\{1, \eta\lambda_{\min}^+(\bar{\boldsymbol{L}})^2\}}\right),$$

$$h_2 = \Theta\left(\frac{2 + \eta^2\lambda_{\max}(\bar{\boldsymbol{L}})^2}{\eta(1-\gamma)w\min\{1, \eta\lambda_{\min}^+(\bar{\boldsymbol{L}})^2\}}\lambda_{\max}(\bar{\boldsymbol{L}})^2 h_1\right)$$

$$= \Theta\left(\frac{\left(2 + \eta^2\lambda_{\max}(\bar{\boldsymbol{L}})^2\right)^2\lambda_{\max}(\bar{\boldsymbol{L}})^2}{\eta^2(1-\gamma)^2w^2\min\{1, \eta\lambda_{\min}^+(\bar{\boldsymbol{L}})^2\}^2}\right).$$

Therefore, the convergence rate becomes

$$\mathbb{E}\left[\left\|\tilde{\boldsymbol{\theta}}_{k+1}\right\|_2^2 + \left\|\bar{\boldsymbol{L}}\bar{\boldsymbol{L}}^\dagger\tilde{\boldsymbol{w}}_{k+1}\right\|_2^2\right]$$

$$\leq \frac{\lambda_{\max}(\boldsymbol{G})}{\lambda_{\min}(\boldsymbol{G})}\left(\frac{h_2}{k+h_2}\right)^2\|\boldsymbol{x}_0\|_2^2$$

$$+ \frac{8h_1^2}{k-1+h_2}\frac{32NR_{\max}^2}{w^2(1-\gamma)^2}\frac{4^{\frac{h_1\kappa}{\lambda_{\max}(\boldsymbol{G})}}}{\frac{h_1\kappa}{\lambda_{\max}(\boldsymbol{G})} - 1} + 16\left(\frac{h_1}{k+h_2}\right)^2\frac{32NR_{\max}^2}{w^2(1-\gamma)^2}$$

$$= \mathcal{O}\left(\frac{1}{k}\frac{(1 + \eta^2\lambda_{\max}(\bar{\boldsymbol{L}})^2)^2}{\eta^2\min\{1, \eta\lambda_{\min}^+(\bar{\boldsymbol{L}})^2\}^2}\frac{NR_{\max}^2}{w^4(1-\gamma)^4}\right).$$

Dividing by the number of agents, $N$, completes the proof.

## A.10 MARKOVIAN OBSERVATION MODEL

We will consider a general stochastic recursive model with Markovian observation samples, for $k \in \mathbb{N}_0$:

$$\boldsymbol{z}_{k+1} = \boldsymbol{z}_k + \alpha_k(\boldsymbol{E}\boldsymbol{z}_k + \boldsymbol{\xi}(o_k; \boldsymbol{z}_k)), \tag{27}$$

where $\boldsymbol{E} \in \mathbb{R}^{2Nq \times 2Nq}$, $\boldsymbol{z}_k \in \mathbb{R}^{2Nq}$ and $\boldsymbol{\xi}(o_k; \boldsymbol{z}_k) := \boldsymbol{W}(o_k)\boldsymbol{z}_k + \boldsymbol{w}(o_k)$ for $\boldsymbol{W} : \mathcal{S} \times \mathcal{S} \times \Pi_{i=1}^N I \to \mathbb{R}^{2Nq \times 2Nq}$, where $I$ is closed interval $[-R_{\max}, R_{\max}]$ in $\mathbb{R}$, and $\boldsymbol{w} : \mathcal{S} \times \mathcal{S} \times \Pi_{i=1}^N I \to \mathbb{R}^{2Nq}$. We assume that the the sequence $\{o_k \in \mathcal{S} \times \mathcal{S} \times \Pi_{i=1}^N I\}_{k\in\mathbb{N}}$ is generated by an ergodic Markov chain. The proof follows the spirit of (Srikant and Ying, 2019). We will denote $T \in \mathbb{N}$ as the total number of iterations and the mixing time $\tau := \tau(\alpha_T)$ will be defined as in (14). We first introduce a set of assumptions:

**Assumption A.14.**  *1. For any $o \in \mathcal{S} \times \mathcal{S} \times \Pi_{i=1}^N I$, we have*

$$\|\boldsymbol{W}(o)\|_2 \leq C_1, \quad \|\boldsymbol{w}(o)\|_2 \leq C_2.$$

*2. For $k \geq \tau$, there exists a positive constant $\Xi$ such that*

$$\|\mathbb{E}\left[\boldsymbol{\xi}(o_k; \boldsymbol{z}_{k-\tau})|\mathcal{F}_{k-\tau}\right]\|_2 \leq \Xi\alpha_T(\|\boldsymbol{z}_{k-\tau}\|_2 + 1),$$

*where $\mathcal{F}_{k-\tau} = \sigma(o_1, o_2, \ldots, o_{k-\tau})$.*

*3. For $k \in \mathbb{N}_0$, there exists a positive definite matrix $\boldsymbol{Q} \in \mathbb{R}^{2Nq \times 2Nq}$ and a positive constant $\kappa$ such that*

$$2\boldsymbol{z}_k^\top\boldsymbol{E}\boldsymbol{Q}\boldsymbol{z}_k \leq -\kappa\|\boldsymbol{z}_k\|_2.$$

For simplicity of the proof, we will denote $E_1 := C_1 + E$ where $\|\boldsymbol{E}\|_2 \leq E$. We first present several useful lemmas.

**Lemma A.15.** *1. For $k \geq \tau$ and $k - \tau + 1 \leq s \leq k - 1$, using constant step-size, i.e., $\alpha_0 = \alpha_1 = \cdots = \alpha_T$ such that $\tau \alpha_0 E_1 \leq \ln 2$, we have*

$$\|\boldsymbol{z}_{s+1}\|_2 \leq 2 \|\boldsymbol{z}_{k-\tau}\|_2 + \frac{4C_2}{E_1}.$$

*2. For $k \geq \tau$ and $k - \tau + 1 \leq s \leq k - 1$, using diminishing step-size, i.e., $\alpha_t = \frac{h_1}{t+h_2}$ for $t \in \mathbb{N}_0$ such that $\frac{\tau - 1 + 2^{1/E_1 h_1}}{2^{1/E_1 h_1} - 1} \leq h_2$, we have*

$$\|\boldsymbol{z}_{s+1}\|_2 \leq 2 \|\boldsymbol{z}_{k-\tau}\|_2 + 4C_2 \tau \alpha_{k-\tau}.$$

*Proof.* Applying triangle inequality to the recursion in (27), we get

$$\|\boldsymbol{z}_{s+1}\|_2 \leq (1 + \alpha_s E_1) \|\boldsymbol{z}_s\|_2 + \alpha_s C_2.$$

Recursive formula leads to

$$\|\boldsymbol{z}_{s+1}\|_2 \leq \Pi_{j=k-\tau}^s (1 + \alpha_j E_1) \|\boldsymbol{z}_{k-\tau}\|_2 + \sum_{i=k-\tau}^{s-1} C_2 \alpha_i \Pi_{j=i+1}^s (1 + \alpha_j E_1) + \alpha_s C_2$$

$$\leq \exp\left(\sum_{i=k-\tau}^s \alpha_i E_1\right) \|\boldsymbol{z}_{k-\tau}\|_2 + \sum_{i=k-\tau}^{s-1} C_2 \alpha_i \exp\left(\sum_{j=i+1}^s \alpha_j E_1\right) + \alpha_s C_2, \quad (28)$$

where the last inequality follows from the relation $1 + x \leq \exp(x)$ for $x \in \mathbb{R}$.

1. We will first prove the case when the step-size is constant. Using the fact that $\alpha_0 = \alpha_1 = \cdots = \alpha_s$, we can rewrite (28) as follows:

$$\|\boldsymbol{z}_{s+1}\|_2 \leq \exp\left(\tau \alpha_0 E_1\right) \|\boldsymbol{z}_{k-\tau}\|_2 + \sum_{i=k-\tau}^{s-1} C_2 \alpha_0 \exp(\alpha_0 E_1(s - i)) + \alpha_0 C_2$$

$$\leq \exp\left(\tau \alpha_0 E_1\right) \|\boldsymbol{z}_{k-\tau}\|_2 + C_2 \alpha_0 \frac{\exp((\tau - 1)\alpha_0 E_1)}{1 - \exp(-\alpha_0 E_1)} + \alpha_s C_2$$

$$= \exp\left(\tau \alpha_0 E_1\right) \|\boldsymbol{z}_{k-\tau}\|_2 + C_2 \alpha_0 \frac{\exp(\tau \alpha_0 E_1)}{\exp(\alpha_0 E_1) - 1} + \alpha_s C_2$$

$$\leq 2 \|\boldsymbol{z}_{k-\tau}\|_2 + C_2 \frac{2}{E_1} + \alpha_0 C_2$$

$$\leq 2 \|\boldsymbol{z}_{k-\tau}\|_2 + \frac{4C_2}{E_1}.$$

The second last inequality follows from the condition on the step-size, $\tau \alpha_0 E_1 \leq \ln 2$, and the fact that $\exp(x) \geq x + 1$ for $x \in \mathbb{R}$.

2. We will prove the case for diminishing step-size. Plugging in $\alpha_t = \frac{h_1}{t+h_2}$ for $t \in \mathbb{N}$ to (28), we have

$$
\begin{aligned}
\|\boldsymbol{z}_{s+1}\|_2 &\leq \exp\left(E_1 h_1 \int_{k-\tau-1}^{s} \frac{1}{t+h_2} dt\right) \|\boldsymbol{z}_{k-\tau}\|_2 \\
&\quad + C_2 \sum_{i=k-\tau}^{s-1} \alpha_i \exp\left(E_1 h_1 \int_{i}^{s} \frac{1}{t+h_2} dt\right) + \alpha_s C_2 \\
&\leq \left(\frac{s+h_2}{k-\tau-1+h_2}\right)^{E_1 h_1} \|\boldsymbol{z}_{k-\tau}\|_2 + C_2 \sum_{i=k-\tau}^{s-1} \alpha_i \left(\frac{s+h_2}{i+h_2}\right)^{E_1 h_1} + \alpha_s C_2 \\
&\leq 2 \|\boldsymbol{z}_{k-\tau}\|_2 + C_2 \sum_{i=k-\tau}^{k-1} 2\alpha_i + \alpha_s C_2 \qquad (29) \\
&\leq 2 \|\boldsymbol{z}_{k-\tau}\|_2 + 2C_2 \tau \alpha_{k-\tau} + \alpha_s C_2 \\
&\leq 2 \|\boldsymbol{z}_{k-\tau}\|_2 + 4C_2 \tau \alpha_{k-\tau}.
\end{aligned}
$$

The first inequality follows from the fact that $\sum_{i=a}^{b} \frac{1}{t+h_2} \leq \int_{a-1}^{b} \frac{1}{t+h_2} dt$ for $a, b \in \mathbb{N}_0$. The inequality in (29) follows from the following relation that for $k \geq \tau$, $k-\tau+1 \leq s \leq k-1$ and $k-\tau \leq i \leq s-1$, the condition $\frac{\tau-1+2^{1/E_1 h_1}}{2^{1/E_1 h_1}-1} \leq h_2$ leads to

$$
\left(\frac{s+h_2}{i+h_2}\right)^{E_1 h_1} \leq \left(\frac{s+h_2}{k-\tau-1+h_2}\right)^{E_1 h_1} \leq \left(\frac{k-1+h_2}{k-\tau-1+h_2}\right)^{E_1 h_1} \leq 2.
$$

The last inequality follows since $\frac{k-1+h_2}{k-\tau-1+h_2}$ is decreasing function in $k$ and it suffices to satisfy the inequality when $k = \tau$. This completes the proof.

$\square$

The following lemma shows that the difference between $\boldsymbol{z}_k$ and $\boldsymbol{z}_{k-\tau}$ for $k \geq \tau$ will not be large:

**Lemma A.16.** *1. Considering constant step-size, i.e., $\alpha_0 = \alpha_1 = \cdots = \alpha_T$, with $\alpha_0 \leq \frac{1}{100\tau \max\{E_1, C_2\}}$, for $k \geq \tau$, we have*

$$
\begin{aligned}
\|\boldsymbol{z}_k - \boldsymbol{z}_{k-\tau}\|_2 &\leq 4E_1 \alpha_0 \tau \|\boldsymbol{z}_k\|_2 + 10C_2 \alpha_0 \tau, \\
\|\boldsymbol{z}_k - \boldsymbol{z}_{k-\tau}\|_2^2 &\leq E_1 \alpha_0 \tau \|\boldsymbol{z}_k\|_2^2 + C_2 \alpha_0 \tau.
\end{aligned}
$$

*2. Considering diminishing step-size, i.e., $\alpha_t = \frac{h_1}{t+h_2}$ for $t \in \mathbb{N}$ such that $\max\left\{\frac{\tau-1+2^{1/E_1 h_1}}{2^{1/E_1 h_1}-1}, 32\tau E_1 h_1, 32\tau C_2 h_1\right\} \leq h_2$, for $k \geq \tau$, we have*

$$
\|\boldsymbol{z}_k - \boldsymbol{z}_{k-\tau}\|_2 \leq 4E_1 \alpha_{k-\tau} \tau \|\boldsymbol{z}_k\|_2 + 4C_2 \alpha_{k-\tau} \tau, \qquad (30)
$$

$$
\|\boldsymbol{z}_k - \boldsymbol{z}_{k-\tau}\|_2^2 \leq E_1 \alpha_{k-\tau} \tau \|\boldsymbol{z}_k\|_2^2 + C_2 \alpha_{k-\tau} \tau. \qquad (31)
$$

*Proof.* We have the following relation:

$$
\begin{aligned}
\|\boldsymbol{z}_k - \boldsymbol{z}_{k-\tau}\|_2 &\leq \sum_{i=0}^{\tau-1} \|\boldsymbol{z}_{i+1+k-\tau} - \boldsymbol{z}_{i+k-\tau}\| \\
&= \sum_{i=0}^{\tau-1} \alpha_{i+k-\tau} \|\boldsymbol{E}\boldsymbol{z}_{i+k-\tau} + \boldsymbol{\xi}(o_{i+k-\tau}; \boldsymbol{z}_{i+k-\tau})\|_2 \\
&\leq \sum_{i=0}^{\tau-1} \alpha_{i+k-\tau}(E_1 \|\boldsymbol{z}_{i+k-\tau}\|_2 + C_2) \qquad (32)
\end{aligned}
$$

The first inequality follows from triangle inequality. The first equality follows from the update in (27). The last inequality follows from the first item in Assumption A.14.

1. Considering the constant step-size, we have

$$\|z_k - z_{k-\tau}\|_2 \leq \alpha_0 \sum_{i=0}^{\tau-1} \left( E_1 \left( 2\|z_{k-\tau}\|_2 + \frac{4C_2}{E_1} \right) + C_2 \right)$$

$$= \alpha_0 \sum_{i=0}^{\tau-1} (2E_1 \|z_{k-\tau}\|_2 + 5C_2)$$

$$= 2E_1 \alpha_0 \tau \|z_{k-\tau}\|_2 + 5C_2 \alpha_0 \tau.$$

The first inequality follows applying Lemma A.15 to (32). Since we have $E_1 \alpha_0 \tau \leq \frac{1}{4}$, using triangle inequality we get

$$\|z_k - z_{k-\tau}\|_2 \leq 2E_1 \alpha_0 \tau \|z_k - z_{k-\tau}\|_2 + 2E_1 \alpha_0 \tau \|z_k\|_2 + 5C_2 \alpha_0 \tau,$$
$$\|z_k - z_{k-\tau}\|_2 \leq 4E_1 \alpha_0 \tau \|z_k\|_2 + 10C_2 \alpha_0 \tau.$$

Moreover, using the relation $(a+b)^2 \leq 2a^2 + 2b^2$ for $a, b \in \mathbb{R}$, we have

$$\|z_k - z_{k-\tau}\|_2^2 \leq 2(4E_1 \alpha_0 \tau)^2 \|z_{k-\tau}\|_2^2 + (10C_2 \alpha_0 \tau)^2$$

$$\leq E_1 \alpha_0 \tau \|z_{k-\tau}\|_2^2 + C_2 \alpha_0 \tau.$$

The last inequality follows from the step-size condition that $\alpha_0 \leq \frac{1}{100\tau \max\{E_1, C_2\}}$.

2. Considering diminishing step-size, applying Lemma A.15 to (32), we get

$$\|z_k - z_{k-\tau}\|_2 \leq \alpha_{k-\tau} \sum_{i=0}^{\tau-1} (E_1 (2\|z_{k-\tau}\|_2 + 4C_2 \tau \alpha_{k-\tau}) + C_2)$$

$$= \alpha_{k-\tau} \left( 2\tau E_1 \|z_{k-\tau}\|_2 + 4E_1 C_2 \tau^2 \alpha_{k-\tau} + C_2 \tau \right)$$

$$\leq \alpha_{k-\tau} (2\tau E_1 \|z_{k-\tau}\|_2 + 2C_2 \tau)$$

$$= 2E_1 \alpha_{k-\tau} \tau \|z_{k-\tau}\|_2 + 2C_2 \alpha_{k-\tau} \tau.$$

The first inequality follows from the second item in Lemma A.15. The condition $h_2 \geq 32E_1 \tau h_1$ leads to the last inequality. Moreover, since $\alpha_{k-\tau} \leq \frac{1}{4\tau E_1}$ for $k \geq \tau$, we have:

$$\|z_k - z_{k-\tau}\|_2 \leq 2E_1 \alpha_{k-\tau} \tau \|z_{k-\tau} - z_k\|_2 + 2E_1 \alpha_{k-\tau} \tau \|z_k\|_2 + 2\alpha_{k-\tau} C_2 \tau$$

$$\leq 4E_1 \alpha_{k-\tau} \tau \|z_k\|_2 + 4C_2 \alpha_{k-\tau} \tau.$$

The first inequality follows triangle inequality. Furthermore, using the relation $(a+b)^2 \leq 2a^2 + 2b^2$ for $a, b \in \mathbb{R}$, we have

$$\|z_k - z_{k-\tau}\|_2^2 \leq 2(4E_1 \alpha_{k-\tau} \tau)^2 \|z_k\|_2^2 + 2(4C_2 \alpha_{k-\tau} \tau)^2$$

$$\leq E_1 \alpha_{k-\tau} \tau \|z_k\|_2^2 + C_2 \alpha_{k-\tau} \tau.$$

The last inequality follows from the step-size condition $\max \{32\tau E_1 h_1, 32\tau C_2 h_1\} \leq h_2$.

$\square$

**Lemma A.17.** *1. Considering constant step-size, i.e., $\alpha_0 = \alpha_1 = \cdots = \alpha_T$, with $\alpha_0 \leq \min\left\{ \frac{1}{100\tau \max\{E_1, C_2\}}, \frac{C_1}{2\Xi} \right\}$, for $k \geq \tau$, we have*

$$\mathbb{E}[z_k^\top Q(\xi(o_k; z_k))]$$

$$\leq \|Q\|_2 \left( (4\Xi + 13C_1 E_1 + 20C_1 C_2 + 4E_1 C_2) \alpha_0 \tau \mathbb{E}\left[ \|z_k\|_2^2 \right] \right.$$

$$\left. + \left( 25C_1 C_2 + 10C_2^2 + 2\Xi + 4E_1 C_2 \right) \alpha_0 \tau \right).$$

2. *Considering diminishing step-size, i.e., $\alpha_t = \frac{h_1}{t + h_2}$ for $t \in \mathbb{N}$ such that $\max\left\{ \frac{\tau - 1 + 2^{1/E_1 h_1}}{2^{1/E_1 h_1} - 1}, 32\tau E_1 h_1, 32\tau C_2 h_1, \frac{\Xi h_1}{2C_1} \right\} \leq h_2$, for $k \geq \tau$, we have*

$$\mathbb{E}[z_k^\top Q(\xi(o_k; z_k))]$$

$$\leq \|Q\|_2 \left( (4\Xi + 13E_1 C_1 + 8C_1 C_2 + 4C_2 E_1) \mathbb{E}\left[ \|z_k\|_2^2 \right] \right.$$

$$\left. + (13C_1 C_2 + 4C_2^2 + 2\Xi + 4C_2 E_1)) \alpha_{k-\tau} \tau. \right.$$

*Proof.* Following the spirit of Srikant and Ying (2019) we can decompose the cross term in to follows four terms:

$$
\begin{aligned}
&\mathbb{E}[\boldsymbol{z}_k^\top \boldsymbol{Q}(\boldsymbol{z}_{k+1} - \boldsymbol{z}_k - \alpha_k \boldsymbol{E}\boldsymbol{z}_k)] \\
=&\alpha_k \mathbb{E}[\boldsymbol{z}_k^\top \boldsymbol{Q}(\boldsymbol{w}(o_k) + \boldsymbol{W}(o_k)\boldsymbol{z}_k))] \\
=&\alpha_k \left( \underbrace{\mathbb{E}[\boldsymbol{z}_{k-\tau}^\top \boldsymbol{Q}(\boldsymbol{w}(o_k) + \boldsymbol{W}(o_k)\boldsymbol{z}_{k-\tau})]}_{I_1} + \underbrace{\mathbb{E}[(\boldsymbol{z}_k - \boldsymbol{z}_{k-\tau})^\top \boldsymbol{Q}(\boldsymbol{w}(o_k) + \boldsymbol{W}(o_k)(\boldsymbol{z}_k - \boldsymbol{z}_{k-\tau}))]}_{I_2} \right. \\
&\left. + \underbrace{\mathbb{E}[(\boldsymbol{z}_k - \boldsymbol{z}_{k-\tau})^\top \boldsymbol{Q}\boldsymbol{W}(o_k)\boldsymbol{z}_{k-\tau}]}_{I_3} + \underbrace{\mathbb{E}[\boldsymbol{z}_{k-\tau}^\top \boldsymbol{Q}\boldsymbol{W}(o_k)(\boldsymbol{z}_k - \boldsymbol{z}_{k-\tau})]}_{I_4} \right).
\end{aligned}
$$

The term $I_1$ can be bounded from the second item in Assumption A.14, which uses the geometric mixing property of the Markov chain.

$$
\begin{aligned}
\mathbb{E}\left[\boldsymbol{z}_{k-\tau}^\top \boldsymbol{Q}(\boldsymbol{w}(o_k) + \boldsymbol{W}(o_k)\boldsymbol{z}_{k-\tau})\right] =&\mathbb{E}\left[\boldsymbol{z}_{k-\tau}^\top \boldsymbol{Q}\mathbb{E}\left[(\boldsymbol{W}(o_k)\boldsymbol{z}_{k-\tau} + \boldsymbol{w}(o_k))|\mathcal{F}_{k-\tau}\right]\right] \\
\leq&\mathbb{E}\left[\|\boldsymbol{z}_{k-\tau}\|_2 \|\boldsymbol{Q}\|_2 \|\mathbb{E}\left[(\boldsymbol{W}(o_k)\boldsymbol{z}_{k-\tau} + \boldsymbol{w}(o_k))|\mathcal{F}_{k-\tau}\right]\|_2\right] \\
\leq&\mathbb{E}\left[\|\boldsymbol{z}_{k-\tau}\|_2 \|\boldsymbol{Q}\|_2 \Xi\alpha_T(\|\boldsymbol{z}_{k-\tau}\|_2 + 1)\right] \\
\leq&\|\boldsymbol{Q}\|_2 \Xi\alpha_T \left(2\mathbb{E}\left[\|\boldsymbol{z}_{k-\tau}\|_2^2\right] + 1\right) \\
\leq&\|\boldsymbol{Q}\|_2 \Xi\alpha_T \mathbb{E}\left[(4\|\boldsymbol{z}_k - \boldsymbol{z}_{k-\tau}\|_2^2 + 4\|\boldsymbol{z}_k\|_2^2 + 2)\right].
\end{aligned}
$$

The first inequality follows from Cauchy-Schwartz inequality. The second inequality follows from the second item in Assumption A.14. The third inequality follows from the relation $a \leq a^2 + 1$ for $a \in \mathbb{R}$. The last inequality follows from the relation $(a+b)^2 \leq 2a^2 + 2b^2$ for $a, b \in \mathbb{R}$.

The term $I_2$ can be bounded as follows:

$$
\begin{aligned}
&\mathbb{E}\left[(\boldsymbol{z}_k - \boldsymbol{z}_{k-\tau})^\top \boldsymbol{Q}(\boldsymbol{\xi}(o_k; \boldsymbol{z}_k - \boldsymbol{z}_{k-\tau}))\right] \\
\leq&\mathbb{E}\left[\|\boldsymbol{z}_k - \boldsymbol{z}_{k-\tau}\|_2 \|\boldsymbol{Q}\|_2 \|\boldsymbol{\xi}(o_k; \boldsymbol{z}_k - \boldsymbol{z}_{k-\tau})\|_2\right] \\
\leq&\mathbb{E}\left[\|\boldsymbol{z}_k - \boldsymbol{z}_{k-\tau}\|_2 \|\boldsymbol{Q}\|_2 (C_1 \|\boldsymbol{z}_k - \boldsymbol{z}_{k-\tau}\|_2 + C_2)\right] \\
=&\mathbb{E}\left[\|\boldsymbol{Q}\|_2 C_1 \|\boldsymbol{z}_k - \boldsymbol{z}_{k-\tau}\|_2^2 + \|\boldsymbol{Q}\|_2 C_2 \|\boldsymbol{z}_k - \boldsymbol{z}_{k-\tau}\|_2\right].
\end{aligned}
$$

The first inequality follows from Cauchy-Schwartz inequality. The second inequality follows from the first item in Assumption A.14.

The term $I_3$ can be bounded as follows:

$$
\begin{aligned}
&\mathbb{E}\left[(\boldsymbol{z}_k - \boldsymbol{z}_{k-\tau})^\top \boldsymbol{Q}\boldsymbol{W}(o_k)\boldsymbol{z}_{k-\tau}\right] \\
\leq&\mathbb{E}\left[\|\boldsymbol{z}_k - \boldsymbol{z}_{k-\tau}\|_2 \|\boldsymbol{Q}\|_2 (C_1 \|\boldsymbol{z}_{k-\tau}\|_2)\right] \\
\leq&\mathbb{E}\left[C_1 \|\boldsymbol{Q}\|_2 \|\boldsymbol{z}_k - \boldsymbol{z}_{k-\tau}\|_2^2\right] + \mathbb{E}\left[C_1 \|\boldsymbol{Q}\|_2 \|\boldsymbol{z}_k - \boldsymbol{z}_{k-\tau}\|_2 \|\boldsymbol{z}_k\|_2\right].
\end{aligned}
$$

The first inequality follows from Cauchy-Schwartz inequality and the first item in Assumption A.14. The second inequality follows from triangle inequality.

The term $I_4$ can be bounded as

$$
\begin{aligned}
&\mathbb{E}\left[\boldsymbol{z}_{k-\tau}^\top \boldsymbol{Q}\boldsymbol{W}(o_k)(\boldsymbol{z}_k - \boldsymbol{z}_{k-\tau})\right] \\
\leq&\mathbb{E}\left[\|\boldsymbol{z}_{k-\tau}\|_2 \|\boldsymbol{Q}\|_2 (C_1 \|\boldsymbol{z}_k - \boldsymbol{z}_{k-\tau}\|_2)\right] \\
\leq&\mathbb{E}\left[C_1 \|\boldsymbol{Q}\|_2 (\|\boldsymbol{z}_k - \boldsymbol{z}_{k-\tau}\|_2^2 + \|\boldsymbol{z}_k - \boldsymbol{z}_{k-\tau}\|_2 \|\boldsymbol{z}_k\|_2)\right].
\end{aligned}
$$

The first inequality follows from Cauchy-Schwartz inequality and the first item in Assumption A.14.

Collecting the terms to bound $I_1, I_2, I_3$ and $I_4$, we get

$$\mathbb{E}[\boldsymbol{z}_k^\top \boldsymbol{Q}(\boldsymbol{\xi}(o_k; \boldsymbol{z}_k))]$$
$$\leq \|\boldsymbol{Q}\|_2 \left( 4\Xi\alpha_T \mathbb{E}\left[\|\boldsymbol{z}_k\|_2^2\right] + (3C_1 + 4\Xi\alpha_T)\mathbb{E}\left[\|\boldsymbol{z}_k - \boldsymbol{z}_{k-\tau}\|_2^2\right] \right.$$
$$+ 2C_1\mathbb{E}\left[\|\boldsymbol{z}_k - \boldsymbol{z}_{k-\tau}\|_2 \|\boldsymbol{z}_k\|_2\right] + C_2\mathbb{E}\left[\|\boldsymbol{z}_k - \boldsymbol{z}_{k-\tau}\|_2\right] + 2\Xi\alpha_T \big)$$
$$\leq \|\boldsymbol{Q}\|_2 \left( 4\Xi\alpha_T \mathbb{E}\left[\|\boldsymbol{z}_k\|_2^2\right] + 5C_1\mathbb{E}\left[\|\boldsymbol{z}_k - \boldsymbol{z}_{k-\tau}\|_2^2\right] \right. \tag{33}$$
$$+ 2C_1\mathbb{E}\left[\|\boldsymbol{z}_k - \boldsymbol{z}_{k-\tau}\|_2 \|\boldsymbol{z}_k\|_2\right] + C_2\mathbb{E}\left[\|\boldsymbol{z}_k - \boldsymbol{z}_{k-\tau}\|_2\right] + 2\Xi\alpha_T \big). \tag{34}$$

The last inequality follows from the step-size condition that $2\Xi\alpha_T \leq C_1$.

1. For constant step-size case, we have

$$\mathbb{E}[\boldsymbol{z}_k^\top \boldsymbol{Q}(\boldsymbol{\xi}(o_k; \boldsymbol{z}_k))]$$
$$\leq \|\boldsymbol{Q}\|_2 \left( 4\Xi\alpha_0 \mathbb{E}\left[\|\boldsymbol{z}_k\|_2^2\right] + 5C_1 \left( \mathbb{E}\left[ E_1\alpha_0\tau \|\boldsymbol{z}_k\|_2^2 + C_2\alpha_0\tau \right] \right) \right.$$
$$+ 2C_1 \left( \mathbb{E}\left[ 4E_1\alpha_0\tau \|\boldsymbol{z}_k\|_2^2 + 10C_2\alpha_0\tau \|\boldsymbol{z}_k\|_2 \right] \right)$$
$$+ C_2 \left( \mathbb{E}\left[ 4E_1\alpha_0\tau \|\boldsymbol{z}_k\|_2 + 10C_2\alpha_0\tau \right] \right) + 2\Xi\alpha_0 \big)$$
$$\leq \|\boldsymbol{Q}\|_2 \left( (4\Xi\alpha_0 + 13C_1 E_1\alpha_0\tau)\mathbb{E}\left[\|\boldsymbol{z}_k\|_2^2\right] \right.$$
$$+ \left( 20C_1 C_2 + 4E_1 C_2 \right)\alpha_0\tau\mathbb{E}\left[\|\boldsymbol{z}_k\|_2\right] + \left( 5C_1 C_2 + 10C_2^2 + 2\Xi \right)\alpha_0\tau \big)$$
$$\leq \|\boldsymbol{Q}\|_2 \left( (4\Xi + 13C_1 E_1 + 20C_1 C_2 + 4E_1 C_2)\alpha_0\tau\mathbb{E}\left[\|\boldsymbol{z}_k\|_2^2\right] \right.$$
$$+ \left( 25C_1 C_2 + 10C_2^2 + 2\Xi + 4E_1 C_2 \right)\alpha_0\tau \big).$$

The first inequality follows from applying Lemma A.16 to (33) and (34). The last inequality follows from the relation $a \leq a^2 + 1$ for $a \in \mathbb{R}$.

2. Considering diminishing step-size, we get

$$\mathbb{E}[\boldsymbol{z}_k^\top \boldsymbol{Q}(\boldsymbol{\xi}(o_k; \boldsymbol{z}_k))]$$
$$\leq \|\boldsymbol{Q}\|_2 \left( 4\Xi\alpha_T \mathbb{E}\left[\|\boldsymbol{z}_k\|_2^2\right] + 5C_1 \left( \mathbb{E}\left[ E_1\alpha_{k-\tau}\tau \|\boldsymbol{z}_k\|_2^2 + C_2\alpha_{k-\tau}\tau \right] \right) \right.$$
$$+ 2C_1 \left( \mathbb{E}\left[ 4E_1\alpha_{k-\tau}\tau \|\boldsymbol{z}_k\|_2^2 + 4C_2\alpha_{k-\tau}\tau \|\boldsymbol{z}_k\|_2 \right] \right)$$
$$+ C_2(4E_1\alpha_{k-\tau}\tau \|\boldsymbol{z}_k\|_2 + 4C_2\alpha_{k-\tau}\tau) + 2\Xi\alpha_T \big)$$
$$\leq \|\boldsymbol{Q}\|_2 \left( (4\Xi\alpha_T + 13E_1 C_1\alpha_{k-\tau}\tau)\mathbb{E}\left[\|\boldsymbol{z}_k\|_2^2\right] \right.$$
$$+ (8C_1 C_2 + 4C_2 E_1)\alpha_{k-\tau}\tau\mathbb{E}\left[\|\boldsymbol{z}_k\|_2\right]$$
$$+ (5C_1 C_2\alpha_{k-\tau}\tau + 4C_2^2\alpha_{k-\tau}\tau + 2\Xi\alpha_T) \big)$$
$$\leq \|\boldsymbol{Q}\|_2 \left( (4\Xi + 13E_1 C_1 + 8C_1 C_2 + 4C_2 E_1)\alpha_{k-\tau}\tau\mathbb{E}\left[\|\boldsymbol{z}_k\|_2^2\right] \right.$$
$$+ (13C_1 C_2 + 4C_2^2 + 2\Xi + 4C_2 E_1)\alpha_{k-\tau}\tau \big).$$

The first inequality follows from applying Lemma A.16 to (33) and (34). The last inequality follows from the relation $a \leq a^2 + 1$ for $a \in \mathbb{R}$.

$\square$

**Lemma A.18.** *For $k \in \mathbb{N}_0$, we have*

$$(\boldsymbol{z}_{k+1} - \boldsymbol{z}_k)^\top \boldsymbol{Q}(\boldsymbol{z}_{k+1} - \boldsymbol{z}_k) \leq 2\alpha_k^2 \|\boldsymbol{Q}\|_2 \left( E_1^2 \|\boldsymbol{z}_k\|_2^2 + C_2^2 \right).$$

*Proof.* We have

$$
\begin{aligned}
(z_{k+1} - z_k)^\top Q(z_{k+1} - z_k) &\leq \|Q\|_2 \|z_{k+1} - z_k\|_2^2 \\
&= \|Q\|_2 \|\alpha_k E z_k + \alpha_k \xi(o_k; z_k)\|_2^2 \\
&\leq \alpha_k^2 \|Q\|_2 (E_1 \|z_k\|_2 + C_2)^2 \\
&\leq 2\alpha_k^2 \|Q\|_2 (E_1^2 \|z_k\|_2^2 + C_2^2).
\end{aligned}
$$

The first inequality follows from Cauchy-Schwartz inequality, and the second inequaltiy follows from the relation $\|a + b\|_2^2 \leq 2\|a\|_2^2 + 2\|b\|_2^2$ for $a, b \in \mathbb{R}^{2Nq}$. $\qquad\square$

**Theorem A.19.** *1. Considering constant step-size, i.e., $\alpha_0 = \alpha_1 = \cdots = \alpha_T$, with $\alpha_0 \leq$*
$\min\left\{\frac{1}{100\tau \max\{E_1, C_2\}}, \frac{C_1}{2\Xi}, \frac{\kappa \lambda_{\min}(Q)}{(4E_1^2 + 4K_1\tau)\lambda_{\max}(Q)\|Q\|_2}\right\}$, *we have, for $\tau \leq k$,*

$$
\begin{aligned}
\mathbb{E}\left[\|z_{k+1}\|_2^2\right] &\leq \frac{\lambda_{\max}(Q)}{\lambda_{\min}(Q)} \exp\left(-\alpha_0 \frac{\kappa}{2\lambda_{\max}(Q)}(k - \tau + 1)\right)\left(2\|z_0\|_2^2 + \frac{4C_2}{E_1}\right) \\
&\quad + \frac{2\|Q\|_2(C_2^2 + K_2\tau)}{\lambda_{\min}(Q)}\left(\alpha_0 \frac{2\lambda_{\max}(Q)}{\kappa} + \alpha_0^2\right),
\end{aligned}
$$

*where*

$$
K_1 := 4\Xi + 13C_1 E_1 + 20C_1 C_2 + 4E_1 C_2, \quad K_2 := 25C_1 C_2 + 10C_2^2 + 2\Xi + 4E_1 C_2.
$$

*2. Considering diminishing step-size, i.e., $\alpha_t = \frac{h_1}{t + h_2}$ for $t \in \mathbb{N}$ such that*
$\max\left\{\frac{\tau - 1 + 2^{1/E_1 h_1}}{2^{1/E_1 h_1} - 1}, 32\tau E_1 h_1, 32\tau C_2 h_1, \frac{\Xi h_1}{2C_1}, h_1 \frac{2\|Q\|_2 \lambda_{\max}(Q)(2E_1^2 + 2L_1\tau)}{\kappa \lambda_{\min}(Q)}\right\} \leq h_2$ *and*
$\max\left\{\frac{4\lambda_{\max}(Q)}{\kappa}, \frac{2}{E_1}\right\} \leq h_1$, *for $\tau \leq k \leq T$, we have*

$$
\begin{aligned}
\mathbb{E}\left[\|z_{k+1}\|_2^2\right] &\leq \frac{\lambda_{\max}(Q)}{\lambda_{\min}(Q)}\left(\frac{\tau + h_2}{k + h_2}\right)^{h_1 \frac{\kappa}{2\lambda_{\max}(Q)}}(2\|z_0\|_2 + 4C_2\tau\alpha_0) \\
&\quad + \frac{1}{\lambda_{\min}(Q)}\frac{16\|Q\|_2(L_2\tau + C_2^2)h_1^2}{k - 1 + h_2}\frac{2^{h_1 \frac{\kappa}{2\lambda_{\max}(Q)} - 1}}{h_1 \frac{\kappa}{2\lambda_{\max}(Q)} - 1} \\
&\quad + \frac{2\|Q\|_2}{\lambda_{\min}(Q)}\left(L_2\tau\alpha_{k-\tau}\alpha_k + C_2^2\alpha_k^2\right),
\end{aligned}
$$

*where*

$$
L_1 := 4\Xi + 13E_1 C_1 + 8C_1 C_2 + 4C_2 E_1, \quad L_2 := 13C_1 C_2 + 4C_2^2 + 2\Xi + 4C_2 E_1.
$$

*Proof.* Let $V(z) = z^\top Q z$ for $z \in \mathbb{R}^{2Nq}$. From simple algebraic manipulation in Srikant and Ying (2019), we have the following decomposition:

$$
\begin{aligned}
&\mathbb{E}\left[V(z_{k+1}) - V(z_k)\right] \\
&= \mathbb{E}\left[(z_{k+1} - z_k)^\top Q(z_{k+1} - z_k)\right] + \mathbb{E}\left[2z_k^\top Q z_{k+1}\right] - 2\mathbb{E}\left[V(z_k)\right] \\
&= \mathbb{E}\left[(z_{k+1} - z_k)^\top Q(z_{k+1} - z_k)\right] + \mathbb{E}\left[2z_k^\top Q(z_{k+1} - z_k)\right] \\
&= \underbrace{\mathbb{E}\left[(z_{k+1} - z_k)^\top Q(z_{k+1} - z_k)\right]}_{I_1} + \underbrace{\mathbb{E}\left[2z_k^\top Q(z_{k+1} - z_k - \alpha_k E z_k)\right]}_{I_2} + \underbrace{2\alpha_k \mathbb{E}\left[z_k^\top Q E z_k\right]}_{I_3}.
\end{aligned}
\tag{35}
$$

1. We will first consider the case using constant step-size. The term $I_1$ can be bounded using Lemma A.18, the term $I_2$ can be bounded using the first item in Lemma A.17, and the bound on $I_3$ follows from the third item in Assumption A.14, which yields

$$
\begin{aligned}
\mathbb{E}[V(z_{k+1}) - V(z_k)] &\leq 2\alpha_0^2 \|Q\|_2\left(E_1^2 \mathbb{E}\left[\|z_k\|_2^2\right] + C_2^2\right) \\
&\quad + 2\alpha_0 \|Q\|_2 K_1 \alpha_0 \tau \mathbb{E}\left[\|z_k\|_2^2\right] \\
&\quad + 2\alpha_0 \|Q\|_2 K_2 \tau\alpha_0 \\
&\quad - \alpha_0 \kappa \mathbb{E}\left[\|z_k\|_2^2\right].
\end{aligned}
$$

Considering that $\lambda_{\min}(\boldsymbol{Q}) \|\boldsymbol{z}_k\|_2^2 \leq \|\boldsymbol{z}_k\|_{\boldsymbol{Q}}^2 \leq \lambda_{\max}(\boldsymbol{Q}) \|\boldsymbol{z}_k\|_2^2$, we get

$$\mathbb{E}\left[V(\boldsymbol{z}_{k+1}) - V(\boldsymbol{z}_k)\right] \leq \frac{\|\boldsymbol{Q}\|_2}{\lambda_{\min}(\boldsymbol{Q})} \left(2E_1^2\alpha_0^2 + 2K_1\alpha_0^2\tau\right) \mathbb{E}\left[V(\boldsymbol{z}_k)\right] - \alpha_0 \frac{\kappa}{\lambda_{\max}(\boldsymbol{Q})} \mathbb{E}\left[V(\boldsymbol{z}_k)\right]$$
$$+ 2\|\boldsymbol{Q}\|_2 (C_2^2 + K_2\tau)\alpha_0^2.$$

The condition on the step-size that

$$\frac{\|\boldsymbol{Q}\|_2}{\lambda_{\min}(\boldsymbol{Q})} \left(2E_1^2\alpha_0^2 + 2K_1\alpha_0^2\tau\right) - \alpha_0 \frac{\kappa}{\lambda_{\max}(\boldsymbol{Q})} \leq -\alpha_0 \frac{\kappa}{2\lambda_{\max}(\boldsymbol{Q})}$$
$$\Longleftrightarrow \alpha_0 \leq \frac{\kappa\lambda_{\min}(\boldsymbol{Q})}{(4E_1^2 + 4K_1\tau)\lambda_{\max}(\boldsymbol{Q}) \|\boldsymbol{Q}\|_2},$$

leads to

$$\mathbb{E}\left[V(\boldsymbol{z}_{k+1})\right] \leq \left(1 - \alpha_0 \frac{\kappa}{2\lambda_{\max}(\boldsymbol{Q})}\right) \mathbb{E}\left[V(\boldsymbol{z}_k)\right] + 2\|\boldsymbol{Q}\|_2 (C_2^2 + K_2\tau)\alpha_0^2.$$

Recursively expanding the terms, we get

$$\mathbb{E}\left[V(\boldsymbol{z}_{k+1})\right]$$
$$\leq \Pi_{i=\tau}^{k} \left(1 - \alpha_0 \frac{\kappa}{2\lambda_{\max}(\boldsymbol{Q})}\right) \mathbb{E}\left[V(\boldsymbol{z}_\tau)\right]$$
$$+ \sum_{i=\tau}^{k-1} 2\|\boldsymbol{Q}\|_2 (C_2^2 + K_2\tau)\alpha_0^2 \Pi_{j=i+1}^{k} \left(1 - \alpha_0 \frac{\kappa}{2\lambda_{\max}(\boldsymbol{Q})}\right) + 2\|\boldsymbol{Q}\|_2 (C_2^2 + K_2\tau)\alpha_0^2$$
$$\leq \exp\left(-\alpha_0 \frac{\kappa}{2\lambda_{\max}(\boldsymbol{Q})}(k - \tau + 1)\right) \mathbb{E}\left[V(\boldsymbol{z}_\tau)\right]$$
$$+ \sum_{i=\tau}^{k-1} 2\|\boldsymbol{Q}\|_2 (C_2^2 + K_2\tau)\alpha_0^2 \exp\left(-\alpha_0 \frac{\kappa}{2\lambda_{\max}(\boldsymbol{Q})}(k - i)\right) + 2\|\boldsymbol{Q}\|_2 (C_2^2 + K_2\tau)\alpha_0^2$$
$$\leq \exp\left(-\alpha_0 \frac{\kappa}{2\lambda_{\max}(\boldsymbol{Q})}(k - \tau + 1)\right) \mathbb{E}\left[V(\boldsymbol{z}_\tau)\right]$$
$$+ 2\|\boldsymbol{Q}\|_2 (C_2^2 + K_2\tau)\alpha_0^2 \frac{\exp\left(-\alpha_0 \frac{\kappa}{2\lambda_{\max}(\boldsymbol{Q})}\right)}{1 - \exp\left(-\alpha_0 \frac{\kappa}{2\lambda_{\max}(\boldsymbol{Q})}\right)} + 2\|\boldsymbol{Q}\|_2 (C_2^2 + K_2\tau)\alpha_0^2$$
$$= \exp\left(-\alpha_0 \frac{\kappa}{2\lambda_{\max}(\boldsymbol{Q})}(k - \tau + 1)\right) \mathbb{E}\left[V(\boldsymbol{z}_\tau)\right]$$
$$+ 2\|\boldsymbol{Q}\|_2 (C_2^2 + K_2\tau)\alpha_0^2 \frac{1}{\exp\left(\alpha_0 \frac{\kappa}{2\lambda_{\max}(\boldsymbol{Q})}\right) - 1} + 2\|\boldsymbol{Q}\|_2 (C_2^2 + K_2\tau)\alpha_0^2$$
$$\leq \exp\left(-\alpha_0 \frac{\kappa}{2\lambda_{\max}(\boldsymbol{Q})}(k - \tau + 1)\right) \mathbb{E}\left[V(\boldsymbol{z}_\tau)\right]$$
$$+ 2\|\boldsymbol{Q}\|_2 (C_2^2 + K_2\tau)\alpha_0^2 \frac{2\lambda_{\max}(\boldsymbol{Q})}{\alpha_0\kappa} + 2\|\boldsymbol{Q}\|_2 (C_2^2 + K_2\tau)\alpha_0^2$$
$$= \exp\left(-\alpha_0 \frac{\kappa}{2\lambda_{\max}(\boldsymbol{Q})}(k - \tau + 1)\right) \mathbb{E}\left[V(\boldsymbol{z}_\tau)\right]$$
$$+ 4\|\boldsymbol{Q}\|_2 (C_2^2 + K_2\tau) \alpha_0 \frac{\lambda_{\max}(\boldsymbol{Q})}{\kappa} + 2\|\boldsymbol{Q}\|_2 (C_2^2 + K_2\tau)\alpha_0^2.$$

The second inequality follows from the fact that $1 - x \leq \exp(-x)$ for $x \in \mathbb{R}$. From the first item in Lemma A.15, we can bound $\mathbb{E}\left[V(\boldsymbol{z}_\tau)\right]$, which leads to

$$\mathbb{E}\left[\left\|\boldsymbol{z}_{k+1}\right\|_2^2\right] \leq \frac{\lambda_{\max}(\boldsymbol{Q})}{\lambda_{\min}(\boldsymbol{Q})} \exp\left(-\alpha_0 \frac{\kappa}{2\lambda_{\max}(\boldsymbol{Q})}(k-\tau+1)\right)\left(2\left\|\boldsymbol{z}_0\right\|_2^2 + \frac{4C_2}{E_1}\right)$$
$$+ \frac{2\left\|\boldsymbol{Q}\right\|_2\left(C_2^2 + K_2\tau\right)}{\lambda_{\min}(\boldsymbol{Q})}\left(\alpha_0 \frac{2\lambda_{\max}(\boldsymbol{Q})}{\kappa} + \alpha_0^2\right).$$

2. We will now consider the case using diminishing step-size. In (35), the term $I_1$ can be bounded using Lemma A.18, the term $I_2$ can be bounded using the second item in Lemma A.17, and the bound on $I_3$ follows from the third item in Assumption A.14, which yields

$$\mathbb{E}[V(\boldsymbol{z}_{k+1}) - V(\boldsymbol{z}_k)] \leq 2\left\|\boldsymbol{Q}\right\|_2 \alpha_k^2 \left(E_1^2 \mathbb{E}\left[\left\|\boldsymbol{z}_k\right\|_2^2\right] + C_2^2\right)$$
$$+ 2\alpha_k \left\|\boldsymbol{Q}\right\|_2 L_1 \alpha_{k-\tau}\tau \mathbb{E}\left[\left\|\boldsymbol{z}_k\right\|_2^2\right]$$
$$+ 2\alpha_k \left\|\boldsymbol{Q}\right\|_2 L_2 \alpha_{k-\tau}\tau$$
$$- \alpha_k \kappa \mathbb{E}\left[\left\|\boldsymbol{z}_k\right\|_2^2\right],$$

where

$$L_1 := 4\Xi + 13E_1C_1 + 8C_1C_2 + 4C_2E_1, \quad L_2 := 13C_1C_2 + 4C_2^2 + 2\Xi + 4C_2E_1.$$

Considering that $\lambda_{\min}(\boldsymbol{Q})\left\|\boldsymbol{z}_k\right\|_2^2 \leq \left\|\boldsymbol{z}_k\right\|_{\boldsymbol{Q}}^2 \leq \lambda_{\max}(\boldsymbol{Q})\left\|\boldsymbol{z}_k\right\|_2^2$, we get

$$\mathbb{E}\left[V(\boldsymbol{z}_{k+1}) - V(\boldsymbol{z}_k)\right] \leq \frac{\left\|\boldsymbol{Q}\right\|_2}{\lambda_{\min}(\boldsymbol{Q})}\left(2E_1^2\alpha_k^2 + 2L_1\alpha_k\alpha_{k-\tau}\tau\right)\mathbb{E}\left[V(\boldsymbol{z}_k)\right] - \alpha_k \frac{\kappa}{\lambda_{\max}(\boldsymbol{Q})}\mathbb{E}\left[V(\boldsymbol{z}_k)\right]$$
$$\tag{36}$$
$$+ 2\left\|\boldsymbol{Q}\right\|_2 C_2^2\alpha_k^2 + 2\left\|\boldsymbol{Q}\right\|_2 L_2\tau\alpha_k\alpha_{k-\tau}.$$

The condition on the step-size that $h_2 \geq h_1 \frac{2\|\boldsymbol{Q}\|_2\lambda_{\max}(\boldsymbol{Q})(2E_1^2+2L_1\tau)}{\kappa\lambda_{\min}(\boldsymbol{Q})}$ implies

$$\frac{\left\|\boldsymbol{Q}\right\|_2}{\lambda_{\min}(\boldsymbol{Q})}\left(2E_1^2\alpha_k^2 + 2L_1\alpha_k\alpha_{k-\tau}\tau\right) - \alpha_k\frac{\kappa}{\lambda_{\max}(\boldsymbol{Q})} \leq -\alpha_k\frac{\kappa}{2\lambda_{\max}(\boldsymbol{Q})}.$$

Applying the above relation to (36) results to

$$\mathbb{E}\left[V(\boldsymbol{z}_{k+1})\right] \leq \mathbb{E}\left[\left(1 - \frac{\kappa}{2\lambda_{\max}(\boldsymbol{Q})}\alpha_k\right)V(\boldsymbol{z}_k)\right] + 2\left\|\boldsymbol{Q}\right\|_2 C_2^2\alpha_k^2 + 2\left\|\boldsymbol{Q}\right\|_2 L_2\tau\alpha_k\alpha_{k-\tau}.$$

Recursively expanding the terms, we get

$$\mathbb{E}\left[V(\boldsymbol{z}_{k+1})\right]$$

$$\leq \left(1 - \frac{\kappa}{2\lambda_{\max}(\boldsymbol{Q})}\alpha_k\right)\mathbb{E}\left[V(\boldsymbol{z}_k)\right] + 2\left\|\boldsymbol{Q}\right\|_2 L_2\tau\alpha_{k-\tau}\alpha_k + 2\left\|\boldsymbol{Q}\right\|_2 C_2^2\alpha_k^2$$

$$\leq \Pi_{i=\tau}^k\left(1 - \frac{\kappa}{2\lambda_{\max}(\boldsymbol{Q})}\alpha_i\right)\mathbb{E}\left[V(\boldsymbol{z}_\tau)\right] + 2\left\|\boldsymbol{Q}\right\|_2\sum_{i=\tau}^{k-1}(L_2\tau\alpha_{i-\tau}\alpha_i + \alpha_i^2 C_2^2)\Pi_{j=i+1}^{k-1}\left(1 - \frac{\kappa}{2\lambda_{\max}(\boldsymbol{Q})}\alpha_j\right)$$

$$+ 2\left\|\boldsymbol{Q}\right\|_2 L_2\tau\alpha_{k-\tau}\alpha_k + 2\left\|\boldsymbol{Q}\right\|_2 C_2^2\alpha_k^2$$

$$\leq \exp\left(-\frac{\kappa}{2\lambda_{\max}(\boldsymbol{Q})}\sum_{i=\tau}^k\alpha_i\right)\mathbb{E}\left[V(\boldsymbol{z}_\tau)\right] + 2\left\|\boldsymbol{Q}\right\|_2\sum_{i=\tau}^{k-1}(L_2\tau\alpha_{i-\tau}\alpha_i + \alpha_i^2 C_2^2)\exp\left(-\frac{\kappa}{2\lambda_{\max}(\boldsymbol{Q})}\sum_{j=i+1}^{k-1}\alpha_j\right)$$

$$+ 2\left\|\boldsymbol{Q}\right\|_2 L_2\tau\alpha_{k-\tau}\alpha_k + 2\left\|\boldsymbol{Q}\right\|_2 C_2^2\alpha_k^2$$

$$\leq \left(\frac{\tau+h_2}{k+h_2}\right)^{h_1\frac{\kappa}{2\lambda_{\max}(\boldsymbol{Q})}}\mathbb{E}\left[V(\boldsymbol{z}_\tau)\right] + 2\left\|\boldsymbol{Q}\right\|_2\sum_{i=\tau}^{k-1}(L_2\tau\alpha_{i-\tau}\alpha_i + \alpha_i^2 C_2^2)\left(\frac{i+1+h_2}{k-1+h_2}\right)^{h_1\frac{\kappa}{2\lambda_{\max}(\boldsymbol{Q})}}$$

$$+ 2\left\|\boldsymbol{Q}\right\|_2 L_2\tau\alpha_{k-\tau}\alpha_k + 2\left\|\boldsymbol{Q}\right\|_2 C_2^2\alpha_k^2$$

$$\leq \left(\frac{\tau+h_2}{k+h_2}\right)^{h_1\frac{\kappa}{2\lambda_{\max}(\boldsymbol{Q})}}\mathbb{E}\left[V(\boldsymbol{z}_\tau)\right] + \frac{2\left\|\boldsymbol{Q}\right\|_2(L_2\tau + C_2^2)h_1^2}{(k-1+h_2)^{h_1\frac{\kappa}{2\lambda_{\max}(\boldsymbol{Q})}}}\sum_{i=\tau}^{k-1}8\left(i+1+h_2\right)^{h_1\frac{\kappa}{2\lambda_{\max}(\boldsymbol{Q})}-2} \tag{37}$$

$$+ 2\left\|\boldsymbol{Q}\right\|_2 L_2\tau\alpha_{k-\tau}\alpha_k + 2\left\|\boldsymbol{Q}\right\|_2 C_2^2\alpha_k^2$$

$$\leq \left(\frac{\tau+h_2}{k+h_2}\right)^{h_1\frac{\kappa}{2\lambda_{\max}(\boldsymbol{Q})}}\mathbb{E}\left[V(\boldsymbol{z}_\tau)\right] + \frac{16\left\|\boldsymbol{Q}\right\|_2(L_2\tau + C_2^2)h_1^2}{(k-1+h_2)^{h_1\frac{\kappa}{2\lambda_{\max}(\boldsymbol{Q})}}}\int_\tau^k\left(t+1+h_2\right)^{h_1\frac{\kappa}{2\lambda_{\max}(\boldsymbol{Q})}-2}dt \tag{38}$$

$$+ 2\left\|\boldsymbol{Q}\right\|_2 L_2\tau\alpha_{k-\tau}\alpha_k + 2\left\|\boldsymbol{Q}\right\|_2 C_2^2\alpha_k^2$$

$$\leq \left(\frac{\tau+h_2}{k+h_2}\right)^{h_1\frac{\kappa}{2\lambda_{\max}(\boldsymbol{Q})}}\mathbb{E}\left[V(\boldsymbol{x}_\tau)\right] + \frac{16\left\|\boldsymbol{Q}\right\|_2(L_2\tau + C_2^2)h_1^2}{(k-1+h_2)^{h_1\frac{\kappa}{2\lambda_{\max}(\boldsymbol{Q})}}}\frac{1}{h_1\frac{\kappa}{2\lambda_{\max}(\boldsymbol{Q})}-1}(k+1+h_2)^{h_1\frac{\kappa}{2\lambda_{\max}(\boldsymbol{Q})}-1}$$

$$+ 2\left\|\boldsymbol{Q}\right\|_2 L_2\tau\alpha_{k-\tau}\alpha_k + 2\left\|\boldsymbol{Q}\right\|_2 C_2^2\alpha_k^2$$

$$\leq \left(\frac{\tau+h_2}{k+h_2}\right)^{h_1\frac{\kappa}{2\lambda_{\max}(\boldsymbol{Q})}}\mathbb{E}\left[V(\boldsymbol{z}_\tau)\right] + \frac{16\left\|\boldsymbol{Q}\right\|_2(L_2\tau + C_2^2)h_1^2}{k-1+h_2}\frac{2^{h_1\frac{\kappa}{2\lambda_{\max}(\boldsymbol{Q})}-1}}{h_1\frac{\kappa}{2\lambda_{\max}(\boldsymbol{Q})}-1} \tag{39}$$

$$+ 2\left\|\boldsymbol{Q}\right\|_2 L_2\tau\alpha_{k-\tau}\alpha_k + 2\left\|\boldsymbol{Q}\right\|_2 C_2^2\alpha_k^2.$$

The inequality (37) follows from the fact that $\alpha_{i-\tau} \leq 2\alpha_i$ for $\tau \leq i$, which holds since $\frac{\tau-1+2^{1/E_1h_1}}{2^{1/E_1h_1}-1} \leq h_2$ and $2 \leq E_1h_1$. Moreover, $\frac{i+1+h_2}{i+h_2} \leq 2$ for $i \geq 0$. This follows from the condition $h_2 > 2\tau$, which can be checked from $h_2 \geq \frac{\tau-1+2^{1/E_1h_1}}{2^{1/E_1h_1}-1}$ and $h_1E_1 \geq 2$. The inequality (38) holds since $\frac{4\lambda_{\max}(\boldsymbol{Q})}{\kappa} \leq h_1$. The inequality (39) follows from the fact that $k+1+h_2 \leq 2k-2+2h_2$, which when holds $h_2 \geq 3$ and it is satisfied by the inequalities $h_2 \geq \frac{\tau-1+2^{1/E_1h_1}}{2^{1/E_1h_1}-1}$ and $h_1E_1 \geq 2$. We can bound $\mathbb{E}\left[V(\boldsymbol{z}_\tau)\right]$ from Lemma A.15, which results to

$$\mathbb{E}\left[\left\|\boldsymbol{z}_{k+1}\right\|_2^2\right] \leq \frac{\lambda_{\max}(\boldsymbol{Q})}{\lambda_{\min}(\boldsymbol{Q})}\left(\frac{\tau+h_2}{k+h_2}\right)^{h_1\frac{\kappa}{2\lambda_{\max}(\boldsymbol{Q})}}(2\left\|\boldsymbol{z}_0\right\|_2 + 4C_2\tau\alpha_0)$$

$$+ \frac{1}{\lambda_{\min}(\boldsymbol{Q})}\frac{16\left\|\boldsymbol{Q}\right\|_2(L_2\tau + C_2^2)h_1^2}{k-1+h_2}\frac{2^{h_1\frac{\kappa}{2\lambda_{\max}(\boldsymbol{Q})}-1}}{h_1\frac{\kappa}{2\lambda_{\max}(\boldsymbol{Q})}-1}$$

$$+ \frac{2\left\|\boldsymbol{Q}\right\|_2}{\lambda_{\min}(\boldsymbol{Q})}\left(L_2\tau\alpha_{k-\tau}\alpha_k + C_2^2\alpha_k^2\right).$$

This completes the proof.

$\square$

### A.11 PROOF OF THEOREM 4.3

We will provide several building blocks for the main proof. First, for $o \in \mathcal{S} \times \mathcal{S} \times \Pi_{i=1}^N I$, let

$$\boldsymbol{w}(o) := \begin{bmatrix} r^1(s, \boldsymbol{a}, s')\boldsymbol{\phi}(s) - \boldsymbol{b}^1 \\ r^2(s, \boldsymbol{a}, s')\boldsymbol{\phi}(s) - \boldsymbol{b}^2 \\ \vdots \\ r^N(s, \boldsymbol{a}, s')\boldsymbol{\phi}(s) - \boldsymbol{b}^N \end{bmatrix} + \left( \boldsymbol{I}_q \otimes (\gamma\boldsymbol{\phi}(s)\boldsymbol{\phi}^\top(s') - \boldsymbol{\phi}(s)\boldsymbol{\phi}(s)^\top) - \bar{\boldsymbol{A}} \right) \boldsymbol{1}_N \otimes \boldsymbol{\theta}_c,$$

$$\boldsymbol{W}(o) := \boldsymbol{I}_q \otimes (\gamma\boldsymbol{\phi}(s)\boldsymbol{\phi}^\top(s') - \boldsymbol{\phi}(s)\boldsymbol{\phi}(s)^\top) - \bar{\boldsymbol{A}}.$$

Note that $\bar{\boldsymbol{\epsilon}}(o_k; \bar{\boldsymbol{\theta}}_k)$ defined in (10) can be expressed as

$$\bar{\boldsymbol{\epsilon}}(o_k; \bar{\boldsymbol{\theta}}_k) = \begin{bmatrix} \boldsymbol{W}(o_k)\tilde{\boldsymbol{\theta}}_k + \boldsymbol{w}(o_k) \\ \boldsymbol{0}_{Nq} \end{bmatrix}.$$

**Lemma A.20.** *For $o \in \mathcal{S} \times \mathcal{S} \times \Pi_{i=1}^N I$, we have*

$$\|\boldsymbol{W}(o)\|_2 \le 6, \quad \|\boldsymbol{w}(o)\|_2 \le \frac{9\sqrt{N}R_{\max}}{(1-\gamma)w}.$$

*Proof.* First, we have

$$\begin{aligned} \|\boldsymbol{W}(o)\| &= \left\| \boldsymbol{I}_q \otimes (\gamma\boldsymbol{\phi}(s)\boldsymbol{\phi}^\top(s') - \boldsymbol{\phi}(s)\boldsymbol{\phi}(s)^\top) - \bar{\boldsymbol{A}} \right\|_2 \\ &\le \left\| \gamma\boldsymbol{\phi}(s)\boldsymbol{\phi}^\top(s') - \boldsymbol{\phi}(s)\boldsymbol{\phi}(s)^\top - \boldsymbol{A} \right\|_2 \\ &\le 6. \end{aligned}$$

The last inequality follows from Lemma A.7 and the assumption that $\|\boldsymbol{\phi}(s)\|_2 \le 1$ for all $s \in \mathcal{S}$.
Moreover, we have

$$\begin{aligned} \|\boldsymbol{w}(o)\|_2 &= \|\bar{\boldsymbol{\epsilon}}(o; \boldsymbol{1}_N \otimes \boldsymbol{\theta}_c)\|_2 \\ &\le 6 \|\boldsymbol{1}_N \otimes \boldsymbol{\theta}_c\|_2 + 3\sqrt{N}R_{\max} \\ &\le 6\sqrt{N}\frac{R_{\max}}{(1-\gamma)w} + 3\sqrt{N}R_{\max} \\ &\le \frac{9\sqrt{N}R_{\max}}{(1-\gamma)w}, \end{aligned}$$

where the first equality follows from the definition of $\bar{\boldsymbol{\epsilon}}$ in (10). The first inequality follows from (17). The second inequality follows from Lemma A.6. $\square$

**Lemma A.21.** *For $k \ge \tau$, we have*

$$\left\| \mathbb{E}\left[ \bar{\boldsymbol{\epsilon}}(o_k; \bar{\boldsymbol{\theta}}_{k-\tau}) \middle| \mathcal{F}_{k-\tau} \right] \right\|_2 \le \max \left\{ \frac{4R_{\max}\sqrt{Nq}}{w(1-\gamma)}, 2q \right\} \alpha_T (\|\bar{\boldsymbol{\theta}}_{k-\tau} - \boldsymbol{1}_N \otimes \boldsymbol{\theta}_c\|_2 + 1).$$

*Proof.* Applying triangle inequality to (10), we get

$$\left\| \mathbb{E}\left[ \bar{\boldsymbol{\epsilon}}(o_k; \bar{\boldsymbol{\theta}}_{k-\tau}) \middle| \mathcal{F}_{k-\tau} \right] \right\|_2 \le \underbrace{\left\| \mathbb{E}\left[ \boldsymbol{w}(o_k) \middle| \mathcal{F}_{k-\tau} \right] \right\|_2}_{I_1} + \underbrace{\left\| \mathbb{E}\left[ \boldsymbol{W}(o_k)(\bar{\boldsymbol{\theta}}_{k-\tau} - \boldsymbol{1}_N \otimes \boldsymbol{\theta}_c) \middle| \mathcal{F}_{k-\tau} \right] \right\|_2}_{I_2}.$$

We will check the conditions to apply Lemma A.5 to bound $I_1$ and $I_2$ separately. Considering $I_1$, note that we have

$$\left\| \begin{bmatrix} r_k^1 \phi(s_k) \\ r_k^2 \phi(s_k) \\ \vdots \\ r_k^N \phi(s_k) \end{bmatrix} + \left( \boldsymbol{I}_q \otimes (\gamma \phi(s_k)\phi^\top(s_k') - \phi(s_k)\phi(s_k)^\top) \right) \boldsymbol{1}_N \otimes \boldsymbol{\theta}_c \right\|_\infty$$

$$\leq \max_{1 \leq i \leq N} \left\| r_k^i \phi(s_k) \right\|_\infty + \left\| \left( \boldsymbol{I}_q \otimes (\gamma \phi(s_k)\phi^\top(s_k') - \phi(s_k)\phi(s_k)^\top) \right) \boldsymbol{1}_N \otimes \boldsymbol{\theta}_c \right\|_\infty$$

$$\leq R_{\max} + \left\| \boldsymbol{I}_q \otimes (\gamma \phi(s_k)\phi^\top(s_k') - \phi(s_k)\phi(s_k)^\top) \right\|_2 \left\| \boldsymbol{1}_N \otimes \boldsymbol{\theta}_c \right\|_\infty$$

$$\leq R_{\max} + \frac{2R_{\max}}{(1-\gamma)w}$$

$$\leq \frac{4R_{\max}}{w(1-\gamma)}.$$

The second inequality follows from the assumption that $|r_k^i| \leq R_{\max}$ for $1 \leq i \leq N, k \in \mathbb{N}_0$, and $\|\phi(s)\|_2 \leq 1$ for $s \in \mathcal{S}$. The third inequality follows from Lemma A.6.

Furthermore, we have, for $1 \leq i \leq N$,

$$\mathbb{E}\left[ r_k^i \phi(s_k) \right] = \sum_{s \in \mathcal{S}} d(s)\phi(s) \sum_{s' \in \mathcal{S}} \sum_{\boldsymbol{a} \in \Pi_{i=1}^N \mathcal{A}^i} \pi(\boldsymbol{a}|s) \mathcal{P}(s, \boldsymbol{a}, s') r^i(s, \boldsymbol{a}, s')$$

$$= \sum_{s \in \mathcal{S}} d(s)\phi(s) [\boldsymbol{R}_i^\pi]_s$$

$$= \boldsymbol{\Phi}^\top \boldsymbol{D}^\pi \boldsymbol{R}_i^\pi,$$

and it is straightforward to check that $\mathbb{E}\left[ \phi(s_k)\phi^\top(s_k') - \phi(s_k)\phi^\top(s_k)) \right] = \boldsymbol{A}$. Therefore, from Lemma A.5, we get

$$\left\| \mathbb{E}\left[ \boldsymbol{w}(o_k)|\mathcal{F}_{k-\tau} \right] \right\|_2 \leq \frac{4R_{\max}\sqrt{Nq}}{w(1-\gamma)} \alpha_T.$$

Now, we will bound $I_2$. Consider the following relations:

$$\left\| \mathbb{E}\left[ \boldsymbol{W}(o_k)|\mathcal{F}_{k-\tau} \right] \right\|_2 = \left\| \mathbb{E}\left[ \gamma \phi(s_k)\phi^\top(s_k') - \phi(s_k)\phi^\top(s_k) - \boldsymbol{A} \big| \mathcal{F}_{k-\tau} \right] \right\|_2$$

and

$$\max_{1 \leq i,j \leq q} \left| \left[ (\gamma \phi(s_k)\phi^\top(s_k') - \phi(s_k)\phi(s_k)^\top) \right]_{ij} \right| \leq \left\| \gamma \phi(s_k)\phi^\top(s_k') - \phi(s_k)\phi^\top(s_k)) \right\|_2 \leq 2,$$

where the second inequality follows from the assumption that $\|\phi(s)\|_2 \leq 1$ for $s \in \mathcal{S}$.

From the third item in Lemma A.5, we have

$$\left\| \mathbb{E}\left[ \boldsymbol{W}(o_k)|\mathcal{F}_{k-\tau} \right] \right\|_2 \leq 2q\alpha_T.$$

Hence, we have

$$\left\| \mathbb{E}\left[ \boldsymbol{W}(o_k)(\bar{\boldsymbol{\theta}}_{k-\tau} - \boldsymbol{1}_N \otimes \boldsymbol{\theta}_c)\big|\mathcal{F}_{k-\tau} \right] \right\|_2 = \left\| \mathbb{E}\left[ \boldsymbol{W}(o_k)|\mathcal{F}_{k-\tau} \right] (\bar{\boldsymbol{\theta}}_{k-\tau} - \boldsymbol{1}_N \otimes \boldsymbol{\theta}_c) \right\|_2$$

$$\leq \left\| \mathbb{E}\left[ \boldsymbol{W}(o_k)|\mathcal{F}_{k-\tau} \right] \right\|_2 \left\| \bar{\boldsymbol{\theta}}_{k-\tau} - \boldsymbol{1}_N \otimes \boldsymbol{\theta}_c \right\|_2$$

$$\leq 2q\alpha_T \left\| \bar{\boldsymbol{\theta}}_{k-\tau} - \boldsymbol{1}_N \otimes \boldsymbol{\theta}_c \right\|_2.$$

Collecting the bounds on $I_1$ and $I_2$, we get

$$\left\| \mathbb{E}\left[ \bar{\boldsymbol{\epsilon}}(o_k; \bar{\boldsymbol{\theta}}_{k-\tau})\big|\mathcal{F}_{k-\tau} \right] \right\|_2 \leq \frac{4R_{\max}\sqrt{Nq}}{w(1-\gamma)} \alpha_T + 2q\alpha_T \left\| \bar{\boldsymbol{\theta}}_{k-\tau} - \boldsymbol{1}_N \otimes \boldsymbol{\theta}_c \right\|_2$$

$$\leq \max\left\{ \frac{4R_{\max}\sqrt{Nq}}{w(1-\gamma)}, 2q \right\} \alpha_T (\left\| \bar{\boldsymbol{\theta}}_{k-\tau} - \boldsymbol{1}_N \otimes \boldsymbol{\theta}_c \right\|_2 + 1).$$

This completes the proof. $\qquad\square$

Now, we are ready to prove Theorem 4.3.

*Proof of Theorem 4.3.* To this end, we will apply Theorem A.19 in the Appendix Section A.10. Let $z_k := \begin{bmatrix} \tilde{\boldsymbol{\theta}}_k \\ \bar{\boldsymbol{L}}\bar{\boldsymbol{L}}^\dagger \tilde{\boldsymbol{w}}_k \end{bmatrix}$. Hence, it is enough to check the conditions in Assumption A.14 in the Appendix Section A.10. The first item in Assumption A.14 can be checked from Lemma A.20, we have

$$C_1 := 6, \quad C_2 := \frac{9\sqrt{N}R_{\max}}{w(1-\gamma)}, \quad E_1 := 8 + 2\lambda_{\max}(\bar{\boldsymbol{L}}).$$

From Lemma A.21, we have

$$\Xi := \max\left\{\frac{4R_{\max}\sqrt{Nq}}{w(1-\gamma)}, 2q\right\},$$

which satisfies the second assumption in Assumption A.14. The third item in Assumption A.14 follows from Lemma 4.1.

1. For constant step-size, $K_1$ and $K_2$ in Theorem A.19 becomes

$$K_1 = 4\max\left\{\frac{4R_{\max}\sqrt{Nq}}{w(1-\gamma)}, 2q\right\} + 624 + 152\frac{9\sqrt{N}R_{\max}}{w(1-\gamma)} + 13\cdot 12\lambda_{\max}(\bar{\boldsymbol{L}}) + \frac{72\sqrt{N}R_{\max}}{w(1-\gamma)}\lambda_{\max}(\bar{\boldsymbol{L}})$$

$$= \mathcal{O}\left(\max\left\{\frac{\sqrt{Nq}R_{\max}}{w(1-\gamma)}\lambda_{\max}(\bar{\boldsymbol{L}}), q\right\}\right),$$

$$K_2 = 2\max\left\{\frac{4R_{\max}\sqrt{Nq}}{w(1-\gamma)}, 2q\right\} + 810\frac{NR_{\max}^2}{w^2(1-\gamma)^2} + 182\frac{9\sqrt{N}R_{\max}}{w(1-\gamma)} + \frac{72\sqrt{N}R_{\max}}{w(1-\gamma)}\lambda_{\max}(\bar{\boldsymbol{L}}),$$

which leads to

$$\Omega\left(\frac{NR_{\max}^2}{w^2(1-\gamma)^2}\right) \leq K_2 \leq \mathcal{O}\left(\max\left\{\frac{N\sqrt{q}R_{\max}^2}{w^2(1-\gamma)^2}\lambda_{\max}(\bar{\boldsymbol{L}}), 2q\right\}\right).$$

Note that from Lemma 4.1, we have $\|\boldsymbol{G}\|_2 = \Theta\left(\frac{\lambda_{\max}(\bar{\boldsymbol{L}})^2}{(1-\gamma)w}\right)$. Therefore, from the step-size condition in the first item in Theorem A.19, we need

$$\alpha_0$$
$$\leq \min\left\{\frac{1}{900\tau\max\left\{\frac{\sqrt{N}R_{\max}}{w(1-\gamma)}, 10\lambda_{\max}(\bar{\boldsymbol{L}})\right\}}, \frac{6}{2\max\left\{\frac{4R_{\max}\sqrt{Nq}}{w(1-\gamma)}, 2q\right\}}, \frac{\min\left\{1, \lambda_{\min}^+(\bar{\boldsymbol{L}})^2\right\}\lambda_{\min}(\boldsymbol{G})}{(400\lambda_{\max}(\bar{\boldsymbol{L}})^2 + 4K_1\tau)\lambda_{\max}(\boldsymbol{G})\|\boldsymbol{G}\|_2}\right\}$$

Hence, there exists $\bar{\alpha}$ such that

$$\bar{\alpha} = \mathcal{O}\left(\frac{\min\left\{1, \lambda_{\min}^+(\bar{\boldsymbol{L}})^2\right\}(1-\gamma)w}{\tau\max\left\{\frac{\sqrt{Nq}R_{\max}}{w(1-\gamma)}, q\right\}\lambda_{\max}(\bar{\boldsymbol{L}})^4}\right).$$

Therefore, the first item in Theorem A.19 leads to

$$\frac{1}{N}\left(\mathbb{E}\left[\left\|\tilde{\boldsymbol{\theta}}_{k+1}\right\|_2^2\right] + \left\|\bar{\boldsymbol{L}}\bar{\boldsymbol{L}}^\dagger\tilde{\boldsymbol{w}}_{k+1}\right\|_2^2\right)$$

$$= \mathcal{O}\left(\exp\left(-\frac{(1-\gamma)w\min\{1, \lambda_{\min}^+(\boldsymbol{L})^2\}}{\lambda_{\max}(\boldsymbol{L})^2}\alpha_0(k-\tau-1)\right)\right.$$

$$\left. + \alpha_0\tau\max\left\{\frac{\sqrt{q}R_{\max}^2\lambda_{\max}(\bar{\boldsymbol{L}})}{w^3(1-\gamma)^3}, \frac{2q}{N(1-\gamma)w}\right\}\frac{\lambda_{\max}(\boldsymbol{L})^2}{\min\{1, \lambda_{\min}^+(\boldsymbol{L})^2\}}\right).$$

2. For diminishing step-size, we get

$$L_1 = 4\max\left\{\frac{4R_{\max}\sqrt{Nq}}{w(1-\gamma)}, 2q\right\} + 624 + 80\frac{9\sqrt{N}R_{\max}}{w(1-\gamma)} + 13\cdot 12\lambda_{\max}(\bar{L}) + \frac{72\sqrt{N}R_{\max}}{w(1-\gamma)}\lambda_{\max}(\bar{L})$$

$$= \mathcal{O}\left(\max\left\{\frac{R_{\max}\sqrt{Nq}}{w(1-\gamma)}\lambda_{\max}(\bar{L}), 2q\right\}\right),$$

$$L_2 = 2\max\left\{\frac{4R_{\max}\sqrt{Nq}}{w(1-\gamma)}, 2q\right\} + 110\frac{9\sqrt{N}R_{\max}}{w(1-\gamma)} + 4\frac{81NR_{\max}^2}{w^2(1-\gamma)^2} + \frac{72\sqrt{N}R_{\max}}{w(1-\gamma)}\lambda_{\max}(\bar{L}),$$

which leads to

$$\Omega\left(\frac{NR_{\max}^2}{w^2(1-\gamma)^2}\right) \le L_2 \le \mathcal{O}\left(\max\left\{\frac{N\sqrt{q}R_{\max}^2}{w^2(1-\gamma)^2}\lambda_{\max}(\bar{L}), q\right\}\right).$$

Following the second item in Theorem A.19, the choice of step-size satisfying

$$h_1 = \Theta\left(\frac{\lambda_{\max}(L)^2}{(1-\gamma)w\min\{1, \lambda_{\min}^+(\bar{L})^2\}}\right)$$

$$h_2 = \Theta\left(\max\left\{1 + \frac{\tau}{2^{1/E_1 h_1}-1}, h_1\tau\frac{\sqrt{N}R_{\max}}{(1-\gamma)w}, h_1\frac{\lambda_{\max}(\bar{L})^4\tau}{\min\{1, \lambda_{\min}^+(\bar{L})^2\}}\max\left\{\frac{R_{\max}\sqrt{Nq}}{w^2(1-\gamma)^2}, \frac{2q}{w(1-\gamma)}\right\}\right\}\right),$$

yields

$$\frac{1}{N}\left(\mathbb{E}\left[\left\|\tilde{\boldsymbol{\theta}}_{k+1}\right\|_2^2\right] + \left\|\bar{L}\bar{L}^\dagger\tilde{\boldsymbol{w}}_{k+1}\right\|_2^2\right) = \mathcal{O}\left(\frac{\tau}{k}\frac{qR_{\max}^2}{w^4(1-\gamma)^4}\frac{\lambda_{\max}(L)^5}{\min\{1, \lambda_{\min}^+(L)^2\}^2}\right).$$

This completes the proof.

$\square$

## A.12 EXPERIMENTS

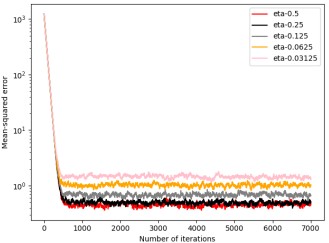 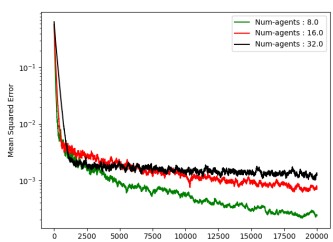

(a) Mean-squared error for Algorithm 1, $N = 16$, $\alpha = 0.1$. When $\eta = 1$, the algorithm diverges.

(b) Mean squared error of Algorithm 1 on star graph after 5000 iterations. We set $\eta = 1$. We omitted the plot when the algorithm diverges.

(c) Mean squared error of Algorithm 1 on star graph with diminishing step-size, $\alpha_k = N^2/(N^3 + k)$, $k \in \mathbb{N}$, where $N$ stands for number of agents.

Figure 2: Additional experiment results of Algorithm 1

Figure (2a) shows the result for different $\eta$ on random graph model explained in Section 5 with number of agents , $N = 16$. When $N = 16$, $\lambda_{\max}(\bar{L}) \approx 14.5$, and $\frac{\sqrt{2}}{\lambda_{\max}(\bar{L})} \approx 0.1$. The algorithm diverges when $\eta = 1$. Moreover, as can be seen in Figure (2a), when $\eta$ is too big or too small compared to $\frac{\sqrt{2}}{\lambda_{\max}(L)}$, the algorithm diverges or shows large bias.

Figure (2b) shows the result for the star graph. For star graph, maximum eigenvalue of the graph Laplacian becomes $2^3, 2^4, 2^5$ for number of agents $2^3, 2^4, 2^5$ respectively. Hence, from Theorem 4.2, the bias term has the tendency to larger as $N$ increases since $\lambda_{\max}(\bar{L})$ scales at the order of $N$, and this can be verified in Figure (2b). Moreover, the convergence result using diminishing step-size can be verified in Figure (2c).

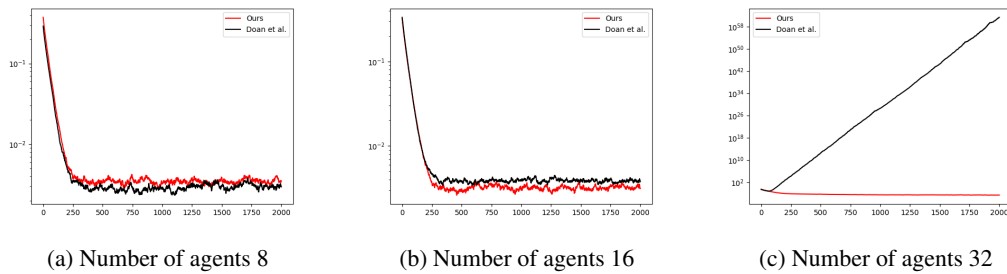

(a) Number of agents 8       (b) Number of agents 16       (c) Number of agents 32

Figure 3: The doubly stochastic matrix was constructed by solving a least squares problem (Bai et al., 2007). We did not plot the result of Wang et al. (2020), since it diverges. The step-size was chosen as $1/2^3$.

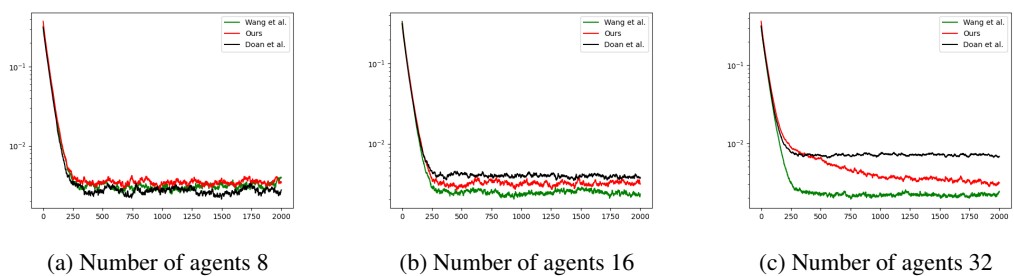

(a) Number of agents 8       (b) Number of agents 16       (c) Number of agents 32

Figure 4: The doubly stochastic matrix was constructed by Sinkhorn-Knobb algorithm (Knight, 2008). The step-size was chosen as $1/2^3$.

### A.13 COMPARISON WITH OTHER ALGORITHMS

To compare the performance with other algorithms, we have experimented under the setting in Section 5, where the results are given in Figure (3) and Figure (4). Note that the performance of distributed TD algorithms in Wang et al. (2020) and Doan et al. (2019) depend on the choice of doubly stochastic matrix. For example, when the doubly stochastic matrix was constructed by least squares method (Bai et al., 2007), there are divergent cases as can be seen from Figure (3).

### A.14 ADDITIONAL RESULTS

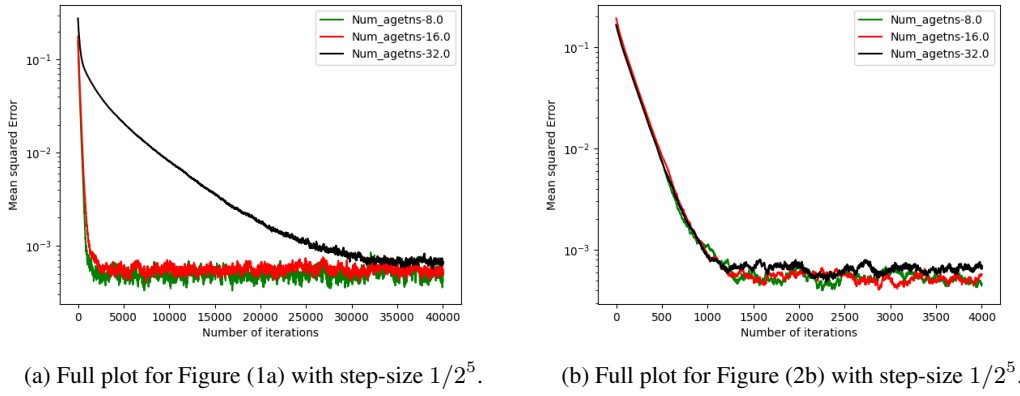

(a) Full plot for Figure (1a) with step-size $1/2^5$.       (b) Full plot for Figure (2b) with step-size $1/2^5$.

Figure 5: Full plots for the result in Figure (1a) and Figure (2b).

