# OpenReview forum: "A primal-dual perspective for distributed TD-learning"
_ICLR.cc/2024/Conference — Submitted to ICLR 2024_

### Official Review · Reviewer_7B2c · 2023-10-31

**Soundness:** 3 good
**Presentation:** 3 good
**Contribution:** 3 good
**Rating:** 8
**Confidence:** 4

**Summary:**

The paper studies the distributed TD learning algorithms for multi-aget MDPs. Using ideas from distributed optimization and control systems, the paper presents a new distributed TD learning algorithm that does not requre of doubly stochastic communication matrix (which is often needed in the existing distributed TD learning algorithms). Finite-time error bounds are developed for the proposed algorithm under both iid and Markovian data models.

**Strengths:**

+ A distributed TD learning algorithm with requiring a doubly stochastic matrix;
+ Convergence rate results for the algorithm under both IID and Markovian data models;
+ Numerical verifications

**Weaknesses:**

- There exist some grammar issues, such as "there exists \bar{h}_1 amd \bar{h}_2..." "the proposed distributed TD-learning do not require". Please check and polish the presentation.

**Questions:**

NA

---

> ### Author Response · Authors · 2023-11-22
> **Response to Reviewer 7B2c**
>
> We are grateful for the reviewer's valuable feedback. We have provided detailed responses to the reviewer's questions and comments.
>
> Thank you for the constructive comments. We will polish the presentation and check the manuscript carefully. Finally, as for clarification, we do not require doubly stochastic matrix and the additional details can be found within the general comments.

---

### Official Review · Reviewer_u6NQ · 2023-10-31

**Soundness:** 3 good
**Presentation:** 2 fair
**Contribution:** 3 good
**Rating:** 6
**Confidence:** 3

**Summary:**

•	This paper analyzes the finite time convergence results for multi-agent distributed TD learning algorithms under a partially connected networking setup. In the paper, they assumed each network agent has local policies and local reward functions, and the goal is to estimate the sum of rewards over all agents with linear local approximation parameters that can be shared with connected neighbors only. In the analysis, they first captured the stochastic algorithm by its continuous-time ODE counterpart, then consider the linear system as the primal-dual gradient dynamics and prove the convergence results through the Lyapunov method applied to the gradient dynamics. The author considered both constant and diminishing step-sizes and both iid and Markovian observation models in their results. Different from previous works, the modified the proof to make the results hold when the graph Laplacian of the associated network graph is not doubly stochastic matrix.

**Strengths:**

•	The strengths come from three parts.
1) The first one is the proof is very sound. Based on that, in this paper, the convergence results are better than existing papers.
2) The second part is that the literature reviews seem to be very detailed, carefully performed, and up to date.
3) The author used a lot of citations throughout the whole paper.

**Weaknesses:**

•	However, there are several weaknesses as far as from my perspective.
•	The first one is I think the paper is not organized very well: the author mentioned several literature many times throughout the whole paper, which feels very tedious; when reading section 3, I was confused since I don’t know the reason of introducing and proving those lemmas until I read section 4, also the notations in section 3 do not closely correspond to the notations used in the rest of papers.
•	The second one is I think the simulation results are too little. In the paper they proved the convergence results for the constant and diminishing step size, but in the simulation section, both figures are for constant step size. I expect seeing diminishing step size case in the main body of the paper.
•	The third one is that the analysis doesn't feel very original. The main difference from the cited existing papers from my understanding are a linear mapping and a multiplication of LL^+. Also, the results rely on the relationship between the smallest and largest eigenvalue of the graph Laplacian, and I don’t know how many network graphs can meet those requirements.
•	There are several other definitions that are not clear, see below.

**Questions:**

•	In the abstract you mentioned your algorithm and method do not require the network structure characterized by a doubly stochastic matrix. But through the whole paper, I didn’t see an introduction to the doubly stochastic matrix and how it is related to communication networks. This is an important contribution of your paper, but I still don’t know what kind of networks correspond to a doubly stochastic matrix and what kind of networks do not. So I don’t know how significant the contribution is.
•	In the proof of theorem 4.2, when using constant step-size, the convergence results rely on the \lambda_max and \lambda_min of the network graph, and the choice of step-size also is based on the \lambda_max. I don’t know how many network graphs can meet the requirements on lambda max and min. Also, you provided an existing bound on lambda_max, so how hard is it to find the lambda_max and how hard is it to find a working eta since you require the eta≈1/ \lambda_max?
•	The citation when you mentioned total variation distance, ergodic and geometric decaying rate is confusing, you may want to cite the original paper that introduced them instead of a recent paper.

---

> ### Author Response · Authors · 2023-11-22
> **Response to Reviewer u6NQ ( Part 1 )**
>
> We are grateful for the reviewer's valuable feedback. We have provided detailed responses to the reviewer's questions and comments.
>
> **1. The first one is I think the paper is not organized very well: the author mentioned several literature many times throughout the whole paper, which feels very tedious; when reading section 3, I was confused since I don’t know the reason of introducing and proving those lemmas until I read section 4, also the notations in section 3 do not closely correspond to the notations used in the rest of papers.**
>
>
> Following the reviewer's comments, we have removed repeated literature citations throughout the manuscript. Moreover, new discussions have been added in Section 3 in order to inform that the results derived in Section 3 will serve as foundation tools for the results of Section 4. Furthermore, the notations in Section~3 have been modified to align more closely to the rest of the paper.
>
> **2. The second one is I think the simulation results are too little. In the paper they proved the convergence results for the constant and diminishing step size, but in the simulation section, both figures are for constant step size. I expect seeing diminishing step size case in the main body of the paper**
>
> In line with the reviewer's comments, we have newly added simulation results with the diminishing step-sizes in Figures (1b) and (2c) of the revised manuscript. Further details on the additional experiments can be found in the ``General Comments'' at the beginning of this document.
>
>
> **3. The third one is that the analysis doesn't feel very original. The main difference from the cited existing papers from my understanding are a linear mapping and a multiplication of $LL^+$. Also, the results rely on the relationship between the smallest and largest eigenvalue of the graph Laplacian, and I don’t know how many network graphs can meet those requirements.**
>
>
> For the first comment, the main difference with other distributed TD-learning algorithms is that our algorithm is based on a primal-dual method, and it works in more relaxed and general scenarios, i.e., it does not require that the underlying communication network graph is characterized by a doubly stochastic matrix. We would like to note that the analysis of primal-dual method with null space constraints and Markovian observation model is not a trivial task, and it requires much more intricate analysis procedures compared to the existing analysis. Moreover, we believe that the analysis in this paper provides additional insights on utilizing the properties of Moore-pseudo inverse.
>
> For the second comment, the condition on the network graph structure is minimal in the context of the related literature (all the papers mentioned in Table 1) that it needs to be connected, undirected and without any loops. Moreover, the minimal condition on the proposed step-size to guarantee convergence only depends on the number of agents, $N$, rather than maximum and minimum eigenvalues of the graph Laplacian as a conservative choice. Therefore, many practical problems can meet these conditions. Further detailed discussion is given in the response to question 5.
>
> **4. In the abstract you mentioned your algorithm and method do not require the network structure characterized by a doubly stochastic matrix. But through the whole paper, I didn’t see an introduction to the doubly stochastic matrix and how it is related to communication networks. This is an important contribution of your paper, but I still don’t know what kind of networks correspond to a doubly stochastic matrix and what kind of networks do not. So I don’t know how significant the contribution is.**
>
>  A doubly stochastic matrix corresponding to the communication network can be constructed by assigning positive weights to each edge in the network. Additionally, every column sum and row sum of the doubly stochastic matrix needs to be one. Further details about doubly stochastic matrix can be found in page 8 and Appendix A.2 in the revised manuscript. The limitations of the existing distributed TD-learning algorithms requiring the doubly stochastic matrix mentioned in the General Comment, can be summarized as follows: 1) Extension to directed graph and time-varying graph scenario is challenging ; 2) The performance of the algorithm is sensitive to the choice of doubly stochastic matrix. The related discussions have been newly added in the revised version on page 8. For more detailed discussions on the doubly stochastic matrix, please refer to our General Comment at the beginning of this document.

---

> > ### Author Response · Authors · 2023-11-22
> > **Response to Reviewer u6NQ ( Part 2 )**
> >
> > **5. In the proof of theorem 4.2, when using constant step-size, the convergence results rely on the $\lambda_{\max}$ and $\lambda_{\min}$ of the network graph, and the choice of step-size also is based on the $\lambda_{\max}$. I don’t know how many network graphs can meet the requirements on $\lambda_{\max}$ and $\lambda_{\min}$. Also, you provided an existing bound on $\lambda_{\max}$, so how hard is it to find the $\lambda_{\max}$ and how hard is it to find a working $\eta$ since you require the $\eta \approx 1/ \lambda_{\max}$?**
> >
> > Just for clarification, we note that we do not impose any conditions on $\lambda_{\max}$ and $\lambda_{\min}$ corresponding to the network graph.
> >
> > Considering the literature of distributed TD-learning, since most of the existing works, e.g., [R2,R3,R4,R5], requires the knowledge of eigenvalues of the doubly stochastic matrix corresponding to the network graph to decide the step-size, we believe that requiring the knowledge of maximum and minimum eigenvalues of the graph Laplacian matrix is not a strong assumption in the context of the associated literature in this field. Moreover, although we do not provide detailed algorithms to calculate the step-size parameters, the proposed analysis is still valuable in the sense that it provides the existence of such step-sizes.
> >
> > Depending on the structure of the network graph, there are several ways to derive the upper and lower bounds of $\lambda_{\max}$ [R1]. One typical bound is  $\Delta+1 \leq \lambda_{\max} \leq \max_{(u,v)\in\mathcal{E}} d(u)+d(v)$ where $\Delta$ is the maximum degree of a graph and $d(\cdot)$ denotes the degree of a vertex. Hence, as for the step-size, we can always make a conservative choice for $\lambda_{\max}$ as $2N-2$ because the maximum degree of a vertex will be at most $N-1$, where $N$ is the number of agents. Moreover, we can choose the value for $\eta$ within $\Delta+1$ and $\max_{(u,v)\in\mathcal{E}} d(u)+d(v)$ because the condition on $\eta$ is not a strict requirement for the convergence.
> >
> > **6. The citation when you mentioned total variation distance, ergodic and geometric decaying rate is confusing, you may want to cite the original paper that introduced them instead of a recent paper.**
> >
> > In line with the reviewer's comment, we have modified the word "geometric decaying rate" to "exponentially fast". Just for clarification, the reference we cited is a widely used textbook about Markov chain rather than a recent paper. Please let us know if there should be further modifications.
> >
> > **References**
> >
> > [R1] Zhang, Xiao-Dong. "The Laplacian eigenvalues of graphs: a survey." arXiv preprint arXiv:1111.2897 (2011).
> >
> > [R2] Doan, Thinh T., Siva Theja Maguluri, and Justin Romberg. "Finite-time performance of distributed temporal-difference learning with linear function approximation." SIAM Journal on Mathematics of Data Science 3.1 (2021): 298-320.
> >
> > [R3] Sun, Jun, et al. "Finite-time analysis of decentralized temporal-difference learning with linear function approximation." International Conference on Artificial Intelligence and Statistics. PMLR, 2020.
> >
> > [R4] Zeng, Sihan, Thinh T. Doan, and Justin Romberg. "Finite-Time Convergence Rates of Decentralized Stochastic Approximation With Applications in Multi-Agent and Multi-Task Learning." IEEE Transactions on Automatic Control (2022).
> >
> > [R5] Wang, Gang, et al. "Decentralized TD tracking with linear function approximation and its finite-time analysis." Advances in Neural Information Processing Systems 33 (2020): 13762-13772.

---

### Official Review · Reviewer_ZkTm · 2023-11-01

**Soundness:** 3 good
**Presentation:** 2 fair
**Contribution:** 3 good
**Rating:** 6
**Confidence:** 3

**Summary:**

This paper provides a primal-dual perspective on the distributed TD-learning approach. The paper considers a distributed TD-learning setup where each agent shares its information to the neighbors. The parameter-update step is formulated as the dynamical system and then the paper uses Lyapunov theory to conclude about the stability (or, convergence to the equilibrium).

**Strengths:**

The distributed TD learning is an important question and this paper has provided new insights.

The results seem to be correct.

**Weaknesses:**

1. The paper only considers the average reward scenario. However, there can be another reward scenario (cooperative or competitive), can the result be extended to those setups?

2. There is already quite a bit of work on the multi-agent RL framework for the average-reward case. Please see [A1]. The authors should discuss both in terms of methodology and the results whether they are related or different. The above paper provides the sample complexity bound, and even consider general function approximation case.

[A1]. Hairi, F. N. U., Jia Liu, and Songtao Lu. "Finite-time convergence and sample complexity of multi-agent actor-critic reinforcement learning with average reward." In International Conference on Learning Representations. 2021.

3. In terms of practicality of the algorithm, there is an inherent assumption that each agent has the same feature space $\phi$, however, this might not be true in practice.

**Questions:**

1. Can the authors emphasize more on the technical challenges?

---

> ### Author Response · Authors · 2023-11-22
> **Response to Reviewer ZkTm**
>
> We are grateful for the reviewer's valuable feedback. We have provided detailed responses to the reviewer's questions and comments.
>
> **1. The paper only considers the average reward scenario. However, there can be another reward scenario (cooperative or competitive), can the result be extended to those setups?**
>
> We believe that our analysis can be extended to other setups as long as the reward scenario can be achieved at the consensus of parameters of each agents. For instance, one can consider different scenarios (cooperative or competitive), where communications among the agents are allowed. In these scenarios, we can consider new distributed algorithms which apply the proposed strategies, which can be potential future topics.
>
>
> **2. There is already quite a bit of work on the multi-agent RL framework for the average-reward case. Please see [A1]. The authors should discuss both in terms of methodology and the results whether they are related or different. The above paper provides the sample complexity bound, and even consider general function approximation case.**
>
> Thank you for pointing out the work, and we have added discussion about [A1] in the revised manuscript. To our best knowledge, the critic part (which corresponds to the TD-learning algorithm) in [A1], uses linear function approximation rather than general function approximation. The algorithm in [A1] uses mini-batch update, where the size of mini-batch depends on number of agent $N$. We believe that our algorithm can be also improved using such mini-batch style updates. Moreover, we can also derive the sample complexity $\mathcal{O}(1/\epsilon)$ or $\mathcal{O}(\log (1/\epsilon)/\epsilon)$ from the mean squared error bound in Theorem 4.2 and Theorem 4.3, respectively. Note that the sample complexity in [A1] is also derived from the mean squared error bound.
>
> **3. In terms of practicality of the algorithm, there is an inherent assumption that each agent has the same feature space $\Phi$ however, this might not be true in practice.**
>
> Indeed, we agree with the reviewer. However, considering the scenarios such as Go and Chess, every agent shares the same feature spaces. Moreover, we believe that the extension to different feature spaces should be possible without difficult technical challenges. Furthermore, most of the algorithms mentioned in Table 1 in the manuscript, except [R1], assume that the agents share the same feature space.
>
> **4. Can the authors emphasize more on the technical challenges?**
>
> The main technical challenge is to derive convergence rates of a stochastic algorithm to a null space rather than a single unique point. This challenge can be addressed by following the analysis of primal-dual ODE with null space constraint introduced in Section 3. Moreover, the analysis of the primal-dual method with null space constraint and Markovian observation model requires much more intricate analysis steps compared to existing analysis.
>
> **References**
>
> [A1]. Hairi, F. N. U., Jia Liu, and Songtao Lu. "Finite-time convergence and sample complexity of multi-agent actor-critic reinforcement learning with average reward." In International Conference on Learning Representations. 2021.
>
>
> [R1] Zeng, Sihan, Thinh T. Doan, and Justin Romberg. "Finite-Time Convergence Rates of Decentralized Stochastic Approximation With Applications in Multi-Agent and Multi-Task Learning." IEEE Transactions on Automatic Control (2022).

---

### Official Review · Reviewer_3CVE · 2023-11-03

**Soundness:** 2 fair
**Presentation:** 2 fair
**Contribution:** 2 fair
**Rating:** 5
**Confidence:** 4

**Summary:**

The paper studies the TD learning for a networked multi-agent Markov decision process. The authors use exponential stability of primal-dual ODE dynamics to study the convergence of TD learning. The authors characterize the solution error rates in both iid and Markovian sampling cases. A numerical example is used to show the performance.

**Strengths:**

- The paper is well organized and key results are explained well.

- The authors provide a nice review of recent works on the exponential stability of primal-dual ODE dynamics when the constraint matrix is rank-deficient.

- The authors study the exponential stability of a primal-dual ODE dynamics, which has improved the dependence on problem parameter.

- The authors propose a new distributed TD learning algorithms, and characterize the solution error rates in both iid and Markovian sampling cases, which has weaker assumptions compared to other distributed TD learning algorithms.

**Weaknesses:**

- The exponential stability of primal-dual ODE dynamics is known in the literature when the constraint matrix is rank-deficient. The improvement is only some constant for a special case of objective and constraint functions, which might be not very important to the TD analysis.

- The proposed distributed TD learning is based on a known distributed primal-dual ODE dynamics. The error rate analyses follow the Lyapunov-based analysis from the previous work. The technique novelty is questionable.

- The conducted primal-dual ODE based analysis can only guarantee mean-path performance, which might be not very useful in practice due to large variance.

- The provided example is artificial, and there is no comparison with existing distributed TD algorithms.

**Questions:**

- Is a missing $\mathbf{P}^\pi$ in projected Bellman equation?

- Can the authors provide numerical experiments to justify the convergence rates in Theorem 3.2? Why do you have an improvement?

- Can the authors explain more how the new TD learning algorithm is built on the result of Wang and Elia (2011)? When does strongly convexity hold?

- The dependence of solution error rates on problem parameters is not clearly explained. What are parameters $w$, $h_1$, $h_2$ in Theorem 4.2 and Theorem 4.3?

- Can the authors conduct comparison experiments with other existing algorithms?

---

> ### Author Response · Authors · 2023-11-22
> **Response to Reviewer 3CVE ( Part 1)**
>
> We are grateful for the reviewer's valuable feedback. We have provided detailed responses to the reviewer's questions and comments.
>
> **1. The exponential stability of primal-dual ODE dynamics is known in the literature when the constraint matrix is rank-deficient. The improvement is only some constant for a special case of objective and constraint functions, which might be not very important to the TD analysis.**
>
> For the first comment, to the authors' knowledge, the exponential stability of primal-dual ODE dynamics when the constraint matrix is rank-deficient has not been fully studied so far. Moreover, as pointed out by the reivewer, while the improvement may be restricted to special scenarios, it offers different insight on utilizing the properties of Moore-pseudo inverse. Furthermore, it provides simplified tool for the analysis of the suggested distributed TD-learning.
>
>
> **2. The proposed distributed TD learning is based on a known distributed primal-dual ODE dynamics. The error rate analyses follow the Lyapunov-based analysis from the previous work. The technique novelty is questionable.**
>
> The distributed primal-dual ODE dynamics studied in the current work is different from those from the existing works in the literature in the sense that the proposed ODE dynamics involves the null space constraints. Therefore, extending the distributed ODE dynamics to a stochastic setting via stochastic approximation methods is non-trivial because it requires to show convergence to a null space rather than a unique fixed point. Moreover, the analysis of primal-dual method with null space constraint and Markovain observation model requires far more intricate analysis procedures compared to existing approaches.
>
>
> **3. The conducted primal-dual ODE based analysis can only guarantee mean-path performance, which might be not very useful in practice due to large variance.**
>
> We would like to note that the primal-dual ODE based analysis conducted in this paper not only guarantees the mean-path performance but also is used for the convergence properties of its stochastic versions (distributed TD-learning), which are given in the second part of the paper. The mean-path part can be seen as a contribution of our paper but it can be seen as a preliminary step for its stochastic version (distributed TD-learning).
>
> Moreover, the majority of the theoretical works on analysis of TD-learning [R1,R2] including most of the distributed TD-learning which are listed in the Table 1 in the manuscript, also deal with the mean-path setting first and investigate its stochastic versions.
>
> **4. The provided example is artificial, and there is no comparison with existing distributed TD algorithms.**
>
> According to the reviewer's comment, we have newly added experiments on comparisons with the existing distributed TD algorithms in Appendix A.13 in the revised version. Further details about the experiments are given in the General Comment.
>
> **5. Is a missing $P^{\pi} $in projected Bellman equation?**
>
> Thank you for pointing out the error, which has been corrected in the revised version.
>
> **6. Can the authors provide numerical experiments to justify the convergence rates in Theorem 3.2? Why do you have an improvement?**
>
> The convergence rate can be verified through the full plot in Figures~(5a) and~(5b) in the revised manuscript, which demonstrates the exponential convergence rates for constant step-sizes. The linear convergence rate with diminishing step-sizes can be verified in the Figures (1b) and (2c).
>
> We would like to note that we do not claim improvement on the convergence rate, $\mathcal{O}(exp(-k))$ or $\mathcal
> O(1/k)$ with the constant and diminishing step-sizes, respectively. Nevertheless, the convergence rate of the suggested algorithm is comparable with the convergence rates of the existing works with the weaker assumption that it does not require the full rank doubly stochastic matrix structure corresponding to the underlying network graph.
>
> **7. Can the authors explain more how the new TD learning algorithm is built on the result of Wang and Elia (2011)? When does strongly convexity hold?**
>
> The original work of Wang-Elia studied distributed optimization algorithms in continuous-time domain with a general objective function $f(x)$ and its gradient $\nabla f(x)$. We replace $\nabla f(x)$ with stochastic TD-error $\phi_k (r_k+\gamma \phi_k^{\prime\top}\theta_k-\phi_k^{\top}\theta_k) $, which is, however, not a (stochastic) gradient of any objective function.
>
>
> For the second question, we do not require the strong convexity. We only require the matrix $A$ to be negative definite, i.e., $A+A^{\top}\prec 0$ where $A$ can be an asymmetric matrix. TD-learning is a stochastic approximation scheme finding a fixed point of the equation $Ax=b$ rather than using a gradient of a certain objective function.

---

> > ### Author Response · Authors · 2023-11-22
> > **Response to Reviewer 3CVE ( Part 2 )**
> >
> > **8. The dependence of solution error rates on problem parameters is not clearly explained. What are parameters $w, h_1, h_2$
> >  in Theorem 4.2 and Theorem 4.3?**
> >
> > The error bound depends on the graph structure (maximum and minimum eigenvalue of the graph Laplacian) and dynamics of TD-learning. As in the standard TD-learning, we can observe that the error bound of the proposed distributed TD is dependent on the problem parameters $w$ and $\frac{1}{1-\gamma}$, where $w$ is the minimum eigenvalue of the matrix $A:=\gamma \Phi^{\top}DP^{\pi}\Phi - \Phi^{\top}D\Phi$, which characterizes the dynamics of TD-learning. Moreover, $h_1$ and $h_2$ are the parameters in thes step-size, $\alpha_k=\frac{h_1}{k+h_2}$, which depends on the maximum and minimum eigenvalue for the graph, and the overall bound in Theorem 4.3 is expressed by replacing $h_1$ and $h_2$ with the problem dependent parameters.
> >
> > **9. Can the authors conduct comparison experiments with other existing algorithms?**
> >
> > Following the reviewer's recommendation, we have added several more experiments in Appendix A.13. As from our experiment, the performance of the existing distributed TD algorithms that require doubly the stochastic matrix structure of the underlying graph Laplacian matrix is sensitive to the choice of the stochastic matrix. Furthermore, there is no one specific algorithm that consistently outperforms the others.
> >
> > **References**
> >
> > [R1] Srikant, Rayadurgam, and Lei Ying. "Finite-time error bounds for linear stochastic approximation andtd learning." Conference on Learning Theory. PMLR, 2019.
> >
> > [R2] Bhandari, Jalaj, Daniel Russo, and Raghav Singal. "A finite time analysis of temporal difference learning with linear function approximation." Conference on learning theory. PMLR, 2018.

---

> ### Comment · Reviewer_3CVE · 2023-11-22
>
> Thank you for the rebuttal. I don't have any further questions. One suggestion is improving experiments that can show the merits in more realistic examples.

---

> > ### Author Response · Authors · 2023-11-23
> > **Response to Reviewer 3CVE**
> >
> > We are glad to hear that most of the concerns have been addressed. In case the concerns have been addressed and the questions have been answered, we would be grateful if the reviewer could consider reflecting this in rating of our paper.  We will consider more realistic examples to improve the experiments. We thank again the reviewer for the time and effort in reviewing our manuscript.

---

### Author Response · Authors · 2023-11-22
**General Comments**

Dear all reviewers, we appreciate the fruitful and constructive comments to improve the manuscript. We listed general comments and changes in the manuscript. The changes in the revised manuscript are marked with ${\color{red} colored}$ fonts.

**Remarks on the doubly stochastic matrix:**

One of the key advantages of our algorithm over the other existing distributed TD algorithms is that it does not require a doubly stochastic matrix. We have outlined several reasons highlighting the importance of removing the requirement on the doubly stochastic matrix structure, and the limitations of the algorithms requiring doubly stochastic matrix:

To begin, constructing a doubly stochastic matrix in directed graph scenario is known to be more challenging than the undirected case, or may not be possible [R3]. However, our algorithm can be extended to the directed graph setting without major modifications.

Moreover, when dealing with a time-varying graph, whenever the graph changes, the doubly stochastic matrix needs to be constructed again. However, our analysis can be easily extended to the time-varying graph setting without any modifications.

Lastly, as from our experiment, the performance of distributed TD algorithms using doubly stochastic matrix is quite sensitive to the choice of doubly stochastic matrix, and the results can be found in Appendix A.13 in the revised version.

**Experiments:**

1. We have added comparison with existing algorithms including [R4] and [R5] in Appendix A.13. The other algorithms in Table 1 in the manuscript are based on the algorithm in [R5]. As from the experiment, performance of distributed TD algorithms that require doubly stochastic matrix is sensitive to the choice of doubly stochastic matrix, while our algorithm does not require such choice.

2. We have added experimental result to verify the convergence under diminishing step-size case in Figure (1b) and (2c).


3. We have added full plots of the experiments for Figure (1a) and Figure (2b) in Appendix A.14. Moreover, We have modified the experiment (Figure (2b) in the original manuscript) to use the same MDP as in the experiment in Figure (1a), since the choice of reward function was artificial.


**References**

[R1] P. A. Knight. The sinkhorn–knopp algorithm: convergence and applications. SIAM Journal on
Matrix Analysis and Applications, 30(1):261–275, 2008.

[R2] Z.-J. Bai, D. Chu, and R. C. Tan. Computing the nearest doubly stochastic matrix with a prescribed
entry. SIAM Journal on Scientific Computing, 29(2):635–655, 2007


[R3] Xin, Ran, and Usman A. Khan. "A linear algorithm for optimization over directed graphs with geometric convergence." IEEE Control Systems Letters 2.3 (2018): 315-320.

[R4] Wang, Gang, et al. "Decentralized TD tracking with linear function approximation and its finite-time analysis." Advances in Neural Information Processing Systems 33 (2020): 13762-13772.

[R5] Doan, Thinh T., Siva Theja Maguluri, and Justin Romberg. "Finite-time performance of distributed temporal-difference learning with linear function approximation." SIAM Journal on Mathematics of Data Science 3.1 (2021): 298-320.

---

### Meta-Review · Area_Chair_xCPt · 2023-12-07

**Metareview:**

The main point cited as the contribution of this paper -- the lack of a doubly stochastic matrix -- does not seem like a strong enough contribution to justify publication in ICLR, but more like a technical point of interest to other theorists.  It is possible that a more compelling motivation can be found for these result, but the discussion in the abstract and the comparison in Table 1 do not make for a strong case for acceptance. Similarly, points 1-3 on pages 1/2 do not make a strong enough statement of improvement relative to the previous work. It is possible that this work can be justified (e.g., by further exploiting the extra freedom the lack of double stochasticity affords) but this needs to be demonstrated before this work can be accepted to ICLR.

**Justification For Why Not Higher Score:**

The significance of the improvement has not been sufficiently demonstrated.

**Justification For Why Not Lower Score:**

N/A

---

### Decision · Program_Chairs · 2024-01-16

Reject